# Multi-level meta-reinforcement learning with skill-based curriculum

## Abstract

We consider problems in sequential decision making with natural multi-level structure, where sub-tasks are assembled together to accomplish complex goals. Systematically inferring and leveraging hierarchical structure has remained a longstanding challenge; we describe an efficient multi-level procedure for repeatedly compressing Markov decision processes (MDPs), wherein a parametric family of policies at one level is treated as a action in the compressed MDPs at higher levels, while preserving the semantic meanings and structure of the original MDP, and mimicking the natural logic to address a complex MDP. Higher-level MDPs are themselves independent, deterministic MDPs, and may be solved using existing algorithms. The multi-level representation delivered by this procedure decouples sub-tasks from each other and usually greatly reduces unnecessary stochasticity and the policy search space, leading to fewer iterations and computations when solving the MDPs. A second fundamental aspect of this work is that these multi-level decompositions plus the factorization of policies into embeddings (problem-specific) and skills (including higher-order functions) yield new transfer opportunities of skills across different problems and different levels. This whole process is framed within curriculum learning, wherein a teacher organizes the student agent's learning process in a way that gradually increases the difficulty of tasks and ensures the abundance of transfer opportunities across different MDPs and different levels within/across curricula. The consistency of this new, general framework and its benefits brought by these multi-level structures and abundant transfer learning opportunities can in general be justified under mild assumptions. We demonstrate abstraction, transferability, and curriculum learning in some illustrative domains, including a more complex version of the MazeBase example.

## 1 Introduction

Discovering and exploiting *multi-level structure* is a longstanding challenge in sequential decision making. Classical hierarchical RL (HRL) decomposes tasks into reusable sub-policies—e.g., options and semi-Markov abstractions (Sutton et al., 1999; Dietterich, 2000; Parr & Russell, 1997; Dayan & Hinton, 1993). These advances are often restricted to one–two levels or rely on hand-specified subgoals, which can hinder principled planning and transfer at scale (Barto & Mahadevan, 2003). Recent deep HRL automates parts of this pipeline (option-critic, FeUdal, HIRO, HAC) yet commonly fixes depth, mixes high/low-level reasoning, or depends on brittle goal-space design (Bac; Vezhnevets et al., 2017; Nachum et al., 2018; Levy et al., 2019; Dwiel et al., 2019).

**Our contribution.** We introduce a teacher–student–assistant meta-RL framework that (i) repeatedly compresses MDPs so that parametric families of policies at one level become single, abstract actions, yet having semantic meaning, at the next level, yielding deterministic higher-level MDPs, which are easier to solve and that can capture the high-level logic needed in complex problems; (ii) factors policies into embeddings and skills to enable transfer across levels and across MDPs, accelerating solutions and enabling the creation of a dictionary of highly reusable policies, at different levels of abstraction, to be applied as new problems are encountered; and (iii) organizes learning as a skill-based curriculum aligned with these abstractions. This unified view yields fewer iterations at lower per-iteration cost when used in conjunction with existing optimization algorithms such as value iteration, scaling in sparse-reward domains where many prior approaches struggle.

**Context and relationships with related work.** A thread in HRL learns skills *without* external rewards to build general-purpose primitives (Eysenbach et al., 2019; Gregor et al., 2016). While broadly useful, naïve long-horizon composition can reintroduce stochasticity and tangled credit assignment; our multi-level compression *decouples sub-tasks and reduces stochasticity* at higher levels. *Abstraction* further motivates compression, with each compression step *preserves the semantics* of the original MDP while shrinking variance and branching, so solving a long-horizon task reduces to a stack of cleaner subproblems solvable with standard methods (Puterman, 1994). We also exploit natural tensorization of action factors and function composition to induce transferable, high-level behaviors. This is conceptually very different from spectral and spatial techniques (Mahadevan & Maggioni, 2007; Machado et al., 2017; Dean & Givan, 1997; Li et al., 2006; Rav).

*Curriculum learning* (Narvekar et al., 2020; Bengio et al., 2009; Mat; Florensa et al., 2017; Sukhbaatar et al., 2017; 2018b) supplies the second pillar. *Many curricula operationalize difficulty by time to solve rather than time to learn, and often restrict the final task to a concatenation of subtasks.* Our *compression-aligned* curriculum instead defines difficulty via natural human logic, mirroring how humans tackle complex tasks and improve optimization and transfer in sparse-reward regimes, with higher levels coarsening space/time while broadening scope as a natural byproduct.

*Transfer* is the third pillar, providing repid learning and policy improvement across related tasks (Barreto et al., 2017; Bar; Andreas et al., 2017; Rusu et al., 2016). We factor policies into *embeddings* (problem-specific perception/featurization) and *skills* (including reusable higher-order functions) and then compress families of such policies into single abstract actions. This supports *transfer across levels and across MDPs, both within and across curricula*—even when state spaces differ—without relying on rote replay, targeting semantic reuse of skills rather than state memorization, without the need of storing and revisiting states (Ecoffet et al., 2021).

Recent evidence from meta-learning for compositionality shows that optimizing a standard neural network for compositional skills yields human-like systematic generalization Lake & Baroni (2023). Because multi-level structure is a prior over *families* of problems, our work connects to meta-RL (Duan et al., 2016; Finn et al., 2017; Rakelly et al., 2019; Frans et al., 2018). Our compression and policy factorization provides constructive meta-generalization: higher levels expose slower, more stable dynamics, while lower levels encapsulate fast feedback, enabling cross-task reuse.

Finally, our framework is compatible with inverse reinforcement learning (IRL) and imitation Ng & Russell (2000); Abbeel & Ng (2004); Ziebart et al. (2008); Finn et al. (2016); Ho & Ermon (2016); Fu et al. (2018), and Hierarchical IRL Krishnan et al. (2016). Because each compression step yields an independent, semantically preserved MDP, we can invert the process: estimate rewards or subgoal structures at appropriate levels, then learn skills and curricula consistent with demonstrations, improving sample-efficiency and interpretability relative to flat IRL Adams et al. (2022).

*An extended literature review, further connections and comparisons are discussed in Appendix A.*

## 2 MULTI-LEVEL MARKOV DECISION PROCESSES

**Basic definitions**. A **Markov decision process (MDP)** is a sequential decision making process where the agent interacts with the environment and learns how to maximize the cumulative rewards from such interactions. An MDP is modeled as $\texttt{MDP} := (\mathcal{S}, \mathcal{S}^{\text{init}}, \mathcal{S}^{\text{end}}, \mathcal{A}, P, R, \Gamma)$, where: the **state space** $\mathcal{S}$, with its distinguished subsets $\mathcal{S}^{\text{init}}$ and $\mathcal{S}^{\text{end}}$ of initial and terminal states; the **action set** $\mathcal{A}$; the **transition probabilities** $P$, corresponding **rewards** $R$ and **discount factors** $\Gamma$. We assume $\mathcal{A}$ is a subset of the product $\mathcal{A}_1 \times \cdots \mathcal{A}_K$ of the actions factors $\mathcal{A}_k$, each containing a universal element $a^{\text{end}}$ indicating the "end" action. Given an active action factor set $I \subseteq [K]$, we "restrict" $a \in \mathcal{A}$ to $a_I$ by setting to the special value 0, an "action" that cannot be taken, the coordinates of $a$ corresponding to any $\mathcal{A}_k$'s with $k \notin I$, and let $\mathcal{A}_I$ be the image of $\mathcal{A}$ under such map. See App. B.1 for detailed definitions. A **policy** $\pi : \mathcal{SA} \to [0, 1]$ is, for each $(s, a) \in \mathcal{SA}$, the probability of selecting $a$ when the agent is in state $s$, and satisfies $\sum_{a \in \mathcal{A}(s)} \pi(s, a) = 1$ for any $s \in \mathcal{S}$. We may assume that $\pi(s, a) = \mathbb{1}_{\{a^{\text{end}}\}}(a)$ for any $s \in \mathcal{S}^{\text{end}}$. Solving an MDP means learning an optimal policy $\pi_*$ maximizing the long-term cumulative rewards, summarized by the **value function** $V_\pi(s) := \mathbb{E}_{\tau, S_{1:\tau}, A_{0:\tau-1}}[R_{0,\tau} | S_0 = s, A_{0:\tau-1} \sim \pi]$, for any $s \in \mathcal{S}^{\text{init}}$. Here, for any two (random) times $T, T'$, $R_{T,T'}$ is the discounted reward accumulated over the interval $[T, T']$ (equation B.1). The optimal policy $\pi_*$ is independent of the initial state $S_0$, by Markovianity.

**Box 1: Geometric configuration and MDPs in the Mazebase+ example**

We introduce the MazeBase+ example to walk the reader through the concepts and techniques we introduce; it is more challenging than MazeBase, a well-known example in the literature on hierarchical RL and curriculum learning, including Sukhbaatar et al. (2018a;c). We solve it with a curriculum containing a (stochastic) MDP of difficulty 3, showcasing the role of the teacher, the assistant, and the student (the actual learning agent), and of curricula, transfer learning, and abstractions in our framework, and building on another example, "navigation and transportation with traffic jams" (introduced later). We show minimal examples to convey the key ideas; we could easily extend them to larger grid worlds, where the computational reduction brought by our framework would be even larger thanks to multi-scale compression.

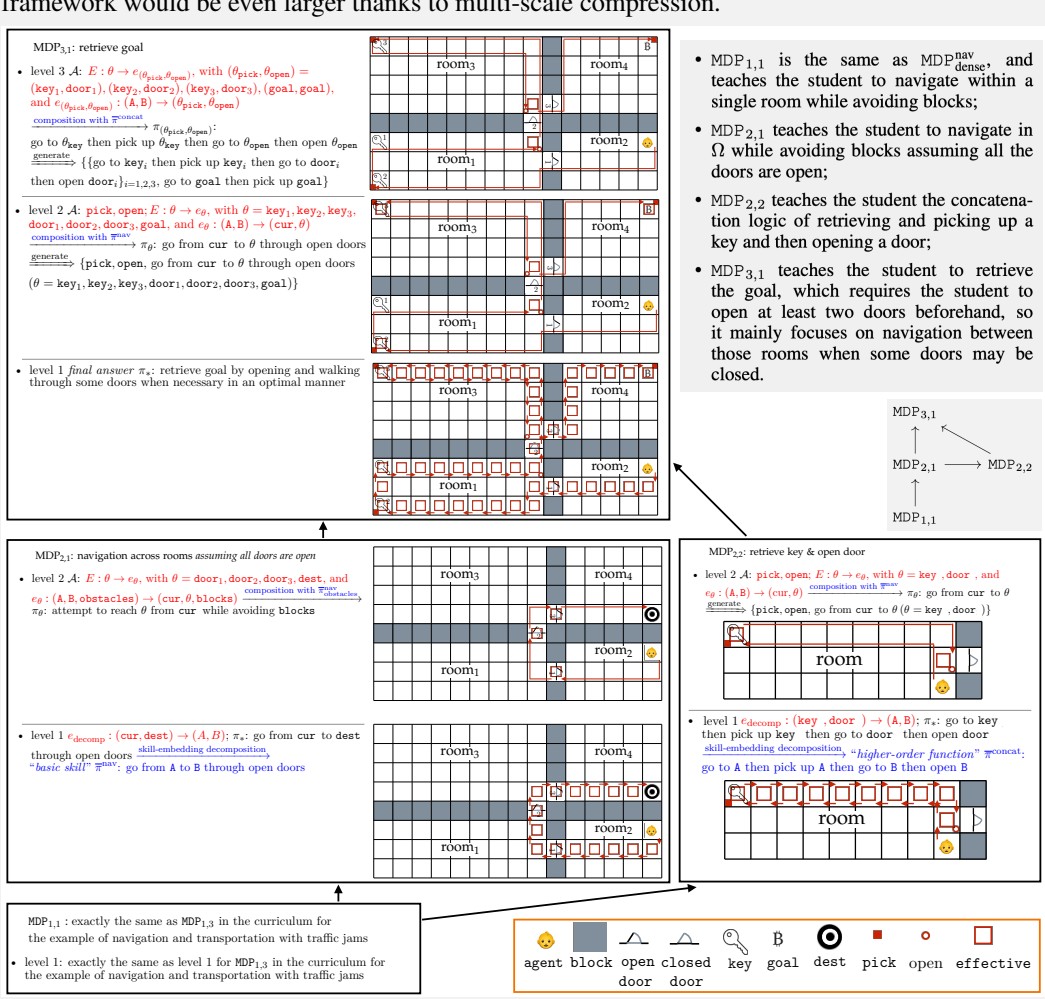

Figure 1: A representation of the first experiment in MazeBase+, detailed in the current Box 1.

We conduct 3 MazeBase+ experiments: (1) Solving MazeBase+ (Box 1, Fig. 1) using a curriculum (Sec. 4), a minimal ordered MDP family that helps the student agent develop skills at varying abstraction levels for efficient solution of a complex MDP; (2) efficient transfer to MazeBase+ worlds with different room, door, key, and goal configurations (Box 3, Fig. 2); (3) demonstrating robustness when the highest-level optimal policy requires significant refinement at finer levels but outperforms naïve value iteration (Box 13, Fig. 5).

The figure illustrates MazeBase+, showing initial door states and with matching door-key indices. The agent navigates a grid world from an initial position to a goal, needing to pick up keys to open doors across multiple rooms. We map this MDP to a three-level MMDP, using a curriculum (top-right figure inset) with four MDPs, $\mathrm{MDP}_{1,1}$, $\mathrm{MDP}_{2,1}$, $\mathrm{MDP}_{2,2}$, and $\mathrm{MDP}_{3,1}$, of difficulties 1, 2, 2, and 3, respectively, and in this order, from the bottom up, provided by the teacher.

MDP$_{1,1}$ teaches the student to navigate within a single room; this is the same as MDP$_{\text{dense}}^{\text{nav}}$ in the example of navigation and transportation with traffic jams (sec. 4), with blocks and doors mapped to a subset of traffic jams. In that MDP we learn the skill $\overline{\pi}_{\text{dense}}^{\text{nav}}$ that teaches the student to navigate while avoiding traffic jams as much as possible. Here we solve MDP$_{1,1}$ by transferring that skill, now ready to be utilized as a single action in MDP$_{2,1}$.

MDP$_{2,1}$ teaches the student to navigate all of $\Omega$ while avoiding blocks, assuming all the doors are open. The agent learns the optimal policy for this problem, which becomes a higher-order function $\overline{\pi}^{\text{nav}}$, to be utilized in MDP$_{2,2}$ and MDP$_{3,1}$.

MDP$_{2,2}$ teaches the student the concatenation logic of retrieving and picking up a key and then opening the corresponding door: the optimal policy for MDP$_{2,2}$ becomes a higher-order function $\overline{\pi}^{\text{concat}}$, to be utilized in MDP$_{3,1}$.

In MDP$_{3,1}$, the student learns to reach a goal by navigating rooms and opening doors.

Actions at this higher level correspond to entire skills at finer levels, composed/decomposed via embeddings (distinct arrows, detailed later), forming long, complex paths (long red arrows). Each skill is a parametric family of policies. At higher levels, the effective state space (hollow red squares, final states of actions) is significantly smaller than at level 1, reducing the policy search space and speeding up MDP solutions. Actions and policies have semantic, interpretable names. The policy from an initial state is shown with red arrows (actions) and red squares (effective states), highlighting the reduced state space and action count at higher levels. For example, in level 2 of MDP$_{3,1}$ (middle row), the agent navigates from room$_2$ to room$_1$ to key$_2$ using $\overline{\pi}^{\text{nav}}$, picks up key$_2$ (filled red square), navigates to door$_2$, and opens it with key$_2$ (empty red circle), reaching state $s_2$. At level 3 (top row), a single action using $\overline{\pi}^{\text{concat}}$, an abstract concatenation skill, reaches $s_2$, with arrow endpoints touching a filled red square or an empty red circle.

Different colors of text correspond to different roles in our framework, namely teacher (red), assistant (blue), and student (black). Note that door$_1$ does not need to be opened, so it is just a "confounder"; clearly an optimal policy avoids picking up key$_1$ and trying to open door$_1$.

See App. E.1.2 for the formal definition of these MDPs. In particular, observe that there are many shared components between the target MDP$_{3,1}$ and other MDPs, including the skill of navigation through doors, and the concatenation logic behind opening a door. In order to efficiently solve this family of problems, we exploit the similarities between them to enable potential transfer.

We now exploit the tensor product structure underlying the action space to introduce policies that use only a few action factors, are easier to learn and transfer, and can be combined (using the notion of outer products) to obtain general policies. We define a **partial policy** as follows: for each $I \subset [K]$, a **partial policy** $\pi_I : \mathcal{SA}_I \to [0,1]$ is, for each $(s, a_I) \in \mathcal{SA}_I$, the probability of selecting $a_I \in \mathcal{A}(s)$ when the agent is in state $s$, and satisfies the normalization property $\sum_{a \in \mathcal{A}_I(s)} \pi_I(s,a) = 1$ for any $s \in \mathcal{S}$. When $I = [K]$, $\pi_I$ is a policy in the usual sense; otherwise, a partial policy does not prescribe valid/useful actions for the agent, as it prescribes actions with some factors equaling to 0.

We now introduce partial policy generators to allow the teacher to provide hints to the student on how to potentially restrict the search of an optimal policy to subsets of combinations of partial policies. A **partial policy generator** $g_I$ maps a parameter set $\Theta$ to the set of all partial policies $\{\pi_I\}$, yielding a parametric family of partial policies $\{g_I(\theta) : \theta \in \Theta\}$. Given a finite set $\mathcal{G} = \{(g_1)_{I_1}, \cdots, (g_M)_{I_M}\}$ of partial policy generators (not necessarily sharing the same active action factor set), we define the **set of partial policies generated from** $\mathcal{G}$ as $\widetilde{\Pi}_{\mathcal{G}} := \cup_{m=1}^{M} \{(g_m)_{I_m}(\theta) : \theta \in \Theta_m\}$. The action factors and partial policy generators for the MazeBase+ example are reported in Box 5.

In order to construct valid policies from partial policies, we need to appropriately "combine" partial policies acting along mutually disjoint action factors that overall "cover" all possible action factors. Given two partial policies $\pi_I, \pi'_{I'}$ with $I \cap I' = \varnothing$, we define their **outer-product** as the improper partial policy $\pi_I \otimes \pi'_{I'} : \{(s, (a_I, a_{I'})) : s \in \mathcal{S}, a_I \in \mathcal{A}_I(s), a_{I'} \in \mathcal{A}_{I'}(s)\} \to [0,1]$ satisfying

$$(\pi_I \otimes \pi'_{I'})(s, (a_I, a_{I'})) := \pi_I(s, a_I) \times \pi'_{I'}(s, a_{I'}). \tag{2.1}$$

We call this "improper" because a (proper) partial policy is defined on $\mathcal{SA}_{I \cup I'}$, but $(s, (a_I, a_{I'}))$ may not belong to $\mathcal{SA}_{I \cup I'}$, since in general $\mathcal{A}_{I \cup I'}(s) \nsubseteq \mathcal{A}_I(s) \times \mathcal{A}_{I'}(s)$. However, we can map every improper partial policy to a proper one by restricting it to $\mathcal{SA}_{I \cup I'}$ and renormalizing it. This

outer product operator, with or without normalization, is commutative and associative, so from equation 2.1 we can define a unique outer product of multiple partial policies with pairwise disjoint active action factor sets. Given the set $\widetilde{\Pi}_{\mathcal{G}}$ of partial policies generated by the finite partial policy generator set $\mathcal{G}$, we define the set of polices $\Pi_{\mathcal{G}}$ generated by $\mathcal{G}$ as the union of all policies $\pi$ that are the outer product of finitely many partial policies in $\widetilde{\Pi}_{\mathcal{G}}$ restricted to $\mathcal{A}(s)$ and then normalized. See App. B.2 for the formal definitions and equations.

We refer the reader to Box 6 for the partial policy generator sets in the MazeBase+ example.

## 2.1 Multi-level Markov Decision Processes (MMDPs)

We construct an MMDP from a single $\texttt{MDP} := (\mathcal{S}, \mathcal{S}^{\text{init}}, \mathcal{S}^{\text{end}}, \mathcal{A}, P, R, \Gamma)$, comprising MDPs with the same state space $\mathcal{S}$ but multiple levels of increasing abstraction, enabling more powerful actions at higher levels. For example, level-two actions are defined using a partial policy generator $\mathcal{G}^1$ for the original MDP (level one), with the action set $\mathcal{A}^2(s) := \overline{\Pi_{\mathcal{G}^1}}$. Taking action $a \in \mathcal{A}^2(s)$ at state $s$ involves running policy $a \in \Pi_{\mathcal{G}^1}$ from $s$ for a specified duration, typically longer at higher levels, where each action corresponds to a sequence of level-one actions from $\mathcal{G}^1$. This construction iterates for higher levels, facilitating faster MDP solutions and transfer across multiple MDPs.

**Definition 2.1.** A **sequence of generator sets** is a sequence of finite partial policy generator sets $\{\mathcal{G}^l\}_{l=1}^{\infty}$ on an $\texttt{MDP}$ that satisfies the following: $\mathcal{G}^l$ is defined on $\mathcal{S}\mathcal{A}^l$ for any $l \in \mathbb{Z}^+$, where $\mathcal{A}^1 := \mathcal{A}$ and $\mathcal{A}^{l+1}(s) := \overline{\Pi_{\mathcal{G}^l}}$ for $l \geq 1$ and all $s \in \mathcal{S}$. We let $\Pi^l := \Pi_{\mathcal{G}^l}$.

The inputs needed for the construction of an MMDP from a given MDP are the following:

1. A **sequence of generator sets** $\{\mathcal{G}^l\}$ that naturally gives a sequence of state space-action set pairs $\{\mathcal{S}\mathcal{A}^l\}_{l=1}^{\infty}$. In addition, the teacher provides another sequence of finite partial policy generator sets $\{\mathcal{G}^l_{\text{test}}\}_{l=1}^{\infty}$ with $\mathcal{G}^l_{\text{test}}$ defined on $\mathcal{S}\mathcal{A}^l$ for any $l \in \mathbb{Z}^+$ that is used by the student to assess the level of difficulty of the MDP being solved.

2. A **timescale** $1 \leq t_\pi \leq +\infty$, for each policy $\pi \in (\cup_{l=1}^{\infty} \Pi^l) \cup (\cup_{l=1}^{\infty} \Pi_{\mathcal{G}^l_{\text{test}}})$, giving probability $1/t_\pi$ of terminating at each time step (if $t_\pi = +\infty$, then the probability $1/t_\pi$ is defined to be 0).

3. Finally, a sequence of **negative rewards** $\{r^l\}_{l=1}^{\infty}$, to penalize choosing an action containing $a^{\text{end}}$ at a non-terminal state. In particular, $R(s, a, s) = r^1$ for any $a \in (\mathcal{A}^1)^{\text{end}}$ and $s \notin \mathcal{S}^{\text{end}}$.

We refer the reader to Box 7 for the inputs for the construction of MMDPs in MazeBase+.

We construct an $\texttt{MMDP}$ in an inductive manner with the inputs provided above. At level one we let $\texttt{MDP}^1 := (\mathcal{S}, \mathcal{S}^{\text{init}}, \mathcal{S}^{\text{end}}, \mathcal{A}^1, P^1, R^1, \Gamma^1)$ to be the original given $\texttt{MDP} = (\mathcal{S}, \mathcal{S}^{\text{init}}, \mathcal{S}^{\text{end}}, \mathcal{A}, P, R, \Gamma)$. If the optimal policy $\pi^1_*$ of $\texttt{MDP}^1$ learned by the student is in $\Pi_{\mathcal{G}^1_{\text{test}}}$, then $\texttt{MDP}^1$ is defined to be of difficulty 1, and we are done. Otherwise, we move to higher levels. Inductively, given $\texttt{MDP}^l := (\mathcal{S}, \mathcal{S}^{\text{init}}, \mathcal{S}^{\text{end}}, \mathcal{A}^l, P^l, R^l, \Gamma^l)$, $l \in \mathbb{N}$, we define the $(l+1)$-st level $\texttt{MDP}^{l+1}$ with:

- **State space** $\mathcal{S}$, $\mathcal{S}^{\text{init}} \subseteq \mathcal{S}$, and $\mathcal{S}^{\text{end}} \subseteq \mathcal{S}$ are the same as those of the original $\texttt{MDP}^1$.

- **Action set** $\mathcal{A}^{l+1}(s) := \overline{\Pi^l}$, represented as a subset of $\mathcal{A}^{l+1}_1 \times \mathcal{A}^{l+1}_2 \times \cdots \times \mathcal{A}^{l+1}_{|\mathcal{G}^l|}$, with $\mathcal{A}^{l+1}_k := \Theta^l_k \cup \{\texttt{null}, a^{\text{end}}\}$ for $1 \leq k \leq |\mathcal{G}^l|$, and $\Theta^l_k$ being the domain of the $k$-th generator in $\mathcal{G}^l$.

- **Transition probabilities** $P^{l+1}(s, a^{l+1}, s')$: when $a^{l+1} \in \Pi^l$ is chosen in state $s$, the policy $a^{l+1}$ will be run starting from $s$ till time $\tau := \min\{\text{first time when } a^{\text{end}} \text{ is chosen}, \text{Geo}(1/t_{a^{l+1}})\}$ [1], and $P^{l+1}(s, a^{l+1}, s')$ is defined to be the total probability of terminating at $s'$, equation B.2

- **Rewards** $R^{l+1} : \mathcal{S}\mathcal{A}\mathcal{S}^{l+1} \to \mathbb{R}$ are set to 0 when $s \in \mathcal{S}^{\text{end}}$, otherwise set to $r^{l+1}$ when $a^{l+1} \in (\mathcal{A}^{l+1})^{\text{end}}$. In all other cases, they are defined to be the expected total discounted reward collected along trajectories associated to $(s, a^{l+1})$, as described above, which end at $s'$, see equation B.3. For any two (random) times $0 \leq T < T' < \infty (a.s.)$, the random variable $R^l_{T,T'}$ is the discounted reward accumulated over the interval $[T, T']$ in $\texttt{MDP}^l$, see equation B.4.

---

[1] $\text{Geo}(\lambda)$ is a geometric distribution of parameter $\lambda$; $t_a = \infty$ is allowed and means that with probability 1 such time is $\infty$.

- **Discount factors** $\Gamma^{l+1} : \mathcal{SAS}^{l+1} \to (0,1]$ are set, for $a^{l+1} \in \Pi^l$, to be the expected product of the discounts applied to rewards along trajectories associated to $(s, a^{l+1})$ which end at $s'$, see equation B.5, and similarly for $\Gamma_{T,T'}^l$, see equation B.6.

With $\text{MDP}^{l+1} := (\mathcal{S}, \mathcal{S}^{\text{init}}, \mathcal{S}^{\text{end}}, \mathcal{A}^{l+1}, P^{l+1}, R^{l+1}, \Gamma^{l+1})$ constructed by the student, if the optimal policy $\pi_*^{l+1}$ of $\text{MDP}^{l+1}$ is in $\Pi_{\mathcal{G}_{\text{test}}^{l+1}}$, then $\text{MDP}$ has difficulty $L := l + 1$, and construction ends. Otherwise, higher levels are explored. This completes the MMDP construction: given $\text{MDP} = (\mathcal{S}, \mathcal{S}^{\text{init}}, \mathcal{S}^{\text{end}}, \mathcal{A}, P, R, \Gamma)$, $\{\mathcal{G}^l\}_{l=1}^\infty$, $\{\mathcal{G}_{\text{test}}^l\}_{l=1}^\infty$, $\{t_\pi\}_{\pi \in \cup_{l=1}^\infty (\Pi^l \cup \Pi_{\mathcal{G}_{\text{test}}^l})}$, and $\{r^l\}_{l=1}^\infty$, the difficulty $L$ and MMDP $\{\text{MDP}^l\}_{l=1}^L$ are defined, where each $\text{MDP}^{l+1}$ (from level two) is a compressed abstraction of $\text{MDP}^l$, with consistent rewards and discount factors. This compression, akin to coarsening, homogenization, or lumping, reduces the problem by locally averaging, preserving MDP structure and semantics via partial policies, potentially coarsening spatial or temporal scales as the agent uses more powerful, longer-running actions at higher levels.

**Solving an MMDP**. To solve the original MDP using an MMDP, we employ a bottom-up then top-down procedure. In the bottom-up phase, we construct $\text{MDP}^{l+1}$ from $\text{MDP}^l$. In the top-down phase, we solve the MDP at the highest level $L$, then iteratively refine the solution down to the original MDP. The refinement step "unpacks" the optimal policy $\pi_*^{l+1}$ of $\text{MDP}^{l+1}$ into an initial policy $\pi^l$ for $\text{MDP}^l$ by concatenating action sequences, forming the "convolution" $\pi_*^{l+1} * \mathcal{G}^l$ (see App. B.4). Iterating yields $\tilde{\pi}_*$ at the initial level, which, though not optimal, is close to $\pi_*$ and aids value iteration. Value iteration at each level refines the policy, passed down iteratively to obtain the optimal policy for the original MDP. The MDP's difficulty, defined by level $L$, reflects the number of convolution steps with partial policy generators and value iteration refinements, or the levels of abstraction from compressing $\pi_*$ with $\mathcal{G}^1, \mathcal{G}^2, \dots$ to level $L$.

See Box 8 for an instance of unpacking compressed policies in the MazeBase+ example.

**Multi-level compression**. See App. D.1 for details, where we provide explicit formulas for higher-level transition probabilities, rewards, and discount factors, both in matrix form and in scalar form, and see Box 9 for how this is realized in the MazeBase+ example.

## 3 TRANSFER LEARNING WITH SKILLS AND EMBEDDINGS

Knowledge, represented by policies, is transferred from one MDP to another to accelerate solving the latter. Our MMDP and curriculum-based framework enables transfer of entire MDPs, specific MDPs in the curriculum, or selected levels within them. The key mechanism is skill-embedding decomposition, which factors a policy into an embedding that abstracts aspects of the state-action space and a highly transferable skill that operates on the abstracted output, applicable to multiple new MDPs with compatible abstracted state-action spaces.

**Definition 3.1.** $(\overline{\pi}, e)$ a **skill-embedding decomposition** of a partial policy $\pi_I : \mathcal{SA}_I \to [0,1]$ on $\mathcal{D}$, if $\pi_I(s,a) = \overline{\pi}(e(s,a))$ for any $(s,a) \in \mathcal{D}$, where $\overline{\pi} : \mathcal{E} \to [0,1]$ is the **skill**, $e : \mathcal{D} \to \mathcal{E}$ is the **embedding**, with $\mathcal{D} \subseteq \mathcal{SA}_I$. When applicable, the timescale of the skill $\overline{\pi}$ is the same as the timescale of the original partial policy $\pi_I$.

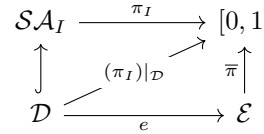

This definition aims at reducing the semantic and sample complexity of $\pi_I$, with the embedding function $e$ extracting "features" of the state-action space that are sufficient for $\pi_I$, and possibly restricting the partial policy domain $\mathcal{SA}_I$ to a smaller subset $\mathcal{D}$; both are usually provided by the teacher as a hint, so that the skill $\overline{\pi}$ is abstract enough to be generalized to other MDPs. The skill $\overline{\pi}$ can be thought of as a higher-order function, taking $e$ as an input; different $e$'s in different problems may be provided as inputs to the same skill $\overline{\pi}$, making a skill a transferable abstraction of policy.

> **Box 2: Skills and embeddings in the MazeBase+ example**
>
> One instance of semantic meanings of skills includes "walking from A to B", which is some basic "navigation" skill $\overline{\pi}^{\text{nav}}$, which appears in the definition of $(g_{2,2}^1)_\beta(\theta)$ in equation E.9 and

as we will see is needed for the student to derive by itself the partial policy generator $(g_{2,2}^1)_\beta(\theta)$ about navigation in the MazeBase+ example.

In another instance, a skill could mean "repeating a policy multiple times", which is a higher-order function $\overline{\pi}^{\text{concat}}$ with $(\overline{\pi}^{\text{concat}}, (e_{\text{decomp}})_{2,2}^2)$ a skill-embedding decomposition of the (partial) policy $\pi_{2,2,*}^2$, where the embedding $(e_{\text{decomp}})_{2,2}^2$ is defined in equation E.26, and the concatenation skill $\overline{\pi}^{\text{concat}}$ in equation E.27. In general, $\overline{\pi}^{\text{concat}}$ concatenates two policies, such as $\pi_{\text{key}}$ followed by $\pi_{\text{pick}}$, and $\pi_{\text{door}}$ followed by $\pi_{\text{open}}$ here. It provides a general logic, independent of the exact location of the agent, key, or door and only depending on the partial information of their relative locations. We do not need the whole grid world to learn opening all the four doors, but only restrict to a much smaller domain $\Omega_{\text{room}_1}$ and learn opening a single door $(\text{door}_1)$ in order to learn the general logic behind opening a door, before applying it to opening more doors.

Reversing Def. 3.1 yields composite partial policies from an embedding and a skill, see App. B.5.

Skills, embeddings, and embedding generators abstract functions in equation B.8 using composition, enabling transfer learning of skills across MDP levels with varying difficulties. A skill's abstraction is proportional to its MMDP level. Higher-level skills, as higher-order functions, take embeddings or embedding generators as inputs, with the product of state space and finer-level policies as domain.

See Box 10 for the composition of partial policy generators in the MazeBase+ example.

# 4 LEARNING MMDPS

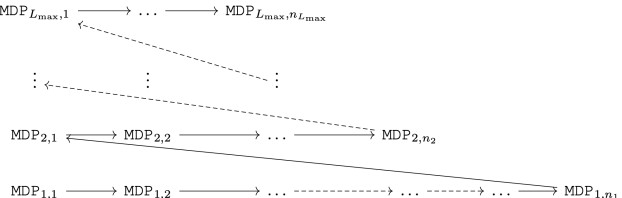

The main framework underlying our algorithm implementation is a **curriculum**, which is an ordered set of MDPs $\{\{\text{MDP}_{L,n}\}_{n=1}^{n_L}\}_{L=1}^{L_{\max}}$ with difficulty $L$ between 1 and $L_{\max}$ and $n_L$ MDPs at difficulty $L$: We have strict lexicographic order "$<$" on the MDPs in a curriculum: for any $L, L' \in [L_{\max}], n \in [n_L], n' \in [n_{L'}], (L, n) < (L', n')$ if and only if one of the following occurs: (1) $L < L'$; (2) $L = L', n < n'$. The student solves the MDPs in this order.

The curriculum for MazeBase+ was described in Box 1 and visualized in Fig. 1.

**Teacher-student-assistant three-way cooperation**. To enable transfer learning and multi-level learning, we introduce three roles: the *teacher*, who provides a curriculum of MDPs (each an MMDP) with information on transfer learning opportunities via shared skills, embeddings, or embedding generators; the *student*, who constructs and solves these MDPs in the given order; and the *assistant*, who aids the student by extracting and recording useful information from previously solved MDPs using *skill-embedding decompositions*. Learning is performed as follows. Having difficulty $L$, each $\text{MDP}_{L,n}$ is an MMDP with $L$ levels; the teacher provides information assisting in its solution to the student, as for an MMDP above, with the following caveats: (i) instead of $\{\Pi_{L,n}^l\}_{l=1}^{L-1}$, the teacher provides the sequence of generator sets $\{\mathcal{G}_{L,n}^l\}_{l=1}^{L-1}$, where each generator in each $\mathcal{G}_{L,n}^l$ is provided either directly or as a skill-embedding generator pair (Def. B.2); (ii) the initial policy for $\text{MDP}_{L,n}^L$ (the highest-level MDP) may be provided either directly or as a skill-embedding pair (Def. B.1). In both cases, the embeddings or embedding generators are directly provided by the teacher, providing an opportunity for transfer; the skill is either directly provided by the teacher or hinted at by the teacher by providing the level and a previous MDP at that level from whose optimal policies the skill is extracted by the assistant, with the latter case yielding a direct transfer of the skill. The student solves $\text{MDP}_{L,n}$ ($L \in [L_{\max}], n \in [n_L]$) using Alg. 1. It first constructs higher-level MDPs bottom-up—the first transfer opportunity—with actions from compositions of skills (provided by the teacher or extracted by the assistant) and embedding generators from the teacher (lines 3–8 of Alg. 6). The student then uses propositions in App. D.1 to compute compressed transition probabilities, rewards, and discount factors for these MDPs (lines 9–10). Next, it solves the MDPs top-down: the highest-level MDP's initial policy (lines 12–16) is either the degenerate *diffusive policy* (uniform over actions) or an informed one from composing a skill (teacher or assistant) with a teacher-provided embedding—the second transfer opportunity—after which finer MDPs are solved easily using higher-level optimal policies as warm-start initializations

(line 17). For each level $l \in [L]$, the student learns an optimal policy $\pi_{L,n,*}^l$ by solving $\text{MDP}_{L,n}^l$, some of which may aid later MDPs (via the two transfer opportunities). If the teacher provides an embedding for $\pi_{L,n,*}^l$, the assistant performs a *skill-embedding decomposition* (Def. 3.1) to extract a skill $skill_{L,n,l}$, added to the public skill set $Skills$, which the student can use anytime (lines 18–22 of Alg. 6). The teacher does not know these skills, but sees the set $Skills$ and can refer to its elements, to provide the students with hints about using a certain skill in an MDP.

---

**Box 3: Transfer learning to a new problem in the MazeBase+ example**

The decomposition of $\text{MDP}_{3,1}$ into multiple levels and extraction of both basic skills and of the higher-order function accelerate learning new difficult problems: we introduce a new problem $\text{MDP}_{3,1}'$, similar to $\text{MDP}_{3,1}$, but with different configuration of the objects (see Box 11).

Figure 2: A representation of the second experiment of the MazeBase+ example, where our framework yields efficient transfer to worlds with different configurations of doors, keys, and the goal. The representation in this figure is similar to the one for the first experiment (Fig. 1).

**Box 4: Algorithmic realization and numerical performance in the MazeBase+ example**

The algorithms for solving MMDPs, detailed in App. E.1.6, applied to the MazeBase+, yield significant speed ups, shown in Fig. 3 (left), displaying $\mathbb{E}_{s_0} V_\pi(s_0)$, the average (over all initial conditions $s_0$) of $V_\pi(s_0)$, where $V_\pi$ is the value function for $\texttt{MDP}_{1,1}$, $\texttt{MDP}_{2,1}$, $\texttt{MDP}_{2,2}$, and $\texttt{MDP}_{3,1}$, during iterations of classical value iteration (in red) and of value iteration within our algorithm, within $\texttt{MDP}_{1,1} = \texttt{MDP}_{\text{dense}}^{\text{nav}}$ in orange, iterations within $\texttt{MDP}_{2,n} (1 \leq n \leq 2)$ at level 2 in blue followed by

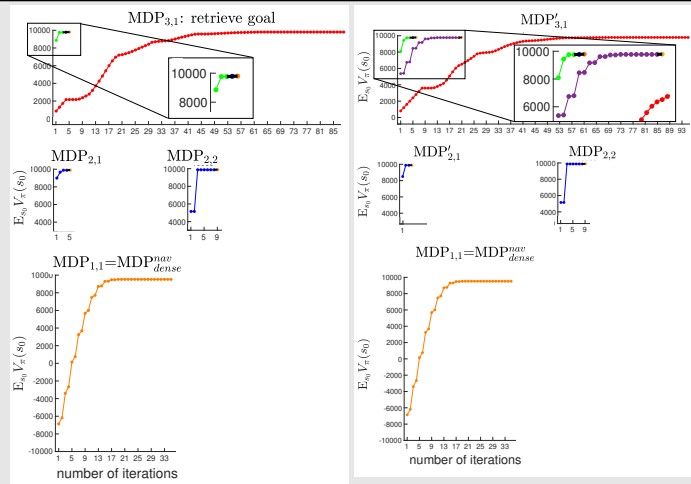

Figure 3: Learning performance for MazeBase+ examples.

iterations at level 1 in orange, and iterations within $\texttt{MDP}_{3,1}$ at level 3 in green followed by iterations at level 2 in blue followed by iterations at level 1 in orange. The extra effort spent in solving the $\texttt{MDP}_{2,n}$'s ($\texttt{MDP}_{1,1}$ is transferred from navigation and transportation with traffic jams), their policies allow us to spend only a few more iterations to solve $\texttt{MDP}_{3,1}$ (green+blue+orange) – far fewer than required by classical value iteration (red). These few iterations correspond to learning how to stitch different steps, well-separated in semantic meaning, including navigating to different objects, or $\texttt{pick}$ or $\texttt{open}$ successfully. For the transfer to new MazeBase+ (Fig. 3, right), we also plot iterations within $\texttt{MDP}_{3,1}'$ at level 2 in purple followed by iterations at level 1 in yellow, assuming the student did not solve $\texttt{MDP}_{2,2}$ and consequently the assistant did not extract $\overline{\pi}^{\text{concat}}$. This corresponds to treating $\texttt{MDP}_{3,1}'$ as an MDP of difficulty 2. The extra effort here (purple+yellow) compared with the original case (green+blue+orange) demonstrates the advantage of treating $\texttt{MDP}_{3,1}'$ as an MDP of difficulty 3 and of extracting the higher-order function $\overline{\pi}^{\text{concat}}$. In all cases the cost of each of our iterations is smaller than that in classical value iteration, as the compressed MDPs have smaller action sets and effective state spaces.

**Navigation and transportation with traffic jams**. This example includes multiple action factors, corresponding to moving and to choosing a means of transportation for the moves in a grid world environment with different traffic conditions. The MazeBase+ uses skills (e.g., an abstract navigation policy) learned from this curriculum. Please see App. C.2, and Fig. 7 in particular, for a comprehensive summary of this example, and see Fig. 8 for the corresponding curriculum.

### 4.1 THEORETICAL ANALYSIS: MMDP SOLVER AND TRANSFER LEARNING

Our MMDP framework can be combined with standard MDP solvers, such as value or policy iteration, and Q-learning is also possible in MMDPs. The multiscale compression both yields savings due to the reduced effective state spaces and policy-search spaces at higher levels, and yet it is robust in situations during the top-bottom phase should high-level policy require significant refinement, yielding a consistent estimate of the optimal policy for the original MDP, for example when combined with value iteration. See App. D.4 for a brief explanation on the consistency of our approach.

## 5 CONCLUSION AND FUTURE WORK

Preliminary work shows very encouraging results when extending this work to learning algorithms such as sorting, and to incorporate recursion. Learning embeddings and skills, instead of having them suggested to the student by the teacher and the assistant, using self-play to discover, at least in part, curricula, and extending to multi-agent systems will be important steps forward.

ChatGPT and Grok were used to compress several paragraphs to satisfy space constraints.

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

## A  RELATED WORK

We expand our discussion of related work in the introduction – of course the related literature is vast, and we restrict ourselves here to mentioning and comparing with some of the techniques we fell are most related to our proposal in this work.

**Hierarchical RL (classic and deep).**  Classic HRL formalizes temporal abstraction via options and SMDPs (Sutton et al., 1999), value decomposition in MAXQ (Dietterich, 2000), and hierarchies of abstract machines (Parr & Russell, 1997); see Barto & Mahadevan (2003) for a survey. Early "feudal" ideas cast hierarchy as managers setting subgoals for workers (Dayan & Hinton, 1993). Deep HRL automates parts of this stack: option-critic learns intra-option policies and termination end-to-end (Bac), FeUdal Networks separate goal setting from control in a learned latent space (Vezhnevets et al., 2017), HIRO corrects off-policy bias for hierarchical goal relabeling (Nachum et al., 2018), and HAC combines hindsight with multi-level goals (Levy et al., 2019). *These systems improve sample efficiency in sparse-reward tasks*, yet many fix hierarchical depth or entangle low-level stochasticity with high-level planning; goal-space specification can be brittle (Dwiel et al., 2019). Our approach repeatedly compresses families of lower-level policies into single abstract actions at higher levels, producing MDPs with less stochasticity that are easier to solve.

**Skill discovery and pretraining.**  Unsupervised methods learn versatile primitives without task rewards, e.g., DIAYN maximizes skill–state mutual information (Eysenbach et al., 2019) and VIC pursues empowerment-like objectives (Gregor et al., 2016). When composed for long horizons, such skills can reintroduce stochasticity and complicate credit assignment; our compression reduces variance at higher levels and aligns the skill library with planning on specific objectives.

**Abstraction, representation, and safe compression.**  Abstraction further motivates multi-level compression. Spectral state representations capture environment geometry and enable multiscale reasoning (Mahadevan & Maggioni, 2007; Machado et al., 2017). Model minimization and homomorphisms formalize when two models or states are "equivalent" for decision making, giving principled conditions for safe compression (Dean & Givan, 1997; Li et al., 2006; Rav). Our method leverages these ideas operationally: each compression step *preserves the semantics* of the original MDP while shrinking variance and branching, so that solving a long-horizon task reduces to solving a stack of smaller, cleaner MDPs with existing algorithms (Puterman, 1994).

**Curriculum learning.**  Curriculum learning provides the second pillar. Classical curricula gradually increase task difficulty to ease optimization (Bengio et al., 2009), with modern variants such as teacher–student curricula (Mat), reverse curricula from goal states (Florensa et al., 2017), and (a-)symmetric self-play that automatically generates tasks (Sukhbaatar et al., 2017; 2018b). However, many curricula define difficulty by *time to solve* rather than *time to learn*, and often restrict the final task to a concatenation of subtasks. In our framework, *difficulty emerges from the compression*: higher-level problems are coarsened in space/time but more global in scope, producing a curriculum that mirrors how humans solve complex tasks—learn sub-skills, compress them into abstract actions, then plan at the higher scale, improving both optimization and transfer on sparse-reward domains.

**Transfer and modularity.**  Our transfer perspective is equally explicit. Successor-feature methods decouple dynamics from rewards and enable generalized policy improvement across tasks with shared structure (Barreto et al., 2017; Bar); policy sketches encourage modular sub-policies that recombine across tasks (Andreas et al., 2017); and progressive nets avoid catastrophic forgetting by lateral connections (Rusu et al., 2016). We factor policies into *embeddings* (problem-specific perception/featurization) and *skills* (reusable higher-order functions), and then compress families of such policies into single abstract actions at higher levels. This creates *transfer across levels and across MDPs*, even when state spaces differ, without resorting to rote replay of previously seen trajectories. As a cautionary contrast, Go-Explore achieves remarkable exploration by explicitly remembering cells and returning before exploring (Ecoffet et al., 2021); while highly effective, this mechanism can resemble *memorization of the state space* and requires additional robustification for stochasticity, whereas our abstraction focuses on reusing *semantics* rather than stored states.

**Meta-reinforcement learning.** Because multi-level structure is a prior over *families* of problems, our work connects to meta-RL: $RL^2$ trains recurrent agents to adapt quickly (Duan et al., 2016), MAML learns initializations for fast policy updates (Finn et al., 2017), PEARL infers latent task variables for rapid adaptation (Rakelly et al., 2019), and MLSH co-trains shared sub-policies with task-specific controllers (Frans et al., 2018). Our compression-and-skill factorization gives a constructive route to meta-generalization: higher levels expose slower, more stable dynamics, while lower levels encapsulate fast feedback, allowing efficient cross-task reuse with fewer iterations and cheaper per-iteration computation.

**Inverse reinforcement learning and imitation.** Finally, our framework is compatible with inverse reinforcement learning (IRL) and imitation. Foundational methods (IRL by Ng & Russell 2000; apprenticeship learning by Abbeel & Ng 2004; Maximum-Entropy IRL by Ziebart et al. 2008) recover rewards from demonstrations; modern deep variants learn cost/reward implicitly (GCL (Finn et al., 2016)), through adversarial imitation (GAIL (Ho & Ermon, 2016)) or adversarial IRL (AIRL (Fu et al., 2018)). Hierarchical IRL further segments demonstrations to recover sub-task rewards (Krishnan et al., 2016). Because each compression step yields an independent, semantically preserved MDP, we can invert the process: estimate rewards or subgoal structures at appropriate levels, then learn skills and curricula consistent with demonstrations, improving sample-efficiency and interpretability relative to flat IRL (Adams et al., 2022).

**Extended discussion and comparison of our work with Sukhbaatar et al. (2018a)** The MazeBase+ example we consider is a more complicated version of MazeBase appearing in many literature, such as in Sukhbaatar et al. (2018a), which is of difficulty 2 in our regime, since we add multiple rooms, with different geometries and arrangements/order of doors and keys, that require an additional level of abstraction/complexity in order to achieve efficient learning.

This MazeBase+ example shows sufficiently how planning like humans is realized by our framework. When trying to solve a difficult problem of picking up a goal, we first consider it at high level: we need to open door A, open door B, and then we pick up the goal. Then, we focus on each subtask and consider it more carefully: to open a door, we need to travel to a key, pick it up, travel to the door, and then open it; to pick up the goal, we need to travel to the goal, and then pick it up. These two logics are similar, so we extract out a single higher-order function $\overline{\pi}^{\text{concat}}$. Then, we also notice that we need to go to another location in $\Omega$ for multiple times while avoiding blocks assuming all the doors are open, so we extract out a single navigation skill $\overline{\pi}^{\text{nav}}$. Finally, we go down to the finest level: to travel to a certain destination while avoiding the blocks, we should first go to door A, then to doors B,C, ... before going directly to that destination. All these processes are similarly about navigation within a single room while avoiding blocks, so we extract a single basic navigation skill $\overline{\pi}^{\text{nav}}_{\text{dense}}$, which we may also transfer from perviously solved problems in a different curriculum, such as here in the example of navigation and transportation with traffic jams.

Another main advantage of our framework is that thanks to the introduction of partial policy generators, embedding generators and skills, we allow for very flexible transfer learning opportunities, not restricted to only concatenation as in Sukhbaatar et al. (2018a). In particular, our embeddings take the current state, action coordinates and goal state as inputs, rather than only taking the goal state as input while keeping fixed the high-dimensional current state as in Sukhbaatar et al. (2018a), giving us the opportunity to reduce the dimension and generate abstractions to a greater extent, as well as utilize the policy structure better. In this way, we encapsulate the knowledge in the policies we have already learned and focus on learning the unknown parts of the policies in new MDPs. Therefore, we avoid learning repeatedly similar subtasks for multiple times, such as traveling to certain destinations while avoiding blocks assuming all the doors are open, which greatly reduces computational costs. Last but not least, the framework in Sukhbaatar et al. (2018a) is unsupervised, whereas our framework is constructed for solving the target MDPs, such as $\text{MDP}_{3,1}$ here, so it is supervised.

## B    DETAILED DEFINITIONS

In this section, we provide more detailed definitions of the important mathematical concepts, starting from MDPs, to generating set of policies essential for constructing and solving MMDPs, and concluding in building connections between different MDPs through transfer opportunities.

## B.1 MDPs

- **State space** $\mathcal{S}$ consisting of all the possible states of the agent; it includes the set of **initial states**, $\mathcal{S}^{\text{init}} \subseteq \mathcal{S}$, and of **terminal states**, $\mathcal{S}^{\text{end}} \subseteq \mathcal{S}$, at which an episode starts and ends, respectively. MDP is episodic if $\mathcal{S}^{\text{end}} \neq \varnothing$. Without losing generality, we let $\mathcal{S}^{\text{init}} \subseteq \mathcal{S} \setminus \mathcal{S}^{\text{end}}$ to be the set of states starting from which the agent can reach some state in $\mathcal{S}^{\text{end}}$ if $\mathcal{S}^{\text{end}} \neq \varnothing$, and set $\mathcal{S}^{\text{init}} = \mathcal{S}$ otherwise.

- **Action set** $\mathcal{A} := \cup_{s \in \mathcal{S}} \mathcal{A}(s)$ with $\mathcal{A}(s)$ the set of actions available in $s$, for $s \in \mathcal{S}$. We let $\mathcal{SA} := \{(s,a) : s \in \mathcal{S}, a \in \mathcal{A}(s)\}$. We assume that $\mathcal{A} \subseteq \mathcal{A}_1 \times \cdots \times \mathcal{A}_K$, where $\mathcal{A}_k$ is called the $k$-th action factor, $k \in [K] := \{1, \dots, K\}$. We assume each $\mathcal{A}_k$ contains a special element $a^{\text{end}}$, *universal* across all different MDPs, used to end a policy. We let $\mathcal{A}^{\text{end}}$ be the set of actions in which at least one factor is $a^{\text{end}}$, and assume $\mathcal{A}^{\text{end}} \subseteq \mathcal{A}(s)$ for any $s \in \mathcal{S}$. We let $a^{\text{end}} \in \mathcal{A}$ be the action whose coordinates are $a^{\text{end}}$ in all action factors. For any $\mathcal{A}' \subseteq \mathcal{A}$, we denote $\overline{\mathcal{A}'} := \mathcal{A}' \cup \mathcal{A}^{\text{end}}$. We can restrict actions to a subset of factors: for $I \subseteq [K]$, called an active action factor set, we map $a \in \mathcal{A}$ to $a_I$ by setting to the special value 0, an "action" that cannot be taken, the coordinates of $a$ corresponding to the action factors $\mathcal{A}_k$ with $k \notin I$, and let $\mathcal{A}_I$ be the image of $\mathcal{A}$ under such map.

- **Transition probabilities** $P : \mathcal{SA} \times \mathcal{S} \to [0,1]$, where $P(s,a,s')$ is the probability of reaching state $s'$ for an agent in state $s$ selecting action $a$. We assume $P(s,a,s') = \mathbb{1}_{\{s\}}(s')$ for any $a \in \mathcal{A}^{\text{end}}$.

- **Rewards** $R : \mathcal{SAS} \to \mathbb{R}$, with $R(s,a,s')$ the reward for an agent in state $s$, selecting action $a$, and reaching state $s'$. We let $R(s,a,s') = 0$ for $s \in \mathcal{S}^{\text{end}}$, $R(s,a,s') = r < 0$ for $s \notin \mathcal{S}^{\text{end}}, a \in \mathcal{A}^{\text{end}}$.

- **Discount factors** $\Gamma : \mathcal{SAS} \to (0,1]$, with $\Gamma(s,a,s')$ the discount applied to reward $R(s,a,s')$. We always define $\Gamma(s,a,s') = 1$ for any $a \in \mathcal{A}^{\text{end}}$.

We assume throughout that $\mathcal{S}, \mathcal{A}$ are finite sets (the generalization to continuous state spaces with the use of basis functions is rather straightforward), and that the rewards $R$ are bounded.

For any two (random) times $T, T'$, $0 \leq T < T' < \infty (a.s.)$, the random variable $R_{T,T'}$ is the discounted reward accumulated over the interval $[T, T']$:

$$R_{T,T'} := R(S_T, A_{T+1}, S_{T+1}) + \sum_{t=T+1}^{T'-1} [\Pi_{t'=T}^{t-1} \Gamma(S_{t'}, A_{t'+1}, S_{t'+1})] \times R(S_t, A_{t+1}, S_{t+1}), \quad \text{(B.1)}$$

and $\tau = +\infty$ for non-episodic MDPs, and otherwise $\tau$ being the first time $t$ such that $S_t \in \mathcal{S}^{\text{end}}$ ($+\infty$ if not existing). The optimal policy $\pi_*$ is independent of the initial state $S_0$, by Markovianity.

## B.2 ON GENERATION OF THE SET OF POLICIES FROM THE PARTIAL POLICY GENERATOR SET

Once we have the partial policy generator set $\mathcal{G}$, we can use it to generate the set $\Pi_{\mathcal{G}}$ of policies, wherein a policy $\pi$, before restriction and normalization, can be written as $(g_{m_1})_{I_{m_1}}(\theta_{m_1}) \otimes \cdots \otimes (g_{m_J})_{I_{m_J}}(\theta_{m_J})$ for some $J \in [M], \{m_j :\in [J]\} \subseteq [M], \theta_{m_j} \in \Theta_{m_j}$ for any $j \in [J]$, and $I_{m_{j_1}} \cap I_{m_{j_2}} = \emptyset$ for $j_1 \neq j_2$; it therefore may be represented by a vector with entries $\theta_{m_j}$ for $j \in [J]$ and $\texttt{null}$ for all the other entries, with indices $[M] - \{m_j : j \in [J]\}$. Note that $\texttt{null}$ is a special value *universal* across all different MDPs, meaning that the corresponding partial policy generator in $\mathcal{G}$ is not selected. All the policies in $\Pi_{\mathcal{G}}$ can then be represented by an element in the product set $(\Theta_1 \cup \{\texttt{null}\}) \times (\Theta_2 \cup \{\texttt{null}\}) \times \cdots \times (\Theta_M \cup \{\texttt{null}\})$.

## B.3 CONSTRUCTING MMDPs

Now, since we have generated the set of policies, we obtain the action set at higher levels of MMDPs, and we now provide equations for various quantities needed in the definition of MMDPs, in sec. 2.1.

Note that $\mathcal{A}^{l+1}(s)$ in the definition of an MMDP does not depend on $s$; this is not very restrictive as a policy in $\overline{\Pi}^l$ may choose the action $a^{\text{end}}$ with probability 1 at certain states. This will be needed in some of the examples.

$$P^{l+1}(s, a^{l+1}, s') := \mathbb{P}^l_{\tau, S_{1:\tau}, A_{0:\tau-1}}[S_\tau = s' | S_0 = s, A_{0:\tau-1} \sim a^{l+1}, \tau \sim \text{Geo}(1/t_{a^{l+1}})]. \quad \text{(B.2)}$$

$$R^{l+1}(s, a^{l+1}, s') := \mathbb{E}^l_{\tau, S_{1:\tau}, A_{0:\tau-1}}[R^l_{0,\tau}|S_0 = s, S_\tau = s', A_{0:\tau-1} \sim a^{l+1}]. \quad \text{(B.3)}$$

$$R^l_{T,T'} := R^l(S_T, A_{T+1}, S_{T+1}) + \sum_{t=T+1}^{T'-1} [\Pi_{t'=T}^{t-1}\Gamma^l(S_{t'}, A_{t'+1}, S_{t'+1})] \times R^l(S_t, A_{t+1}, S_{t+1}). \quad \text{(B.4)}$$

To clarify, the condition in equation B.3 may happen with probability 0, in which case $R^{l+1}$ is not well-defined at such input triples $(s, a^{l+1}, s')$; this is immaterial since such values are not needed when solving $\text{MDP}^{l+1}$. The same applies to the discount factor $\Gamma^{l+1}$.

$$\Gamma^{l+1}(s, a^{l+1}, s') := \mathbb{E}^l_{\tau, S_{1:\tau}, A_{0:\tau-1}}[\Gamma^l_{0,\tau}|S_0 = s, S_\tau = s', A_{0:\tau-1} \sim a^{l+1}]. \quad \text{(B.5)}$$

For any two (random) times $T, T'$ satisfying $0 \le T < T' < \infty(a.s.)$, the random variable $\Gamma^l_{T,T'}$ is the cumulative discount applied to trajectories $(S_T, S_{T+1}, \cdots, S_{T'})$ over the interval $[T, T']$ in $\text{MDP}^l$:

$$\Gamma^l_{T,T'} := \Pi_{t=T}^{T'-1}\Gamma^l(S_t, A_{t+1}, S_{t+1}). \quad \text{(B.6)}$$

## B.4 SOLVING MMDPs

Once we complete the definition of MMDPs, the next step is to solve them: at this point the essential ingredient is the ability of moving a policy from a coarse scale to a policy at a finer scale, which we think as a sort of "convolution". The "convolution" $\pi * \mathcal{G}$ between a generator set $\mathcal{G} = \{g|_I : \Theta \to \{\pi|_I : \sum_{a \in \mathcal{A}_I(s)} \pi|_I(s, a) = 1 \text{ for any } s \in \mathcal{S}\}\}$ and a function $\pi$ mapping from $\mathcal{S} \times \Pi_{\mathcal{G}}$, is a policy on $\mathcal{S}\mathcal{A}$ defined by:

$$(\pi * \mathcal{G})(s, a) := \sum_{g_1|_{I_1}, \cdots, g_M|_{I_M}} \sum_{\theta_1, \cdots, \theta_M} \pi\big(s, \otimes_{m=1}^M g_m|_{I_m}(\theta_m)\big) \times \big[\otimes_{m=1}^M g_m|_{I_m}(\theta_m)\big](s, a),$$

$$\text{(B.7)}$$

where the sum is taken over all $g_1|_{I_1}, g_2|_{I_2}, \cdots, g_M|_{I_M} \in \mathcal{G}$ mapping from $\Theta_1, \Theta_2, \cdots, \Theta_M$ respectively, such that their active action factor sets $I_1, I_2, \cdots, I_M$ are a partition of $[K]$, where $K$ is the number of action factors in $\mathcal{A}$, and the second sum is taken over all parameters in their domains [2]. Iterating over levels (from top to bottom), the optimal policy $\pi^L_*$ of $\text{MDP}^L$ is convolved with the sequence of generator sets $\{\mathcal{G}^l\}_{l=1}^{L-1}$ to a policy $\tilde{\pi}_*$ for the original MDP, and so on:

$$\tilde{\pi}_* := [\cdots [(\pi^L_* * \mathcal{G}^{L-1}) * \mathcal{G}^{L-2}] * \cdots] * \mathcal{G}^1. \quad \text{(B.8)}$$

## B.5 TRANSFER

Before we are ready to achieve transfer between different MDPs, we need to further decompose or compose the terms appearing in equation B.8, where we need the following additional definitions, besides the ones in Sec. 3:

**Definition B.1.** Given $\mathcal{S}\mathcal{A}_I$, $\mathcal{D} \subseteq \mathcal{S}\mathcal{A}_I$, an embedding $e : \mathcal{D} \to \mathcal{E}$, and a skill $\overline{\pi} : \mathcal{E} \to [0, 1]$, we define their **composite partial policy** $\pi_I : \mathcal{S}\mathcal{A}_I \to [0, 1]$, denoted (with abuse of notation) by $\overline{\pi} \circ e$:

$$\widehat{\pi}_I(s, a) := \overline{\pi}(e(s, a))\mathbb{1}_{\mathcal{D}}((s, a)) \quad , \quad \pi_I(s, a) := \begin{cases} \frac{\widehat{\pi}_I(s,a)}{\sum_{a \in \mathcal{A}_I(s)} \widehat{\pi}_I(s,a)} & , \sum_{a \in \mathcal{A}_I(s)} \widehat{\pi}_I(s, a) \neq 0 \\ \mathbb{1}_{\{a^{\mathbf{end}}\}}(a) & , \text{otherwise} \end{cases},$$

and the timescale of the composite partial policy $\pi_I$ is the same as the timescale of the skill $\overline{\pi}$. The intuition behind the normalization is that the action $a^{\mathbf{end}}$ provides the option to terminate when the agent reaches states new to it.

The generalization of this definition to that of a composite partial policy generator is straightforward:

**Definition B.2.** Given a partial policy domain $\mathcal{S}\mathcal{A}_I$, an embedding generator $E : \Theta \to \{e : \mathcal{D}_e \subseteq \mathcal{S}\mathcal{A}_I \to e(\mathcal{D}_e)\}$, and a skill $\overline{g} : \mathcal{E} \to \overline{g}(\mathcal{E})$ satisfying $\cup_{e \in E(\Theta)} e(\mathcal{D}_e) \subseteq \mathcal{E}$, we define their **composite**

---

[2] here, we assume the domains of all the partial policy generators in $\mathcal{G}$ are discrete domains; the generalization to the case when some of the domains are continuous is straightforward.

**partial policy generator** $g|_I : \Theta \to \{\pi_I : \sum_{a \in \mathcal{A}_I(s)} \pi_I(s,a) = 1 \text{ for any } s \in \mathcal{S}\}$ by function composition and normalization starting from $E, \overline{g}$:

$$\text{composition}: \quad \widehat{g}_I(\theta)(s,a) := \overline{g}(E(\theta)(s,a))\mathbb{1}_{\mathcal{D}_{E(\theta)}}(s,a)$$

$$\text{normalization}: \quad g_I(\theta)(s,a) := \begin{cases} \frac{\widehat{g}_I(\theta)(s,a)}{\sum_{a \in \mathcal{A}_I(s)} \widehat{g}_I(\theta)(s,a)} & , \sum_{a \in \mathcal{A}_I(s)} \widehat{g}_I(\theta)(s,a) \neq 0 \\ \mathbb{1}_{\{a^{\mathbf{end}}\}}(a) & , \text{otherwise} \end{cases}.$$

## C  INSTANTIATION OF MATHEMATICAL CONCEPTS IN OUR EXAMPLES

Now that we have provided more detailed definitions of the important mathematical concepts in App. B, we instantiate them within the setup of each of the two examples considered in the main text: MazeBase+ (and its variations) in the next subsection, and Navigation and Transportation with Traffic Jams in the one after that..

### C.1  MAZEBASE+

---

**Box 5: Action factors and partial policy generators in the MazeBase+ example**

App. E.1.4 details the equation(s) needed for this box. In this example there is a single action factor, and to simplify the notation, we will sometimes omit the "$|.$" notations in partial policies or partial policy generators when there is only a single action factor. In the forthcoming example of navigation and transportation with traffic jams, there are two action factors $\mathcal{A}_{\mathtt{dir}} \cup \{a^{\mathtt{end}}\}$ and $\mathcal{A}_{\mathtt{means}} \cup \{a^{\mathtt{end}}\}$, with active action factor sets $I_1 := \mathtt{dir}$ and $I_2 := \mathtt{means}$, respectively. Because of this, all the partial policy generators in the MazeBase+ example are policy generators; one policy generator is $(g^1_{2,2})_\alpha$ used for $\mathtt{MDP}_{2,2}$. It is defined on $(\Theta^1_{2,2})_\alpha := \{\mathtt{pick}, \mathtt{open}\}$, and $(g^1_{2,2})_\alpha : (\Theta^1_{2,2})_\alpha \to \{(\pi^1_{2,2})^\alpha_\theta : \theta \in (\Theta^1_{2,2})_\alpha\}$, mapping $\theta$ to $\{(\pi^1_{2,2})^\alpha_\theta : \theta \in (\Theta^1_{2,2})_\alpha\}$, with the partial policy $(\pi^1_{2,2})^\alpha_\theta : \mathcal{S}\mathcal{A}^1_{2,2} \to \{0,1\}$

$$(\pi^1_{2,2})^\alpha_\theta((s_{\mathrm{cur}}, s_{\mathtt{pick}}, s_{\mathtt{open}}), a) := \mathbb{1}_{\{\theta\}}(a), \tag{C.1}$$

which selects the action $\theta \in \{\mathtt{pick}, \mathtt{open}\}$. The role of the lower and upper indices in the names of the policy generators will become apparent later, as indicator of which MDP the generator is used on, and, respectively, of the "level" of the MDP within an MMDP.

The other partial policy generator $(g^1_{2,2})_\beta$ used for $\mathtt{MDP}_{2,2}$ is more interesting. $(g^1_{2,2})_\beta : (\Theta^1_{2,2})_\beta \to \{(\pi^1_{2,2})^\beta_\theta : \theta \in (\Theta^1_{2,2})_\beta\}$ is given by $(g^1_{2,2})_\beta(\theta) = (\pi^1_{2,2})^\beta_\theta$, and here $(\pi^1_{2,2})^\beta_\theta : \mathcal{S}\mathcal{A}^1_{2,2} \to [0,1]$ $(\theta \in (\Theta^1_{2,2})_\beta)$ are defined as in equation E.9.

---

**Box 6: Partial policy generator set in the MazeBase+ example**

Take $\mathtt{MDP}_{2,2}$ for instance. For the first level, $\mathtt{MDP}^1_{2,2} := (\mathcal{S}^1_{2,2}, (\mathcal{S}^{\mathrm{init}}_{2,2})^1, (\mathcal{S}^{\mathrm{end}}_{2,2})^1, \mathcal{A}^1_{2,2}, P^1_{2,2}, R^1_{2,2}, \Gamma^1_{2,2}) = (\mathcal{S}_{2,2}, \mathcal{S}^{\mathrm{init}}_{2,2}, \mathcal{S}^{\mathrm{end}}_{2,2}, \mathcal{A}_{2,2}, P_{2,2}, R_{2,2}, \Gamma_{2,2})$. The teacher provides

$$\mathcal{G}^1_{2,2} := \{(g^1_{2,2})_\alpha, (g^1_{2,2})_\beta\},$$

whose elements were defined in the previous Box 6. From the set of partial policies

$$\{(\pi^1_{2,2})^\alpha_\theta : \theta \in (\Theta^1_{2,2})_\alpha\} \cup \{(\pi^1_{2,2})^\beta_\theta : \theta \in (\Theta^1_{2,2})_\beta\},$$

the student generates the set of policies $\Pi^1_{2,2}$, which in this case is exactly the same as the set itself as there is only a single action factor in this example. More interesting instantiations utilizing equation 2.1 can be seen in $\mathtt{MDP}_{3,1}$. Note that each policy in $\Pi^1_{2,2}$ is represented by an element in the product set $((\Theta^1_{2,2})_\alpha \cup \{a^{\mathrm{end}}, \mathtt{null}\}) \times ((\Theta^1_{2,2})_\beta \cup \{a^{\mathrm{end}}, \mathtt{null}\})$. For instance, $(\pi^1_{2,2})^\alpha_{\mathtt{pick}}$ is represented by $(\mathtt{pick}, \mathtt{null})$. So, $\Pi^1_{2,2}$ has two action factors.

---

**Box 7: Inputs for the construction of MMDPs in the MazeBase+ example**

App. E.1.4 details the equation(s) needed for this box. The provided **sequence of generator sets** $\{\mathcal{G}_{2,2}^l\}_{l=1}^{\infty}$ for $\text{MDP}_{2,2}$ in this example consists of $\mathcal{G}_{2,2}^1$ as in Box 6, and $\mathcal{G}_{2,2}^l := \varnothing$ for $l \geq 2$. There are multiple options for $\{\mathcal{G}_{2,2,\text{test}}^l\}_{l=1}^{\infty}$, with the restrictions: $\pi_{2,2,*} \notin \mathcal{G}_{2,2,\text{test}}^1$, $\pi_{2,2,*}^2 \in \mathcal{G}_{2,2,\text{test}}^2$, with $\pi_{2,2,*} = \pi_{2,2}$, $\pi_{2,2,*}^2$ being defined in equation E.13 and equation E.12 respectively. These conditions will guarantee that $\text{MDP}_{2,2}$ is of difficulty 2. The timescale of $(g_{2,2}^1)_\alpha$ is 1, and the timescale of $(g_{2,2}^1)_\beta$ is $+\infty$. We let $r^1 = -10$. The student then constructs the second-level $\text{MDP}_{2,2}^2 := (\mathcal{S}_{2,2}, \mathcal{S}_{2,2}^{\text{init}}, \mathcal{S}_{2,2}^{\text{end}}, \overline{\Pi_{2,2}^1}, P_{2,2}^2, R_{2,2}^2, \Gamma_{2,2}^2)$, using the inputs above and the procedures we describe momentarily.

**Box 8: Unpacking compressed policies in the MazeBase+ example**

App. E.1.4 details the equation(s) needed for this box. Once the second-level MDP $\text{MDP}_{2,2}^2 := (\mathcal{S}_{2,2}, \mathcal{S}_{2,2}^{\text{init}}, \mathcal{S}_{2,2}^{\text{end}}, \overline{\Pi_{2,2}^1}, P_{2,2}^2, R_{2,2}^2, \Gamma_{2,2}^2)$ is constructed using the procedures above, it can be solved to find the optimal policy $\pi_{2,2,*}^2$. We construct stochastic trajectories starting from each state $s \in \mathcal{S}$ by "gluing" together the actions in $\{(\pi_{2,2}^1)_\theta^\alpha : \theta \in (\Theta_{2,2}^1)_\alpha\} \cup \{(\pi_{2,2}^1)_\theta^\beta : \theta \in (\Theta_{2,2}^1)_\beta\}$ (defined in Box 5) following an order of the form $(\pi_{2,2}^1)_{\theta_0}^{\lambda_0}, \cdots, (\pi_{2,2}^1)_{\theta_t}^{\lambda_t}, \cdots$: the agent starts from $S_0 = s$ in $\text{MDP}_{2,2}^2$, and chooses actions in $A_t = (\pi_{2,2}^1)_{\theta_t}^{\lambda_t}$ for any $0 \leq t \leq \tau - 1$ with $\tau$ being the first time $t$ such that $S_t \in \mathcal{S}^{\text{end}}$ (if such event does not occur, we set $\tau = +\infty$). For each state $s$, the sequence of actions needs to be optimized to maximize the expected cumulative rewards along the stochastic trajectories. Following the description here, the value $\pi_{2,2,*}^2(s,a)$, for each $a = (\pi_{2,2}^1)_{\theta_0}^{\lambda_0} \in \{(\pi_{2,2}^1)_\theta^\alpha : \theta \in (\Theta_{2,2}^1)_\alpha\} \cup \{(\pi_{2,2}^1)_\theta^\beta : \theta \in (\Theta_{2,2}^1)_\beta\}$, $s = (s_{\text{cur}}, s_{\text{pick}}, s_{\text{open}})$, equals equation E.10, where recall that for any two (random) times $0 \leq T < T' < \infty$ (a.s.), $(R_{2,2}^2)_{T,T'}$ is defined as in equation B.1 for $\text{MDP}_{2,2}^2$. Note that the minimization of $\theta_t'$ is trivial at any state $s$. Solving this optimization problem, the student derives equation E.12, representing the concatenation logic that the agent will go to the key, pick it up, go the door, and then open the door. The agent could focus on learning this higher-order function, which is the new knowledge to be learned from $\text{MDP}_{2,2}$, because the details of going from A to B have been encapsulated by the navigation skill $\overline{\pi}^{\text{nav}}$.

$\text{MDP}_{2,2}^2$ has at least two key advantages compared to the level-1 MDP: first, it has much shorter time horizon, as a single action moves the agent by multiple steps, until achieving a small subgoal including going to $\text{key}$, $\mathcal{N}(s_{\text{door}})$, picking up $\text{key}$ or opening $\text{door}$; second, the stochasticity is greatly reduced as it is absorbed into each higher-level navigation policy, further simplifying the optimization in equation E.11. Note that it is crucial here that the navigation policies used at this level terminate, by choosing $a^{\text{end}}$ at very precise times and locations, instead of relying on random stopping times.

Finally, the student solves the original $\text{MDP}_{2,2}$, by using equation B.7 to pass the optimal policy of $\text{MDP}_{2,2}^2$ down to level one, resulting in the policy equation E.13. In this particular example, $\pi_{2,2}$ is in fact the optimal policy $\pi_{2,2,*}$ of the original $\text{MDP}_{2,2}$, requiring no additional refinement by value iteration.

**Box 9: Multi-level compression in the MazeBase+ example**

App. E.1.3 details the equation(s) needed for this box. For this example, we actually can provide the explicit formulas for the compressed MDPs to help the reader see what the effects of the compression are in abstracting out the part of knowledge provided and let the agent focus on learning higher-level policies. To simplify these expressions, assume $\overline{\pi}_{\text{dense}}^{\text{nav}}$ extracted from $\text{MDP}_{1,1}$ is deterministic, and hence $\overline{\pi}^{\text{nav}}$ extracted from $\text{MDP}_{2,1}$ is deterministic. We first take $\text{MDP}_{2,2}$ for instance: the student constructs the second level MDP $\text{MDP}_{2,2}^2 = (\mathcal{S}_{2,2}, \mathcal{S}_{2,2}^{\text{init}}, \mathcal{S}_{2,2}^{\text{end}}, \overline{\Pi_{2,2}^1}, P_{2,2}^2, R_{2,2}^2, \Gamma_{2,2}^2)$, with $P_{2,2}^2, R_{2,2}^2, \Gamma_{2,2}^2$ as defined in equation E.4.

**Box 10: Composing partial policy generators in the MazeBase+ example**

App. E.1.5 details the equation(s) needed for this box. In the MazeBase+ example, we now explain how the partial policy generators $(g_{2,2}^1)_\alpha$ and $(g_{2,2}^1)_\beta$ in $\mathcal{G}_{2,2}^1$ can be constructed by composing skills with two embedding generators $E_\alpha^1$ and $E_\beta^1$.

$(g_{2,2}^1)_\alpha$ is the composition of the degenerate skill, which is the identity map on $[0,1]$, and $(E_{2,2}^1)_\alpha : (\Theta_{2,2}^1)_\alpha \to \{(e_{2,2}^1)_\theta^\alpha : \theta \in (\Theta_{2,2}^1)_\alpha\}$ is defined as $(E_{2,2}^1)_\alpha(\theta) := (e_{2,2}^1)_\theta^\alpha$, with $(e_{2,2}^1)_\theta^\alpha : \mathcal{SA}_{2,2}^1 \to \{0,1\}$ being defined as $(e_{2,2}^1)_\theta^\alpha((s_{\text{cur}}, s_{\text{pick}}, s_{\text{open}}), a) := \mathbb{1}_{\{\theta\}}(a)$, which is the $(\pi_{2,2}^1)_\theta^\alpha$ we introduced in equation C.1.

$(g_{2,2}^1)_\beta$ is the composition of $\overline{\pi}^{\text{nav}}$ and $(E_{2,2}^1)_\beta$, with $(E_{2,2}^1)_\beta : (\Theta_{2,2}^1)_\beta \to \{(e_{2,2}^1)_\theta^\beta : \theta \in (\Theta_{2,2}^1)_\beta\}$ with $(\Theta_{2,2}^1)_\beta := \{\texttt{key}, \texttt{door}\}$ being defined as in equation E.25. In particular, we ruled out the final step of reaching $s_{\text{door}}$ from the domain of the embedding $(e_{2,2}^1)_{\text{door}}^\beta$ generated from $(E_{2,2}^1)_\beta$, and replaced it by $a^{\text{end}}$, so that the policy generator $(g_{2,2}^1)_\beta$, composed from $\overline{\pi}^{\text{nav}}$ and $(E_{2,2}^1)_\beta$, generated the policy $(\pi_{2,2}^1)_{\text{door}}^\beta$ stopping at $\mathcal{N}(s_{\text{door}})$ instead of stopping at $s_{\text{door}}$, because the door may be closed initially.

**Box 11: Transfer learning to a new problem in the MazeBase+ example: more information**

App. E.1.4 details the equation(s) needed for this box. First of all, $\texttt{door}_2$ is now connecting between $\texttt{room}_2$ and $\texttt{room}_4$ rather than $\texttt{room}_1$ and $\texttt{room}_3$ (i.e., the constraints for $s_{\text{door}_2}$ change to $s_{\text{door}_2}(1) > s_{\text{door}_1}(1), s_{\text{door}_1}(2) < s_{\text{door}_2}(2) < s_{\text{door}_3}(2)$), and secondly, initially, $s_{\text{key}_2} \in \Omega_{\text{room}_2}, s_{\text{key}_3} \in \Omega_{\text{room}_4}, s_{\text{goal}} \in \Omega_{\text{room}_3}, \mathcal{S}_{3,2}^{\text{init}} = \{(s_{\text{cur}}, s_{\textbf{pick}}, s_{\text{open}}, s_{\text{done}}) \in \mathcal{S}_{3,2} : s_{\text{done}} = 0\}$. The other objects and notations are defined as above, taking into consideration the updated configuration just described.

In $\text{MDP}_{3,1}'$, $\texttt{door}_2$ is indeed useful in order to reach $\texttt{goal}$, and even though $\texttt{goal}, \texttt{door}_2$, $\texttt{key}_2$, and $\texttt{key}_3$ now have significantly different positions, and all the other objects could have other fixed locations within the given constraints, in order to solve $\text{MDP}_{3,1}'$ we only need a few iterations of value iteration inside our algorithm, with no need to re-solve $\text{MDP}_{\text{dense}}^{\text{nav}}$ and $\text{MDP}_{2,2}$. The reason is that we do not change the overall geometry, so the skill $\overline{\pi}_{\text{dense}}^{\text{nav}}$ about navigation within a single room while avoiding blocks still applies; the logic of how to open a door is ubiquitous and universally applies everywhere; the only parts that need to be relearned include: $(i)$ navigation through $\Omega$ while avoiding blocks assuming all the doors are open (since the doors have significantly changed positions); $(ii)$ the policy of how to navigate between rooms when some doors may be closed at level 3 (since doors and keys have significantly changed positions). For $(i)$, the teacher just needs to construct a new MDP $\text{MDP}_{2,1}'$, similar to $\text{MDP}_{2,1}$; the student solves it and the assistant extracts a new skill $(\overline{\pi}^{\text{nav}})'$ about navigation through the whole grid world $\Omega$ while avoiding blocks assuming all the doors are open, by stitching together several policies coming from $\overline{\pi}_{\text{dense}}^{\text{nav}}$. Only a few iterations of value iteration will suffice to achieve this, see Fig. 4. For $(ii)$, after the student constructs $(\text{MDP}_{3,1}^3)'$, the student needs to solve it as if it is a completely new problem, but of tiny size. The optimal policy $(\pi_{3,1,*}^3)'$ for $(\text{MDP}_{3,1}^3)'$ has length 4, longer than the previous $\pi_{3,1,*}^3$ for $\text{MDP}_{3,1}^3$, and can be discovered in very few iterations of value iteration. $(\pi_{3,1,*}^3)'$ is given by equation E.21.

So, even if $(\overline{\pi}^{\text{nav}})'$ and $(\pi_{3,1,*}^3)'$ need to be relearned, we still achieve a significant reduction in the effort to learn and in computations thanks to transfer learning and compression, achieving "few-shot learning", because $(i)$ $\overline{\pi}_{\text{dense}}^{\text{nav}}$ is still usable when building $(\text{MDP}_{2,1}^2)'$, level 2 of $\text{MDP}_{2,1}'$, and thus essentially the process of traveling between any two locations within a single room is encapsulated in a single action. Given that in "grid world type" examples, the number of iterations in value iteration is proportional to the average number of actions needed before reaching the goal state, working at level 2 of $\text{MDP}_{2,1}'$ where each step/action is at level of a whole room, only a few iterations (proportional to the number of rooms the student needs to walk through before reaching $s_{\text{dest}}$) are needed for solving $\text{MDP}_{2,1}'$. $(ii)$ Similarly, at level 3 for $\text{MDP}_{3,1}'$, all the actions within a single room are encapsulated in a single action, so we only need a few iterations

(proportional to the number of rooms the student needs to walk through before picking up goal) for solving $\text{MDP}'_{3,1}$.

Without transfer and without our framework, solving $\text{MDP}'_{3,1}$ directly would be significantly more expensive, requiring a number of iterations proportional to the average diameter of the rooms the student needs to walk through before picking up goal multiplied by the number of rooms the student needs to walk through before picking up goal.

Last but not least, this variant $\text{MDP}'_{3,1}$ also further justifies why we need three levels in this MazeBase+ example: if we stopped at level 2, we would still need at least three times more iterations when learning new problems, because we would not have encapsulated the learned knowledge of $\overline{\pi}^{\text{concat}}$, so we would need to let the student learn the process represented by $\overline{\pi}^{\text{concat}}$ repeatedly, while at the same time finding the correct navigation between rooms. See Fig. 4 for the comparison and more detailed analysis.

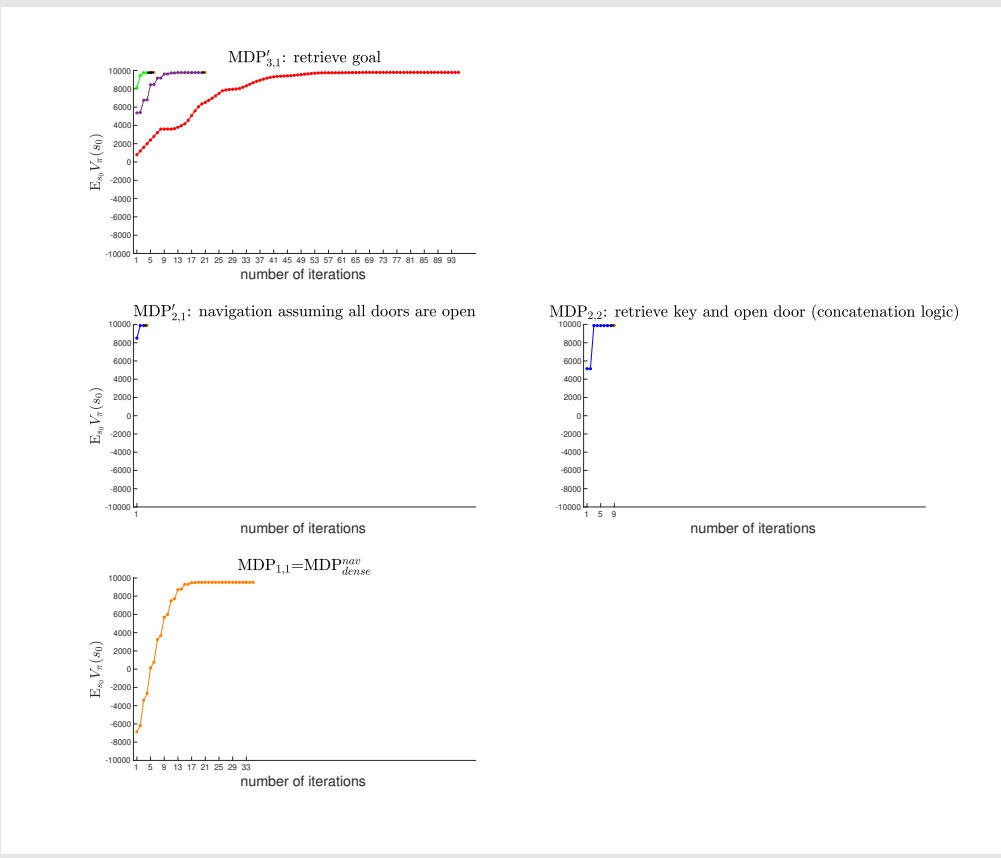

Figure 4: We display $\mathbb{E}_{s_0} V_\pi(s_0)$, where $V_\pi$ is the value function for $\text{MDP}_{1,1}$, $\text{MDP}'_{2,1}$, $\text{MDP}_{2,2}$, and $\text{MDP}'_{3,1}$ respectively, as $\pi$ is optimized during iterations of classical value iteration and of value iteration within our algorithm. See Fig. 3 for the detailed explanation, and Fig. 2 for the representation of this second experiment of the MazeBase+ example. One addition here compared with fig.3 is that here we also plot iterations within $\text{MDP}'_{3,1}$ at level 2 in purple followed by iterations at level 1 in yellow, assuming the student did not solve $\text{MDP}_{2,2}$ and consequently the assistant did not extract $\overline{\pi}^{\text{concat}}$. This corresponds to treating $\text{MDP}'_{3,1}$ as an MDP of difficulty 2. The extra effort here (purple+yellow) compared with the original case (green+blue+orange) demonstrates the advantage of treating $\text{MDP}'_{3,1}$ as an MDP of difficulty 3 and of extracting the higher-order function $\overline{\pi}^{\text{concat}}$.

**Box 12: Online learning illustrated in the MazeBase+ example**

App. E.1.5 details the mathematical notation(s) mentioned for this box. When new MDPs come into the curriculum, such as $\text{MDP}'_{2,1}$ and $\text{MDP}'_{3,1}$ here, the student can solve them following the strict lexicographic order defined on the MDPs, while the assistant extracts out new skills, such as $(\overline{\pi}^{\text{nav}})'$ and adds to the public skill set. The student can also utilize all the skills in the skill set, both the ones already there and ones newly added, when solving the MDPs. For instance, the student utilizes $\overline{\pi}^{\text{concat}}$ which are already there in the skill set, and $(\overline{\pi}^{\text{nav}})'$, which is newly added when solving $\text{MDP}'_{3,1}$, in order to solve it faster. In this way, all the current learned knowledge does not need to be relearned again, so this whole process is in a fully online fashion.

---

**Box 13: Robustness in the refinement procedure in the example of navigation and transportation with traffic jams**

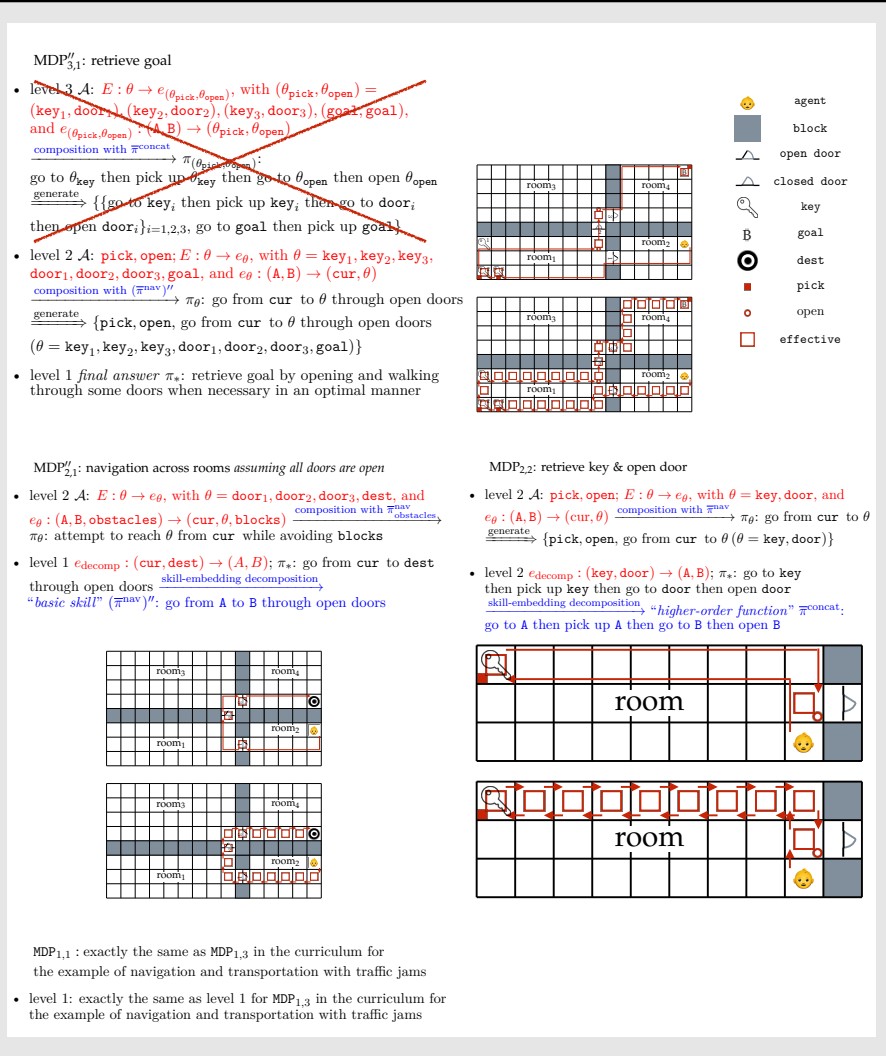

Figure 5: A representation of the third experiment of the MazeBase+ example. The representation in this figure is similar to the one for the first experiment (Fig. 1). In the third experiment, we consider the situation where the optimal policy at the highest difficulty level, when pushed down to a finer level, yields a suboptimal policy where the student collects $\text{key}_2$, opens $\text{door}_2$, and then collects $\text{key}_3$ (very close to $\text{key}_2$) to then open $\text{door}_3$. Within our algorithm, this suboptimal policy gets refined in order to yield the optimal policy, demonstrating the robustness of our optimization procedure; in this case it is also the case that our algorithm still outperforms naïve value iteration (see Fig. 6).

App. E.2.4 details the mathematical notation(s) mentioned for this box. We introduce a new problem $\mathrm{MDP}''_{3,1}$, similar to $\mathrm{MDP}_{3,1}$, with the main difference lying in the geometric configuration of the objects. We omit the details of the changes, but hope to emphasize that one main change is that now both $\mathrm{key}_2$ and $\mathrm{key}_3$ are in $\mathrm{room}_1$, and they are next to each other, so the optimal policy is to pick up both keys, before opening $\mathrm{door}_2$, rather than pick up $\mathrm{key}_2$ then open $\mathrm{door}_2$ and go back to pick up $\mathrm{door}_3$ then open $\mathrm{door}_3$ as hinted by $\overline{\pi}^{\mathrm{concat}}$. In this situation, the optimal policy at the highest difficulty level, when pushed down to a finer level, is suboptimal and requires, within our algorithm, significant refinement in order to yield the optimal policy, demonstrating the robustness of our optimization procedure: see Fig. 5; Figure 6 shows that even in this setting our algorithm outperforms naïve value iteration.

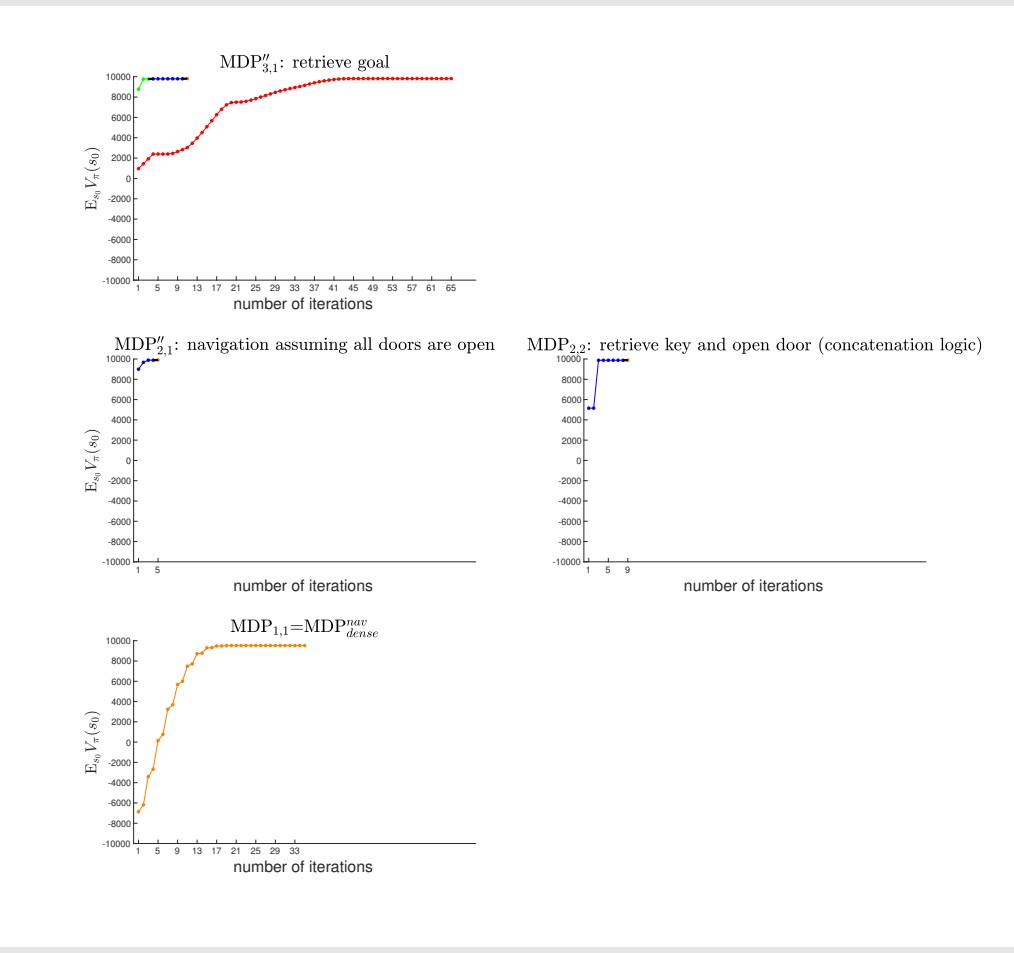

Figure 6: We display $\mathbb{E}_{s_0} V_\pi(s_0)$, where $V_\pi$ is the value function for $\mathrm{MDP}_{1,1}$, $\mathrm{MDP}''_{2,1}$, $\mathrm{MDP}_{2,2}$, and $\mathrm{MDP}''_{3,1}$ respectively, as $\pi$ is optimized during iterations of classical value iteration and of value iteration within our algorithm. See Fig. 3 for the detailed explanation, and see Fig. 5 for the representation of this third experiment of the MazeBase+ example.

## C.2 NAVIGATION AND TRANSPORTATION WITH TRAFFIC JAMS

In this section, we provide another example curricula: navigation and transportation with traffic jams, an example with multiple action factors that the first example MazeBase+ is based on. Please see Fig. 7 for a comprehensive summary of this example, and see Fig. 8 for the corresponding curriculum.

**Box 14: MDPs in the example of navigation and transportation with traffic jams**

$\{\text{MDP}_{2,n}\}_{n=1}^{6} = \{\text{MDP}_\kappa\}_{\kappa \in \mathcal{K}}$:
navigation & transportation with a few heavy-traffic roads

- level 2 $\mathcal{A}$: mc, car; $E: \theta \to e_\theta$, with $\theta = \mathcal{D}_{\text{pause}}, \mathcal{D}_{\text{pause}}^c$, and

  $e_\theta : (\text{A}, \text{B}, \text{obstacles}) \to (\text{cur}, \text{dest}, \text{jams})$ supported only on $\theta \xrightarrow{\text{composition with } \overline{\pi}_{\text{obstacles}}^{\text{nav}}}$
  $\pi_\theta$: attempt to reach dest from cur while avoiding traffic jams within $\theta \xrightarrow{\text{generate}}$ {attempt to reach dest from cur following $\pi_\theta$ within $\theta$ using $\theta'$ :
  $\theta = \mathcal{D}_{\text{pause}}, \mathcal{D}_{\text{pause}}^c; \theta' = \text{mc, car}\}$

- level 2 $e_{\text{decomp}} : (\mathcal{D}_{\text{pause}}, \mathcal{D}_{\text{jams}}) \to (\mathcal{D}_\alpha, \mathcal{D}_\beta); \pi_*$: go from cur to dest following $\pi_{\mathcal{D}_{\text{pause}}}$ if in $\mathcal{D}_{\text{pause}}$ and $\pi_{\mathcal{D}_{\text{pause}}^c}$ otherwise, while using car if in $\mathcal{D}_{\text{jams}}$ and mc otherwise $\xrightarrow{\text{skill-embedding decomposition}}$ "higher-order function" $\overline{\pi}^{\text{transport}}$: go from cur to dest following $\pi_{\mathcal{D}_\alpha}$ if in $\mathcal{D}_\alpha$ and $\pi_{\mathcal{D}_\alpha^c}$ otherwise, while using car if in $\mathcal{D}_\beta$ and mc otherwise

- level 1 *final answer* $\pi_*$: go from cur to dest while avoiding traffic jams and selecting the correct transportation tool in each movement

$\text{MDP}_{2,7}$: navigation & transportation with more light-traffic roads

- level 2 $\mathcal{A}$: exactly the same construction as of level 2 $\mathcal{A}$ for $\{\text{MDP}_{2,n}\}_{n=1}^{6}$

- level 2 $e_{\text{comp}} : (\mathcal{D}_\alpha, \mathcal{D}_\beta) \to (\mathcal{D}_{\text{pause}}, \mathcal{D}_{\text{jams}})$
  $\xrightarrow[\text{composition with } \overline{\pi}^{\text{transport}}]{\text{value iteration}}$
  $\pi_*$: go from cur to dest following $\pi_{\mathcal{D}_{\text{pause}}}$ if in $\mathcal{D}_{\text{pause}}$ and $\pi_{\mathcal{D}_{\text{pause}}^c}$ otherwise, while using car if in $\mathcal{D}_{\text{jams}}$ and mc otherwise

- level 1 *final answer* $\pi_*$: go from cur to dest while avoiding traffic jams and selecting the correct transportation tool in each movement

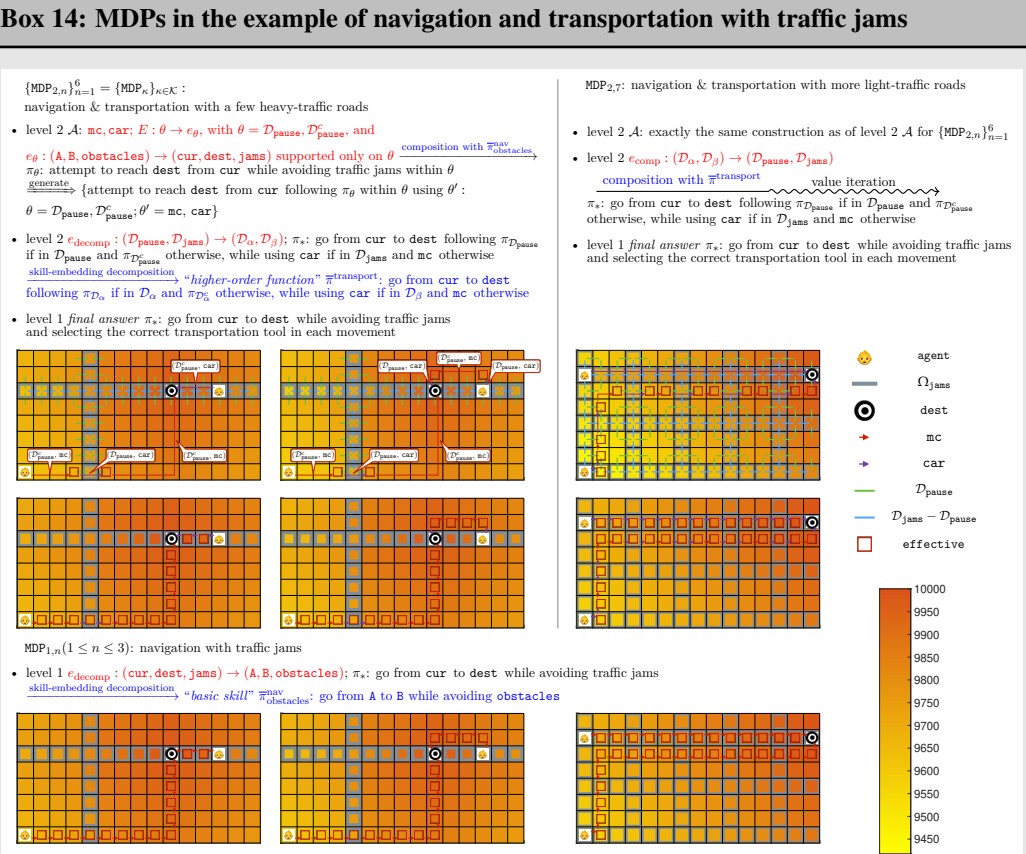

$\text{MDP}_{1,n}(1 \leq n \leq 3)$: navigation with traffic jams

- level 1 $e_{\text{decomp}} : (\text{cur}, \text{dest}, \text{jams}) \to (\text{A}, \text{B}, \text{obstacles}); \pi_*$: go from cur to dest while avoiding traffic jams $\xrightarrow{\text{skill-embedding decomposition}}$ "basic skill" $\overline{\pi}_{\text{obstacles}}^{\text{nav}}$: go from A to B while avoiding obstacles

Figure 7: A representation of the example of navigation and transportation with traffic jams, with curriculum in Fig. 8. The representation in this figure is similar to the one for the first experiment (Fig. 1). The goal of the agent is to learn optimal policies to travel from an initial position in a grid world to a known final state in any position in the grid world. Some regions, $\Omega_{\text{jams}}$, in the grid world have traffic jams (of different degrees of severity). The agent will have at its disposal two means of transportation, mc and car, with car having different velocities in $\Omega_{\text{jams}}$ and $\Omega_{\text{jams}}^c$. The curriculum has two sets of problems, corresponding to two difficulties. The problems $\{\text{MDP}_{1,n}\}_{n=1}^{3}$ of difficulty 1, at the bottom of the figure, are designed to teach the agent how to navigate, without a choice of means of transportation, and only basic up/down/left/right moves (red arrows). Different $n$'s correspond to different velocities in the $\Omega_{\text{jams}}$ regions with car, and, for $n = 3$, a different choice of $\Omega_{\text{jams}}$. The agent will learn optimal policies for these problems that become basic skills $\overline{\pi}_{\text{obstacles}}^{\text{nav}}$, to be utilized in higher level MDPs (upper part of the figure). The problems $\{\text{MDP}_\kappa\}_{\kappa \in \mathcal{K}}$ of difficulty 2 involve both navigation and means of transportation; different $\kappa$'s correspond to different velocities in the $\Omega_{\text{jams}}$ regions with car, separated into two classes $\mathcal{K}_1$ (first column) and $\mathcal{K}_2$ (second column), requiring two different optimal navigation rules because of the different velocities in $\Omega_{\text{jams}}$ vs. $\Omega_{\text{jams}}^c$. A single action at level 2 corresponds to an entire skill at level 1, up to a composition/decomposition with family of maps that we call embeddings (represented by corresponding different types of arrows; this will be detailed later), and therefore leads to a long and complex path (long red arrow). Roughly speaking, each skill is a parametric family of policies, and in some of the figures above we display the actual parameter for each action (at level 2); for example $(\mathcal{D}_{\text{pause}}^c, \text{mc})$ denotes the action at level 2 consisting of navigating with mc in a region not requiring changes in the means of transportation; $(\mathcal{D}_{\text{pause}}, \text{car})$ denotes the action at level 2 consisting of navigating with car in a region that does require a change in the means of transportation. The effective state space (red squares) at level 2, consisting of final states of level 2 actions, is much smaller than at level 1. This leads to a significant speed-up in learning MDPs at level 2, because we reduce the space of policies we search over.

Figure 7: (Continued) The gray lines between adjacent grid points represent that the roads between them have certain traffic conditions: the thicker the lines, the heavier the traffic. The value functions are represented by heat maps; optimal policies starting from certain initial states (which could be at any location) are represented by concatenations of red arrows (single actions at that level) and red squares (effective states at that level), again showing the significant reduction in the effective state space and in the number of actions at level 2. The third column shows an MDP with a different configuration of traffic jam regions, often requiring many changes in the means of transportation, increasing the difficulty of learning: we will use transfer learning of higher-order functions to speed up the learning process.

App. E.2.1 details the equation(s) needed for this box. We now introduce a pedagogical "navigation and transportation with traffic jams" example to walk the reader through the concepts and notations we introduce. Please see Fig. 7 for a comprehensive summary of this example, and see Fig. 8 for the corresponding **curriculum** (as defined in Sec. 4).

Given a two-dimensional grid world $\Omega := [x_1, x_2] \times [y_1, y_2] \subseteq \mathbb{N} \times \mathbb{N}$, a first action factor $\mathcal{A}_1 := \mathcal{A}_{\mathrm{dir}} \cup \{a^{\mathrm{end}}\}$ with the set of moves in different directions given by $\mathcal{A}_{\mathrm{dir}} := \{(1,0), (0,1), (-1,0), (0,-1)\}$, and a second action factor $\mathcal{A}_2 := \mathcal{A}_{\mathrm{means}} \cup \{a^{\mathrm{end}}\}$, with the set of uses of different means of transportation given by $\mathcal{A}_{\mathrm{means}} := \{\mathtt{mc}, \mathtt{car}\}$,

$$\{\{\mathrm{MDP}_\kappa\}_{\kappa \in \mathcal{K}_1}, \{\mathrm{MDP}_\kappa\}_{\kappa \in \mathcal{K}_2}\} \qquad \mathrm{MDP}_{2, |\mathcal{K}_1| + |\mathcal{K}_2| + 1}$$

$$\uparrow \qquad\qquad\qquad\qquad \uparrow$$

$$\{\mathrm{MDP}_{1,1}, \mathrm{MDP}_{1,2}\} \qquad\qquad \mathrm{MDP}_{1,3}$$

Figure 8: Curriculum for the example of navigation and transportation with traffic jams. The meaning of the arrows here are the same as in Fig. 1: an arrow means the knowledge learned from one MDP (starting point of the arrow) is utilized when solving another MDP (endpoint of the arrow). The line in the middle separates the two groups of MDPs.

motorcycle and car respectively, having different velocities (and corresponding rewards, which we detail momentarily). We try to teach an agent (student) how to travel from point $s$ to point $s'$ while selecting the most efficient means of transportation for each step along the way. To mimic a variety of road conditions, we add two roads of heavy traffic: $[x_1, x_2] \times \{y_3\}$ and $\{x_3\} \times [y_1, y_2]$, where $x_1 < x_3 < x_2, y_1 < y_3 < y_2$ (here we take $x_1 = 1, x_2 = 15, x_3 = 5, y_1 = 1, y_2 = 8, y_3 = 6$), and we denote their union as $\Omega_{\mathrm{jams}}$. The traffic rules are: generally, the agent could use either the motorcycle or the car, with corresponding speeds $v_{\mathrm{mc}}, v_{\mathrm{car}}$, satisfying $v_{\mathrm{mc}} = 1 > v_{\mathrm{car}} = 0.6$; if the agent is traveling along, entering, or leaving any of these two roads with $\mathtt{mc}$, it will receive a large negative penalty in the reward; with $\mathtt{car}$ its speed will be $\kappa$ for some $\kappa < v_{\mathrm{car}}$. The reward at each step is $\frac{r_0}{v}$, with $r_0 < 0$ (set to $-10$ here), and $v$ the agent's current speed, i.e., $v = v_{\mathrm{mc}}$ if the agent has chosen $\mathtt{mc}$ in the current action, $v = \kappa$ if the agent is traveling along, entering, or leaving any of the two roads of heavy traffic with $\mathtt{car}$, and otherwise $v = v_{\mathrm{car}}$.

We consider a family $\{\mathrm{MDP}_\kappa\}_{\kappa \in \mathcal{K}}$ of MDPs as above, with varying $\kappa$, which can and will affect the optimal policy. The formal definition of these MDPs is postponed to App. E.2.1; here we specify the state space and action set as follows:

$$\mathcal{S} := \{(s_{\mathrm{cur}}, s_{\mathrm{dest}}) : s_{\mathrm{cur}}, s_{\mathrm{dest}} \in \Omega\} = \Omega \times \Omega \quad, \quad \mathcal{A} := (\mathcal{A}_{\mathrm{dir}} \cup \{a^{\mathrm{end}}\}) \times (\mathcal{A}_{\mathrm{means}} \cup \{a^{\mathrm{end}}\}).$$

Here the teacher has the role to set all the parameters of the problems, including the reward at $\mathtt{dest}$ or the negative reward for driving along/entering/leaving $\Omega_{\mathrm{jams}}$ using $\mathtt{mc}$. The value of $\kappa$ affects the speed of the car in traffic regions.

Observe that there will be entanglement between the two action factors, modeling navigation and means of transportation, along typical optimal routes. In order to efficiently solve this family of problems, we exploit the similarities between them to enable potential transfer; we start by

disentangling the action factors and introducing partial policies, and we will then combine these partial policies across action factors to obtain full optimal policies.

---

**Box 15: Partial policy generators in the example of navigation and transportation with traffic jams**

App. E.2.3 details the equation(s) needed for this box. We define the transition region $\mathcal{D}_{\text{pause}} \subseteq \mathcal{SA}_{\text{dir}}$ as the set of all the state-action pairs that either enter or leave the region $\Omega_{\text{jams}}$. See Fig. 7 for an illustration of $\mathcal{D}_{\text{pause}}$, where we use an undirected edge to represent two directed elements in $\mathcal{D}_{\text{pause}}$, for the sake of not cluttering the image.

For the previously defined $\text{MDP}_\kappa$, we will have two types of partial policy generators, corresponding to the two action factors $\mathcal{A}_{\text{dir}} \cup \{a^{\text{end}}\}$ and $\mathcal{A}_{\text{means}} \cup \{a^{\text{end}}\}$, which we *name* $(g_\kappa^1)_{\text{dir}}(\kappa \in \mathcal{K})$ and $(g^1)_{\text{means}}$, with active action factor sets $I_1 := \text{dir}$ and $I_2 := \text{means}$, respectively. The role of the upper index $1$ in the names will become apparent later as indicator of the "level" of these MDPs within an MMDP.

For each $\kappa \in \mathcal{K}$, $(g_\kappa^1)_{\text{dir}} : \Theta_{\text{dir}} \to \{(\pi_\theta^{\text{nav}_\kappa})_{\text{dir}}\}_{\theta \in \Theta_{\text{dir}}}$ is defined by $(g_\kappa^1)_{\text{dir}}(\theta) := (\pi_\theta^{\text{nav}_\kappa})_{\text{dir}}$, where $\Theta_{\text{dir}}$ indexes partial policies $(\pi_\theta^{\text{nav}_\kappa})_{\text{dir}} : \mathcal{SA}_{\text{dir}}^1 \to [0,1]$, such that each $(\pi_\theta^{\text{nav}_\kappa})_{\text{dir}}$ takes positive values only in $\mathcal{D}_{\text{pause}}^c$ or only in $\mathcal{D}_{\text{pause}} \cup \{(s_{\text{cur}}, s_{\text{goal}}), (a^{\text{end}}, 0)\}$. This partial policy is oblivious of the means of transportation, and represents only navigation along paths. This partial policy generator conveys information: $\mathcal{D}_{\text{pause}}$ suggests where the agent may change the means of transportation, and $\mathcal{D}_{\text{pause}}^c$ suggests that there might not be a benefit in changing the means of transportation, so the agent may focus on navigation only.

The other partial policy generator $(g^1)_{\text{means}}$ is simpler. It is defined on $\Theta_{\text{means}} := \mathcal{A}_{\text{means}}$, and $(g^1)_{\text{means}} : \mathcal{A}_{\text{means}} \to \{(\pi_{\theta'})_{\text{means}}\}_{\theta' \in \mathcal{A}_{\text{means}}}$, mapping $\theta'$ to $(\pi_{\theta'})_{\text{means}}$. It generates two partial policies $\{(\pi_{\theta'})_{\text{means}} : \theta' \in \mathcal{A}_{\text{means}}\}$, with $(\pi_{\theta'})_{\text{means}} : \mathcal{SA}_{\text{means}}^1 \to \{0,1\}$ (defined in equation E.33) representing the partial policy of choosing a fixed means of transportation $\theta'$.

---

**Box 16: Partial policy generator set in the example of navigation and transportation with traffic jams**

App. E.2.3 details the equation(s) needed for this box. For the first level, $\text{MDP}_\kappa^1 := (\mathcal{S}^1, (\mathcal{S}^{\text{init}})^1, (\mathcal{S}^{\text{end}})^1, \mathcal{A}^1, P^1, R_\kappa^1, \Gamma^1) = (\mathcal{S}, \mathcal{S}^{\text{init}}, \mathcal{S}^{\text{end}}, \mathcal{A}, P, R_\kappa, \Gamma)$. The teacher provides

$$\mathcal{G}^1 := \{\{(g_\kappa^1)_{\text{dir}}\}_{\kappa \in \mathcal{K}}, (g^1)_{\text{means}}\}, \tag{C.2}$$

whose elements were defined above. Notice that it may happen that $(g_\kappa^1)_{\text{dir}} = (g_{\kappa'}^1)_{\text{dir}}$ for $\kappa \neq \kappa'$. The student constructs the set of policies $\Pi^1$ generated from the set of partial policies $\{(\pi_\theta^{\text{nav}_\kappa})_{\text{dir}}\}_{\theta \in \Theta_{\text{dir}}, \kappa \in \mathcal{K}} \cup \{(\pi_{\theta'})_{\text{means}}\}_{\theta' \in \mathcal{A}_{\text{means}}}$:

$$\Pi^1 = \{(\pi_\theta^{\text{nav}_\kappa})_{\text{dir}} \otimes (\pi_{\theta'})_{\text{means}} : \theta \in \Theta_{\text{dir}}, \theta' \in \mathcal{A}_{\text{means}}, \kappa \in \mathcal{K}\},$$

where, as in equation 2.1, $(\pi_\theta^{\text{nav}_\kappa})_{\text{dir}} \otimes (\pi_{\theta'})_{\text{means}} : \mathcal{SA}^1 \to [0,1]$ is, using the definition of $(\pi_{\theta'})_{\text{means}}$ in equation E.33,

$$((\pi_\theta^{\text{nav}_\kappa})_{\text{dir}} \otimes (\pi_{\theta'})_{\text{means}})((s_{\text{cur}}, s_{\text{dest}}), (a_{\text{dir}}, a_{\text{means}}))$$
$$= (\pi_\theta^{\text{nav}_\kappa})_{\text{dir}}((s_{\text{cur}}, s_{\text{dest}}), (a_{\text{dir}}, 0)) \times \mathbb{1}_{\{\theta'\}}(a_{\text{means}}).$$

The right hand side, once $(\pi_\theta^{\text{nav}_\kappa})_{\text{dir}}$ will be learned, represents a policy for navigating from $s_{\text{cur}}$ to $s_{\text{dest}}$ within the domain $\text{dom}_\theta \in \{\mathcal{D}_{\text{pause}}, \mathcal{D}_{\text{pause}}^c\}$ using only $a_{\text{means}} = \theta'$. Note that each policy in $\Pi^1$ is represented by an element in the product set $(\Theta_1 \cup \{\texttt{null}\})^{|\mathcal{K}|} \times (\Theta_2 \cup \{\texttt{null}\})$, with $\Theta_1 := \Theta_{\text{dir}} \cup \{a^{\text{end}}\}$, and $\Theta_2 := \mathcal{A}_{\text{means}} \cup \{a^{\text{end}}\}$. For instance, $(\pi_\theta^{\text{nav}_\kappa})_{\text{dir}} \otimes (\pi_{\text{car}})_{\text{means}}$ is represented by a vector with $\theta$ for the entry corresponding to $\kappa$ and $\texttt{car}$ for the last entry, and $\texttt{null}$ for all the other $|\mathcal{K}| - 1$ entries; it corresponds to a policy of navigating inside $\text{dom}_\theta$ from $s_{\text{cur}}$ to $s_{\text{dest}}$ with $\texttt{car}$; if $\text{dom}_\theta = \mathcal{D}_{\text{pause}}^c$, note that this policy will lead to $s_{\text{dest}}$ only when $s_{\text{cur}}$ and $s_{\text{dest}}$ are not separated by $\Omega_{\text{jams}}$.

These partial policy generators are ripe for being used to build higher-level actions for MMDPs, as we now discuss.

---

**Box 17: Inputs for the construction of MMDPs in the example of navigation and transportation with traffic jams**

App. E.2.3 details the equation(s) needed for this box. The provided **sequence of generator sets** $\{\mathcal{G}^l\}_{l=1}^\infty$ (same for any $\text{MDP}_\kappa$) for this example consists of $\mathcal{G}^1$ defined as in Box 6, and $\mathcal{G}^l := \varnothing$ for $l \geq 2$. There are multiple options for $\{\mathcal{G}^l_{\kappa,\text{test}}\}_{l=1}^\infty$ here, with these restrictions: (1) $\pi_{\kappa,*} \notin \mathcal{G}^1_{\kappa,\text{test}}$; (2) $\pi^2_{\kappa,*} \in \mathcal{G}^2_{\kappa,\text{test}}$, with $\pi_{\kappa,*} = \pi_\kappa$, $\pi^2_{\kappa,*}$ being defined in equation E.37, equation E.35 respectively. These two conditions together guarantee that $\{\text{MDP}_\kappa\}_{\kappa \in \mathcal{K}}$ are all of difficulty 2 as shown later. The timescales of both $(g^1_\kappa)_{\text{dir}}$ and $(g^1)_{\text{means}}$ are $+\infty$. We let $r^1 = -10$.

The student then constructs the second-level $\text{MDP}^2_\kappa := (\mathcal{S}, \mathcal{S}^{\text{init}}, \mathcal{S}^{\text{end}}, \overline{\Pi^1}, P^2, R^2_\kappa, \Gamma^2)$, using the inputs above and the procedures we describe momentarily.

---

**Box 18: Unpacking compressed policies in the example of navigation and transportation with traffic jams**

App. E.2.3 details the equation(s) needed for this box. Once the second-level MDP $\text{MDP}^2_\kappa = (\mathcal{S}, \mathcal{S}^{\text{init}}, \mathcal{S}^{\text{end}}, \overline{\Pi^1}, P^2, R^2_\kappa, \Gamma^2)$, for each $\kappa \in \mathcal{K}$, is constructed using the procedures above, it can be solved to find the optimal policy $\pi^2_{\kappa,*}$. We construct stochastic trajectories starting from each state $s \in \mathcal{S}$ by "gluing" together the actions $(\pi^{\text{nav}_{\kappa'}}_\theta)_{\text{dir}} \otimes (\pi_{\theta'})_{\text{means}}$ (defined in Box 16) following an order of the form $(\pi^{\text{nav}_{\kappa_0}}_{\theta_0})_{\text{dir}} \otimes (\pi_{\theta'_0})_{\text{means}}, \cdots, (\pi^{\text{nav}_{\kappa_t}}_{\theta_t})_{\text{dir}} \otimes (\pi_{\theta'_t})_{\text{means}}, \cdots$: the agent starts from $S_0 = s$ in $\text{MDP}^2_\kappa$, and chooses actions in $A_t = (\pi^{\text{nav}_{\kappa_t}}_{\theta_t})_{\text{dir}} \otimes (\pi_{\theta'_t})_{\text{means}}$ for any $0 \leq t \leq \tau - 1$ with $\tau$ being the first time $t$ such that $S_t \in \mathcal{S}^{\text{end}}$ (if such event does not occur, we set $\tau = +\infty$). For each state $s$, the sequence of actions needs to be optimized to maximize the expected cumulative rewards along the stochastic trajectories. Following the description here, the value $\pi^2_{\kappa,*}(s, (\pi^{\text{nav}_\kappa}_\theta)_{\text{dir}} \otimes (\pi_{\theta'})_{\text{means}})$, for each $\theta \in \Theta_{\text{dir}}, \theta' \in \mathcal{A}_{\text{means}}$, $s = (s_{\text{cur}}, s_{\text{dest}})$, equals equation E.35. Note that the minimization of $\theta'_t$ is trivial at any state $s$. Here $\mathcal{D}_{\text{jams}} := \{(s, (\text{dir}, 0)) \in \mathcal{SA}^1_{\text{dir}} : s_{\text{cur}} \in \Omega_{\text{jams}} \text{ or } s_{\text{cur}} + a_{\text{dir}} \in \Omega_{\text{jams}}\}$ is the set of state-action pairs such that either the agent's current location is in $\Omega_{\text{jams}}$, or the agent intends to move to $\Omega_{\text{jams}}$; $\mathcal{D}^c_{\text{jams}} := \mathcal{SA}^1_{\text{dir}} - \mathcal{D}_{\text{jams}}$. See Fig. 7 for an illustration of these important subsets of $\mathcal{SA}_{\text{dir}} = \{(s_{\text{cur}}, s_{\text{dest}}), (a_{\text{dir}}, 0)\}$ appearing in this higher-order policy $\pi^2_{\kappa,*}$, with the directions of edges omitted because the existence and colors of the edges are always the same when reversing the directions of the edges. In addition, recall that for any two (random) times $0 \leq T < T' < \infty$ (a.s.), $(R^2_\kappa)_{T,T'}$ is defined as in equation B.1 for $\text{MDP}^2_\kappa$. $\text{MDP}^2_\kappa$ has at least two key advantages compared to the level 1 MDP: first, it has much shorter time horizon, as a single action moves the agent by multiple steps all within a connected region of $\mathcal{D}_{\text{pause}}$ or $\mathcal{D}^c_{\text{pause}} - \{(s_{\text{cur}}, s_{\text{dest}}), (a^{\text{end}}, 0)\}$; second, the stochasticity is greatly reduced as it is absorbed into each higher-level navigation policy, further simplifying the optimization in equation E.36.

Note how equation E.35 and equation E.36 cause an interplay between the tensor-product structure of $A_t$ and the geometry of $\Omega_{\text{jams}}$: the student needs to stop and reconsider which means of transportation to use every time it enters or leaves $\Omega_{\text{jams}}$. Also note that it is crucial here that the navigation policies used at this level terminate, by choosing $a^{\text{end}}$ at very precise times and locations, instead of relying on random stopping times.

Finally, the student solves the original MDPs, by using equation B.7 to pass the optimal policy of each $\text{MDP}^2_\kappa$ down to level one, resulting in the policy $\pi_\kappa$ as in equation E.37. In this particular example, $\pi_\kappa$ is in fact the optimal policy $\pi_{\kappa,*}$ of the original MDP, requiring no additional refinement by value iteration.

**Box 19: Multi-level compression in the example of navigation and transportation with traffic jams**

Box 15 details the mathematical notation(s) mentioned for this box. For this example, we do not provide the explicit formulas for the compressed MDPs, because the policies here are relatively complicated (in particular for the parts $\{(\pi_\theta^{\mathrm{nav}_\kappa})_{\mathrm{dir}}\}_{\theta \in \Theta_{\mathrm{dir}}, \kappa \in \mathcal{K}}$).

**Box 20: Composing partial policy generators in the example of navigation and transportation with traffic jams**

App. E.2.4 details the equation(s) needed for this box. In the example of navigation and transportation with traffic jams, we now explain how the partial policy generators $(g_\kappa^1)_{\mathrm{dir}}(\kappa \in \mathcal{K})$, $(g^1)_{\mathrm{means}}$ in $\mathcal{G}^1$ can be constructed by composing skills with two embedding generators $E_\alpha^1$ and $E_\beta^1$.

For each $\kappa \in \mathcal{K}$, $(g_\kappa^1)_{\mathrm{dir}}$ is the composition of $\overline{\pi}_{\mathrm{obstacles}}^{\mathrm{nav}_\kappa}$ and $E_\alpha^1$, which we now define. We let the navigation skill $\overline{\pi}_{\mathrm{obstacles}}^{\mathrm{nav}_\kappa}$ : $\{(s_{\mathrm{cur}}, s_{\mathrm{dest}}, a_{\mathrm{dir}}) : s_{\mathrm{cur}}, s_{\mathrm{dest}} \in \Omega, a_{\mathrm{dir}} \in \mathcal{A}_{\mathrm{dir}} \cup \{a^{\mathrm{end}}\}\} \to [0, 1]$, with timescale $+\infty$, be essentially the same as $(\pi^{\mathrm{nav}_\kappa})_{\mathrm{dir}}$: $\overline{\pi}_{\mathrm{obstacles}}^{\mathrm{nav}_\kappa}(s_{\mathrm{cur}}, s_{\mathrm{dest}}, a_{\mathrm{dir}}) := (\pi^{\mathrm{nav}_\kappa})_{\mathrm{dir}}((s_{\mathrm{cur}}, s_{\mathrm{dest}}), (a_{\mathrm{dir}}, 0))$, which coincides with $(\pi_{\mathcal{D}_{\mathrm{pause}}}^{\mathrm{nav}_\kappa})_{\mathrm{dir}}$ on $\mathcal{D}_{\mathrm{pause}}$ and $(\pi_{\mathcal{D}_{\mathrm{pause}}^c}^{\mathrm{nav}_\kappa})_{\mathrm{dir}}$ on $\mathcal{D}_{\mathrm{pause}}^c$, introduced in Box 16. Such a skill $\overline{\pi}_{\mathrm{obstacles}}^{\mathrm{nav}_\kappa}$ will be learned from some other auxiliary MDPs in a "curriculum" focusing on navigation only (as discussed momentarily).

$E_\alpha^1$ is defined as in equation E.40. While $(e^1)_\theta^{\mathrm{dir}}$ in the definition of $E_\alpha^1$ may look like a glorified identity, the crucial point here is that $E_\alpha^1$ lets the teacher provide the student with the information that $\mathcal{D}_{\mathrm{pause}}$ and $\mathcal{D}_{\mathrm{pause}}^c$ is a partition of $\mathcal{SA}_{\mathrm{dir}}^1$ important for learning an optimal policy, since $(e^1)_\theta^{\mathrm{dir}}$ is supported on $\theta$, where $\theta \in \Theta_{\mathrm{dir}}$.

$(g^1)_{\mathrm{means}}$ is the composition of the degenerate skill, which is the identity map on $[0, 1]$, and $E_\beta^1$ as defined in equation E.41.

**Box 21: Designed curriculum in the example of navigation and transportation with traffic jams**

App. E.2.1 details the equation(s) needed for this box. In the example of navigation and transportation with traffic jams, the teacher provides a curriculum containing two types of MDPs: $\mathrm{MDP}_{1,n}(1 \le n \le n_1 := 2)$ and $\mathrm{MDP}_{2,n}(1 \le n \le n_2 := |\mathcal{K}| = 6)$. $\mathrm{MDP}_{1,n}$, of difficulty 1 and with detailed definition in equation E.32, teaches the student to navigate through $\Omega$, with no choice of means of transportation, but with $n_1 = 2$ different values of the parameter $\kappa$, corresponding to light and heavy traffic jams, and inducing different optimal policies. The MDPs $\{\mathrm{MDP}_{2,n}\}_{1 \le n \le n_2}$, of difficulty 2, are $\{\mathrm{MDP}_\kappa\}_{\kappa \in \mathcal{K}}$ as previously mentioned (with detailed definition in equation E.31), with $n_2 = |\mathcal{K}| = 6$; these are our main objectives and require combining navigation with transportation.

**Box 22: Merits of action factors in the example of navigation and transportation with traffic jams**

App. E.2.4 and Box 16 detail the mathematical notation(s) mentioned for this box. In each $\mathrm{MDP}_{2,n}$, the action set consists of two action factors $\mathcal{A}_{\mathrm{dir}} \cup \{a^{\mathrm{end}}\}$, $\mathcal{A}_{\mathrm{means}} \cup \{a^{\mathrm{end}}\}$ independent of each other, which takes charge of navigation and transportation respectively. This tensor product structure enables transfer to $\mathrm{MDP}_{2,n}$, for $1 \le n \le n_2$ of the skills $\overline{\pi}_{\mathrm{obstacles}}^{\mathrm{nav}_n}(1 \le n \le n_1)$ in the action factor $\mathcal{A}_{\mathrm{dir}} \cup \{a^{\mathrm{end}}\}$, with $\{\overline{\pi}_{\mathrm{obstacles}}^{\mathrm{nav}_n} : 1 \le n \le n_1\}$ exactly the same as $\{\overline{\pi}_{\mathrm{obstacles}}^{\mathrm{nav}_\kappa} : \kappa \in \mathcal{K}\}$, the set of skills we need for deriving $\mathcal{G}^1$ defined as in equation C.2 and constructing $\mathrm{MDP}_{2,n}^2$. The optimization over $\mathcal{A}_{\mathrm{means}}$ occurs at level 2 of $\mathrm{MDP}_{2,n}$, and the combination with $\mathcal{A}_{\mathrm{dir}}$ is optimized at level 1 of $\mathrm{MDP}_{2,n}$. In words: before going to a certain destination, the student first solves $\mathrm{MDP}_{1,n}$ to find virtual routes to the destination, then solving level 2 $\mathrm{MDP}_{2,n}$ yields the optimal means for those routes, based on traffic conditions, and finally

solving level 1 of $\mathtt{MDP}_{2,n}$ yields the optimal combination of routes and means of transportation. The embedding generator $E_\alpha^1$, provided by the teacher, allows the student to break routes at locations in $\mathcal{D}_{\mathtt{pause}}$, where it can then optimally change the means of transportation.

Merits of this are threefold. First, just like for a human, focusing on one problem at each time is more efficient than thinking about several problems all at once. Second, it can speed up the solution of the MDPs by leveraging transfer of skills within the curriculum, see Fig. 9, which generally comes from extracting the "repetitions"/repeated patterns (shared skills/components) and learning them for only once, or extracting the patterns that could be merged together into a single one and learning them in parallel. Third, it promotes further opportunities of transfer to other curricula; for instance, the first example MazeBase+ utilizes one of the navigation skills extracted in this example.

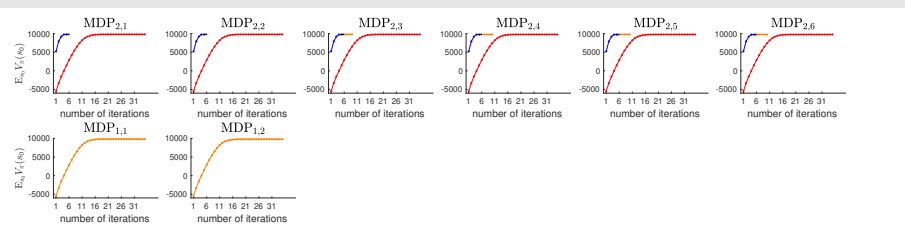

Figure 9: Similar to Fig. 3, we display $\mathbb{E}_{s_0} V_\pi(s_0)$, where $V_\pi$ is the value function for $\mathtt{MDP}_{1,n}$ and $\mathtt{MDP}_{2,n}$, as $\pi$ is optimized during iterations of classical value iteration (in red) and of value iteration within our algorithm, with iterations within $\mathtt{MDP}_{1,n}$ in orange, and iterations within $\mathtt{MDP}_{2,n}$ at level 2 in blue followed by iterations at level 1 in orange. Although we spend extra effort in solving the $\mathtt{MDP}_{1,n}$'s, they prepare us well enough so that we only need a few more iterations for solving the $\mathtt{MDP}_{2,n}$'s (blue+orange), much fewer than if we solved them from scratch using classical value iteration (red). These extra iterations correspond to learning how to stitch different pieces of routes separated by $\mathcal{D}_{\mathtt{pause}}$, which contains the turning points at which the student may need to switch the means of transportation. Of course, the cost of solving the $\mathtt{MDP}_{1,n}$'s (which for our algorithm is the same as for classical value iteration) is amortized over solving possibly many $\mathtt{MDP}_{2,n}$'s, showcasing the power of transfer in our framework. This is also analyzed in general by Sec. 4.1.

---

**Box 23: Transfer learning of the higher-order function in the example of navigation and transportation with traffic jams**

Apps. E.2.3–E.2.4 detail the equation(s) needed for this box. One issue of our framework in this example occurs when there are many roads of traffic (say, $\Omega'_{\mathtt{jams}} := [x_1, x_2] \times \{y_1, y_1 + 3, \cdots\}, \{x_1, x_1 + 3, \cdots\} \times [y_1, y_2]$), forcing the agent to switch the means of transportation every few steps. However, this problem can be resolved in our framework by using transfer learning to greatly speed up learning the second-level policy.

The main idea is that if we take out one element of $\mathcal{K}$, say $\kappa_1$, we observe that the second-level policy $\pi^2_{\kappa_1,*}$, given by equation E.35, depends on the exact choice of $\Omega_{\mathtt{jams}}$, but only mildly, and such dependence could be decomposed using an embedding. Semantically, $\pi^2_{\kappa_1,*}$ selects the index of the navigation partial policy, which determines both the support of the navigation policy and which navigation policy, and the means of transportation, independently. (Here for simplicity we restrict the embedding in the decomposition only to a single navigation policy and focus on selecting onto which support it is restricted.) We use the following logic: the agent selects the index $\theta$ if $(s_{\mathtt{cur}}, a_{\mathtt{dir}}) \in \theta$ for some $\theta \in \Theta_{\mathtt{dir}}$, and selects the means of transportation $\theta' = \mathtt{car}$ if $(s_{\mathtt{cur}}, a_{\mathtt{dir}}) \in \mathcal{D}_{\mathtt{jams}}$ and $\theta' = \mathtt{mc}$ otherwise. Here, $s_{\mathtt{cur}}$ is the agent's current location, and $a_{\mathtt{dir}}$ is the intended direction according to $(\pi^{\mathtt{nav}_{\kappa_1}})_{\mathtt{dir}}$, as introduced in Box 16. Notice here $\mathcal{D}_{\mathtt{jams}}$ or $\mathcal{D}_{\mathtt{pause}}$ are different when we change $\Omega_{\mathtt{jams}}$ to $\Omega'_{\mathtt{jams}}$, so we need this embedding to encapsulate the "if-conditions" in the logic above, after which a higher-order function could be extracted from the second-level policy $\pi^2_{\kappa_1,*}$, which is purely about the "if-then" logic, the core

part in $\pi^2_{\kappa_1,*}$ of interest to us, and also transferrable across different geometries, in some sense achieving "few-shot learning".

Formally, after learning $\pi^2_{\kappa_1,*}$, if the teacher provides the embedding defined as in equation E.42, then the assistant extracts a higher-order function for selecting the index of the navigation policy and the means of transportation $\overline{\pi}^{\text{transport}}$, defined in equation E.43 and with with timescale $t_{\overline{\pi}^{\text{transport}}} = +\infty$.

Next, we show how the transfer learning of this higher-order function could be achieved in this example. Given the new region of traffic jams $\Omega'_{\text{jams}} = [x_1, x_2] \times \{y_1, y_1 + 3, \cdots\} \cup \{x_1, x_1 + 3, \cdots\} \times [y_1, y_2]$ containing many more roads, displayed in the insets on the right column in Fig. 7, the teacher adds to the curriculum $\text{MDP}_{1,n_1+1}$, similar to $\text{MDP}_{1,n}$, and $\text{MDP}_{2,n_2+1}$, similar to $\text{MDP}_{2,n}$, except that $1/\kappa = 1.1$, and correspondingly $1/v_{\text{car}} = 1.05$, for both these new MDPs, describing a new scenario where the traffic jams are light. From now on we may also refer to $\text{MDP}_{1,n_1+1}$ as $\text{MDP}^{\text{nav}}_{\text{dense}}$.

The process to solve the two new MDPs in the curriculum uses the same Algs. 5–6, but we also extract a navigation skill $\overline{\pi}^{\text{nav}}_{\text{dense}}$, and we also have in hand the higher-order function for selecting the means of transportation $\overline{\pi}^{\text{transport}}$ extracted previously. Therefore, after the student constructs $\text{MDP}^2_{2,n_2+1}$, the teacher provides the hint to use the skill $\overline{\pi}^{\text{transport}}$ and the embedding $(e_{\text{comp}})_{2,n_2+1}$ for solving it, where $(e_{\text{comp}})_{2,n_2+1}$ is defined similarly to $(e_{\text{decomp}})^2_{2,1}$. Then, the student uses the composition of skill $\overline{\pi}^{\text{transport}}$ and $(e_{\text{comp}})_{2,n_2+1}$, which is similar to $\pi^2_{2,1,*}$, as the initial policy for solving $\text{MDP}^2_{2,n_2+1}$. This leads to a very fast learning of $\text{MDP}^2_{2,n_2+1}$. On the other hand, if we do not utilize this opportunity of transfer learning when following our framework to solve $\text{MDP}^2_{2,n_2+1}$, we will need many more iterations, because there are many pieces of routes separated by $\mathcal{D}'_{\text{pause}}$ when selecting means of transportation given that the roads with traffic are much more densely-distributed in $\Omega$ now. See Fig. 10 for a comparison between the two options.

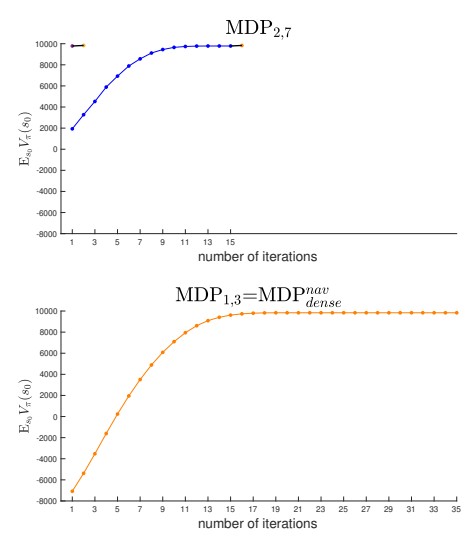

Figure 10: Similar to Fig. 9, we display $\mathbb{E}_{s_0} V_\pi(s_0)$, where $V_\pi$ is the value function for $\text{MDP}_{1,n+1}$ (in orange) and $\text{MDP}_{2,n_2+1}$, as $\pi$ is optimized during iterations of value iteration within our algorithm. More specifically, within $\text{MDP}_{2,n_2+1}$ we have iterations at level 2 in blue followed by iterations at level 1 in orange, where $\pi$ is optimized during iterations of value iteration within our algorithm without utilizing $\overline{\pi}^{\text{transport}}$; we also have iterations at level 2 in purple followed by iterations at level 1 in yellow, where $\pi$ is optimized within our algorithm utilizing $\overline{\pi}^{\text{transport}}$. Although we spend extra effort in extracting $\overline{\pi}^{\text{transport}}$, it prepares us well enough so that we could solve $\text{MDP}_{2,n_2+1}$ (purple+yellow) almost instantly, with much fewer iterations than if we solved it from scratch within our algorithm (blue+orange). This is also analyzed in general by Sec. 4.1.**??**.

**Box 24: Online learning illustrated in the example of navigation and transportation with traffic jams**

App. E.2.4 details the mathematical notation(s) mentioned for this box. Another comment is that Box 23 also shows that the student can achieve online learning in the current framework. When new MDPs come into the curriculum, such as $\mathrm{MDP}^1_{1,n_1+1}$ and $\mathrm{MDP}^2_{2,n_2+1}$ here, the student can solve them following the strict lexicographic order defined on the MDPs, while the assistant extracts out new skills, such as $\overline{\pi}^{\mathrm{transport}}$ and adds to the public skill set. The student can also utilize all the skills in the skill set, both the ones already there and ones newly added, when solving the MDPs. For instance, the student utilizes the two navigation skills, which are already there in the skill set, and $\overline{\pi}^{\mathrm{transport}}$, which is newly added when solving $\mathrm{MDP}^1_{1,n_1+1}$, in order to solve $\mathrm{MDP}^2_{2,n_2+1}$ faster. In this way, all the current learned knowledge does not need to be relearned again, so this whole process is in a fully online fashion.

# D  MULTI-LEVEL COMPRESSION AND ALGORITHMIC REALIZATIONS

## D.1  MULTI-LEVEL COMPRESSION

The results in this section are analytical and allow the effective construction of higher-level transition probabilities, rewards and discount factors from a current level, essentially by only solving linear systems.

### D.1.1  STATEMENT OF RESULTS

Now we derive the closed-form formulas for $P^{l+1}(s, \pi^l, s')$, $R^{l+1}(s, \pi^l, s')$, $\Gamma^{l+1}(s, \pi^l, s')$ given the finer MDP $\mathrm{MDP}^l = (\mathcal{S}, \mathcal{S}^{\mathrm{init}}, \mathcal{S}^{\mathrm{end}}, \mathcal{A}^l, P^l, R^l, \Gamma^l)$ and a policy $\pi^l$ on $\mathcal{S}\mathcal{A}^l$.

To do that, we first bring in the following notations before stating three propositions. Given a policy $\pi^l$, we compute policy-specific Markov transition matrices by averaging out the actions in $\mathcal{A}^l$ according to $\pi^l$:

$$P^{\pi^l}(s, s') := \sum_{a \in \mathcal{A}^l(s)} P^l(s, a, s') \pi^l(s, a), \tag{D.1}$$

and if we restrict the domain in the sum above to $(\mathcal{A}^l)^{\mathbf{end}}$, then we have:

$$P^{\pi^l}_{(\mathcal{A}^l)^{\mathbf{end}}}(s, s') := \sum_{a \in (\mathcal{A}^l)^{\mathbf{end}}} P^l(s, a, s') \pi^l(s, a) = \sum_{a \in (\mathcal{A}^l)^{\mathbf{end}}} \mathbb{1}_{\{s\}}(s') \pi^l(s, a), \tag{D.2}$$

where the second equality comes from the fact that $P^l(s, a, s') = \mathbb{1}_{\{s\}}(s')$ for any $a \in (\mathcal{A}^l)^{\mathbf{end}}$.

For any pair of tensors $X = X(s, a, s')$, $Y(s, a, s')$, indexed by $s, s' \in \mathcal{S}$, $a \in \mathcal{A}^l(s)$, we define the matrix $(X \circ Y)^{\pi^l}$ to be the expectation w.r.t. the extended policy $\pi^l$ of the elementwise (Hadamard) product between $X$ and $Y$:

$$[(X \circ Y)^{\pi^l}]_{s,s'} := \sum_{a \in \mathcal{A}^l(s)} X(s, a, s') Y(s, a, s') \pi^l(s, a).$$

Note that $(X \circ Y)^{\pi^l} = (Y \circ X)^{\pi^l}$.

Now we can state three propositions.

**Proposition D.1.** For the finer MDP $\mathrm{MDP}^l = (\mathcal{S}, \mathcal{S}^{\mathrm{init}}, \mathcal{S}^{\mathrm{end}}, \mathcal{A}^l, P^l, R^l, \Gamma^l)$ and the policy $\pi^l$ on $\mathcal{S}\mathcal{A}^l$, we have $P^{l+1}(s, \pi^l, s') = H_{s,s'}$, for all $s, s' \in \mathcal{S}$, where $H$ is the minimal non-negative solution to the linear system

$$[1 - (1 - \frac{1}{t_{\pi^l}})(P^{\pi^l} - P^{\pi^l}_{(\mathcal{A}^l)^{\mathbf{end}}})]H = (1 - \frac{1}{t_{\pi^l}})P^{\pi^l}_{(\mathcal{A}^l)^{\mathbf{end}}} + \frac{P^{\pi^l}}{t_{\pi^l}}. \tag{D.3}$$

**Proposition D.2.** With the same settings as in Prop. D.1, we have $R^{l+1}(\cdot, \pi^l, s') = h_{s'}$, for all $s' \in \mathcal{S}$, where $h$ is the (unique, bounded) solution, for each $s' \in \mathcal{S}$, to the linear system

$$[1 + (1 - \frac{1}{t_{\pi^l}})(P^{\pi^l}_{(\mathcal{A}^l)^{\mathbf{end}}} - (P^{\pi^l}_{s'} \circ \Gamma^l)^{\pi^l})]h_{s'}$$

$$=(1 - \frac{1}{t_{\pi^l}})[(P^{\pi^l}_{s'} \circ R^l)^{\pi^l}\mathbf{1} - r^l P^{\pi^l}_{(\mathcal{A}^l)^{\mathbf{end}}}\mathbf{1} \tag{D.4}$$

$$+ r^l[P^{l+1}(\cdot, \pi^l, s')]^{-1}_{\text{diag}}P^{\pi^l}_{(\mathcal{A}^l)^{\mathbf{end}}}v_{s'}] + \frac{(\overline{P}^{\pi^l}_{s'} \circ R^l)^{\pi^l}v_{s'}}{t_{\pi^l}} ,$$

where

$$P^{\pi^l}_{s'}(s, a, s'') := \frac{P^l(s, a, s'')P^{l+1}(s'', \pi^l, s')}{P^{l+1}(s, \pi^l, s')} , \quad \text{and} \quad \overline{P}^{\pi^l}_{s'}(s, a, s'') := \frac{P^l(s, a, s'')}{P^{l+1}(s, \pi^l, s')} ,$$

$\mathbf{1}$ is a vector whose $|\{s \in \mathcal{S} : P^{l+1}(s, \pi^l, s') > 0\}|$ coordinates are all ones, $v_{\text{diag}}$ is a diagonal matrix whose diagonal elements are in $v$, $v_s$ is a vector whose $|\mathcal{S}|$ coordinates are all zeros except for the position corresponding to the state $s$, whose value is one.

**Proposition D.3.** With the same settings as in Prop. D.1, we have $\Gamma^{l+1}(\cdot, \pi^l, s') = h_{s'}$, for all $s' \in \mathcal{S}$, where $h$ is the minimal non-negative solution, for each $s' \in \mathcal{S}$, to the linear system

$$[1 + (1 - \frac{1}{t_{\pi^l}})(P^{\pi^l}_{(\mathcal{A}^l)^{\mathbf{end}}} - (P^{\pi^l}_{s'} \circ \Gamma^l)^{\pi^l})]h_{s'}$$

$$=(1 - \frac{1}{t_{\pi^l}})[P^{l+1}(\cdot, \pi^l, s')]^{-1}_{\text{diag}}P^{\pi^l}_{(\mathcal{A}^l)^{\mathbf{end}}}v_{s'} + \frac{(\overline{P}^{\pi^l}_{s'} \circ \Gamma^l)^{\pi^l}v_{s'}}{t_{\pi^l}} . \tag{D.5}$$

### D.1.2 PROOFS

**Proof of Prop. D.1.** First, according to the Markov property, we have $P^{l+1}(s, \pi^l, s') = H_{s,s'}$, for all $s, s' \in \mathcal{S}$, where $H$ is the minimal non-negative solution, for each $s' \in \mathcal{S}$, to the linear system

$$[(1 - \frac{1}{t_{\pi^l}})\sum_{a \in (\mathcal{A}^l)^{\mathbf{end}}} \pi^l(s, a) + 1]H_{s,s'}$$

$$=(1 - \frac{1}{t_{\pi^l}})(\sum_{s'' \in \mathcal{S}} P^{\pi^l}(s, s'')H_{s'',s'}) + (1 - \frac{1}{t_{\pi^l}})\sum_{a \in (\mathcal{A}^l)^{\mathbf{end}}} \pi^l(s, a)\mathbb{1}_{\{s\}}(s') + \frac{P^{\pi^l}(s, s')}{t_{\pi^l}} .$$

Transforming this to matrix-vector form, we have equation D.3.

**Proof of Prop. D.2.** First, according to the Markov property, we have $R^{l+1}(s, \pi^l, s') = H_{s,s'}$, for all $s, s' \in \mathcal{S}$, where $H$ is the (unique, bounded) solution, for each $s' \in \mathcal{S}$, to the linear system

$$[1 + (1 - \frac{1}{t_{\pi^l}})\sum_{a \in (\mathcal{A}^l)^{\mathbf{end}}} \pi^l(s, a)]H_{s,s'}$$

$$=(1 - \frac{1}{t_{\pi^l}})[\sum_{s'' \in \mathcal{S}}\sum_{a \in \mathcal{A}^l(s)} P^{\pi^l}_{s'}(s, a, s'')\Gamma^l(s, a, s'')\pi^l(s, a)H_{s'',s'} + \sum_{s'' \in \mathcal{S}}\sum_{a \in \mathcal{A}^l(s)} P^{\pi^l}_{s'}(s, a, s'')R^l(s, a, s'')\pi^l(s, a)$$

$$- r^l\sum_{a \in (\mathcal{A}^l)^{\mathbf{end}}} \pi^l(s, a) + \frac{r^l}{P^{l+1}(s, \pi^l, s')}\sum_{a \in (\mathcal{A}^l)^{\mathbf{end}}} \pi^l(s, a)\mathbb{1}_{\{s\}}(s')] + \frac{1}{t_{\pi^l}}\sum_{a \in \mathcal{A}^l(s)} \overline{P}^{\pi^l}_{s'}(s, a, s')R^l(s, a, s')\pi^l(s, a) .$$

Transforming this to matrix-vector form, we have equation D.4.

**Proof of Prop. D.3.** First, according to the Markov property, we have $\Gamma^{l+1}(s, \pi^l, s') = H_{s,s'}$, for all $s, s' \in \mathcal{S}$, where $H$ is the minimal non-negative solution, for each $s' \in \mathcal{S}$, to the linear system

$$[1 + (1 - \frac{1}{t_{\pi^l}})\sum_{a \in (\mathcal{A}^l)^{\mathbf{end}}} \pi^l(s, a)]H_{s,s'}$$

$$=(1 - \frac{1}{t_{\pi^l}})[\sum_{s'' \in \mathcal{S}}\sum_{a \in \mathcal{A}^l(s)} P^{\pi^l}_{s'}(s, a, s'')\Gamma^l(s, a, s'')\pi^l(s, a)H_{s'',s'} + \frac{1}{P^{l+1}(s, \pi^l, s')}\sum_{a \in (\mathcal{A}^l)^{\mathbf{end}}} \pi^l(s, a)\mathbb{1}_{\{s\}}(s')]$$

$$+ \frac{1}{t_{\pi^l}}\sum_{a \in \mathcal{A}^l(s)} \overline{P}^{\pi^l}_{s'}(s, a, s')\Gamma^l(s, a, s')\pi^l(s, a) .$$

Transforming this to matrix-vector form, we have equation D.5.

## D.2 ALGORITHMIC REALIZATION WITHOUT TRANSFER LEARNING

With the multi-level compression provided in detail in the previous App. D.1, we are ready to discuss the algorithmic realization, first without transfer learning. We have the following Alg. 1 as well as the auxiliary Algs. 2–4. The inputs of Alg. 1 are mostly as discussed in Sec. 2.1. There are two exceptions: (1) for simplification, the teacher does not provide the sequence of finite partial policy generator sets $\{\mathcal{G}_{\text{test}}^l\}_{l=1}^{\infty}$, and provides instead the difficulty $L$ of MDP directly, which is set as an extra property of MDP. Consequently, the teacher only needs to provide $\{\mathcal{G}^l\}_{l=1}^{L'-1}$, the first $L'-1$ elements of the infinite sequence $\{\mathcal{G}^l\}_{l=1}^{\infty}$, with $L' = L$ for this section. (2) The timescales of policies could be reduced to the timescales of partial policies and then further reduced to partial policy generators by assuming: $(i)$, the timescale of a policy equals the minimum of the timescales of partial policies producing it using outer product as in equation 2.1; $(ii)$ the timescale of a partial policy equals the timescale of the partial policy generator generating it. In this way, the timescales of partial policy generators are provided instead, which are set as an extra property of partial policy generators. A lower bound and an upper bound of such timescales $t_{\text{bounds}}. \min, t_{\text{bounds}}. \max$ are also provided.

The algorithm consists of two main parts: constructing the MMDP following the recursive definition in Sec. 2.1 and solving each MDP in it from the top most compressed MDP down til the bottom finest original MDP.

For completeness, we also add the error detection mechanism in Alg. 1 according to the criteria $\{\text{thresh}^l\}_{l=1}^{L'} = \{(N_{\max}^l, T_{\max}^l, v_{\min}^l)\}_{l=1}^{L'}$, where for $l$-th level MDP $\text{MDP}^l$, $N_{\max}^l$ is the upper bound on the number of allowed iterations before convergence; $T_{\max}^l$ is the maximum number of steps per episode upon convergence, averaged across episodes upon convergence from possibly multiple initial states, and $T_{\max}^l$ is set to $+\infty$ for non-episodic $\text{MDP}^l$ by default; $v_{\min}^l$ is the lower bound for the value of initial states, possibly averaged across multiple initial states. Meanwhile, we record the corresponding actual values as $\{\text{stats}^l\} = \{(N^l, T^l, v^l)\}$ when the student solves $\text{MDP}^l$ for $l \in [L']$. If for some $l \in [L']$, one of the actual values are out of the three thresholds in $\text{thresh}^l$, then the output error message err is $\{\text{exist}: 1, \text{level}: l\}$, indicating that there is an error occurring at level $l$, so the algorithm stops there. Otherwise, all levels of MDPs are solved successfully, and the output error message err is $\{\text{exist}: 0, \text{level}: 1\}$, indicating that there is no error across all levels and the algorithm finishes on level one.

## D.3 ALGORITHMIC REALIZATION WITH TRANSFER LEARNING

Now we incorporate transfer learning into the algorithmic realization discussed in the previous App. D.2. After incorporating transfer learning into Alg. 1, we have the following Algs.5–6. See Sec. 4 for the description of the algorithms.

## D.4 EXPLANATION ON THE CONSISTENCY OF OUR ALGORITHM

One main reason behind consistency of our approach is that even in non-benign regimes, where we may encounter scenarios in which the policy learned at a higher-level leads to a bad initialization when pushed down to the next finer level, and therefore at the next finer level we need significant refinement in order to obtain the optimal policy. Even in such a scenario, the MMDP solver could still converge to the optimal policy of the original MDP as long as the MDP solver applied to the next finer level, started from this bad initialization, converges to the optimal policy as the number of iterations goes to infinity.

**Input:** MDP: the original MDP with difficulty $L$; $\{\mathcal{G}^l\}_{l=1}^{L'-1}$: the first $L'-1$ elements of sequence of generator sets; $t_{\text{bounds}}$: bounds for the timescale of any policy; $\{r^l\}_{l=1}^{L'}$: negative rewards on choosing actions in $(\mathcal{A}^l)^{\mathbf{end}}$ for $l \in [L']$; solver_init$^{L'}$: initial value function and policy for the most compressed MDP MDP$^{L'}$; $\{$solver_end$^l\}_{l=1}^{L'}$: stopping criteria for solver of MDP$^l$ for $l \in [L']$, possibly including error tolerance of solved value function or maximum number of iterations; $\{$thresh$^l\}_{l=1}^{L'}$: thresholds for error detection on MDP$^l$ for $l \in [L']$.

  1  **Initialize:** MDP$^1$ = MDP, solution$^l$ = NA for $l \in [L']$, stats$^l$ = NA for $l \in [L']$
  2  **for** $l = 1, 2, \cdots, L'-1$ **do**
  3      $\mathcal{A}^{l+1}$ = generate(MDP$^l$, $\mathcal{G}^l$)
  4      MDP$^{l+1}$ = compress(MDP$^l$, $\mathcal{A}^{l+1}$, $r^{l+1}$)
  5  **end for**
  6  **for** $l = L', L'-1, \cdots, 1$ **do**
  7      (solution$^l$, stats$^l$, err$^l$) = Solve_MDP(MDP$^l$, $t_{\text{bounds}}$, solver_init$^l$, solver_end$^l$, thresh$^l$)
  8      **if** err$^l$.exist = 1 **then**
  9          **return** $(\{$MDP$^{l'}\}_{l'=1}^{L'}, \{$solution$^{l'}\}_{l'=1}^{L'}, \{$stats$^{l'}\}_{l'=1}^{L'}, \{$exist : 1, level : $l\})$
 10      **end if**
 11      **if** $l > 1$ **then**
 12         Compute solver_init$^{l-1}$ from solution$^l$ and $\mathcal{G}^{l-1}$ using equation B.7
 13      **end if**
 14  **end for**
 15  **return** $(\{$MDP$^l\}_{l=1}^{L'}, \{$solution$^l\}_{l=1}^{L'}, \{$stats$^l\}_{l=1}^{L'}, \{$exist : 0, level : 1$\})$

**Output:** $\{$MDP$^l\}_{l=1}^{L'}, \{$solution$^l\}_{l=1}^{L'}, \{$stats$^l\}_{l=1}^{L'}$: the MMDP, the solutions of the MDPs in it, and summary of statistics when solving the MDPs; err: error information.

Algorithm 1: Solve an MDP from top to bottom: $(\{$MDP$^l\}_{l=1}^{L'}, \{$solution$^l\}_{l=1}^{L'}, \{$stats$^l\}_{l=1}^{L'}$, err$) = $ solve_MMDP(MDP, $\{\mathcal{G}^l\}_{l=1}^{L'-1}, t_{\text{bounds}}, \{r^l\}_{l=1}^{L'}$, solver_init$^{L'}, \{$solver_end$^l\}_{l=1}^{L'}, \{$thresh$^l\}_{l=1}^{L'})$

**Input:** MDP: the MDP; $\mathcal{G}$: the partial policy generator set, with domains of generators in it being $\Theta_1, \Theta_2, \cdots, \Theta_M$.

  1  **if** $\mathcal{G} = \varnothing$ **then**
  2      $\mathcal{G}$ = MDP.$\mathcal{A}$
  3  **end if**
  4  partitions_list = list_all_partitions(MDP.$\mathcal{A}$, $\mathcal{G}$)
  5  $\Pi = \varnothing$
  6  **for** each partition $\in$ partitions_list **do**
  7      Generate policies $\Pi_{\text{new}}$ using outer product as in equation 2.1
  8      Set timescales of policies in $\Pi_{\text{new}}$ as $\min_{g|_I \in \text{partition}}\{g|_I.\text{timescale}\}$
  9      Add $\Pi_{\text{new}}$ to $\Pi$
 10  **end for**
 11  $\widetilde{\mathcal{A}} = \Pi \cup \left(\Pi_{m=1}^M(\Theta_m \cup a^{\text{end}}) - \Pi_{m=1}^M \Theta_m\right)$
**Output:** $\widetilde{\mathcal{A}}$: the action set.

Algorithm 2: Generate the action set $\widetilde{\mathcal{A}}$ = generate(MDP, $\mathcal{G}$)

**Input:** MDP: the finer MDP; $\widetilde{\mathcal{A}}$: the action set for the compressed MDP; $\widetilde{r}$: negative reward for choosing actions in $(\widetilde{\mathcal{A}})^{\mathbf{end}}$ within the compressed MDP.

1  $\widetilde{\mathrm{MDP}}.\mathcal{S} = \mathrm{MDP}.\mathcal{S}$
2  $\widetilde{\mathrm{MDP}}.\mathcal{S}^{\mathrm{init}} = \mathrm{MDP}.\mathcal{S}^{\mathrm{init}}$
3  $\widetilde{\mathrm{MDP}}.\mathcal{S}^{\mathrm{end}} = \mathrm{MDP}.\mathcal{S}^{\mathrm{end}}$
4  $\widetilde{\mathrm{MDP}}.\mathcal{A} = \widetilde{\mathcal{A}}$
5  $\widetilde{\mathrm{MDP}}.\mathcal{SA} = \widetilde{\mathrm{MDP}}.\mathcal{S} \times \widetilde{\mathcal{A}}$
6  **for** each $\pi \in \widetilde{\mathcal{A}} - (\widetilde{\mathcal{A}})^{\mathbf{end}}$ **do**
7      Compute matrix $P^\pi, P^\pi_{\mathcal{A}^{\mathbf{end}}}$ using equation D.1, equation D.2
8      Solve for matrix $\widetilde{\mathrm{MDP}}.P(\cdot, \pi, \cdot)$ using equation D.3
9      **for** each $s' \in \mathcal{S}$ **do**
10         Solve for vector $\widetilde{\mathrm{MDP}}.R(\cdot, \pi, s')$ using equation D.4, with $r^l = \mathrm{MDP}.R(\cdot, a, \cdot)$ for any $a \in \mathcal{A}^{\mathbf{end}}$
11         Solve for vector $\widetilde{\mathrm{MDP}}.\Gamma(\cdot, \pi, s')$ using equation D.5
12      **end for**
13  **end for**
14  Set $P(s, a, s') = \mathbb{1}_{\{s\}}(s'), R(s, a, s') = \widetilde{r}, \Gamma(s, a, s') = 1$ for **any** $a \in (\widetilde{\mathcal{A}})^{\mathbf{end}}$
**Output:** $\widetilde{\mathrm{MDP}}$: the compressed MDP.

Algorithm 3: Compress an MDP $\widetilde{\mathrm{MDP}} = \mathtt{compress}(\mathrm{MDP}, \widetilde{\mathcal{A}}, \widetilde{r})$

**Input:** MDP: the MDP to be solved; $t_{\mathrm{bounds}}$: bounds for the timescale of any policy; solver_init: initial value function and policy for MDP; solver_end: stopping criteria for solver of MDP; thresh: thresholds for error detection on MDP.

1  **Initialize:** $N = 0, \mathrm{solution} = \mathrm{solver\_init}, \mathrm{solution}_{\mathrm{prev}} = \mathrm{solver\_init}, \mathrm{err.exist} = 1$
2  **while** $0 < N < \mathrm{solver\_end}.N_{\mathrm{max}}$ **and** $\|\mathrm{solution}.V - \mathrm{solution}_{\mathrm{prev}}.V\|_\infty > \mathrm{solver\_end}.\epsilon$ **do**
3      $\mathrm{solution}_{\mathrm{prev}} = \mathrm{solution}$
4      $\mathrm{solution} = \mathtt{MDP\_solve\_update}(\mathrm{MDP}, \mathrm{solution})$
5      $N = N + 1$
6  **end while**
7  **Update** the distributions of $\mathrm{solution}.\pi$ at $\mathrm{MDP}.\mathcal{S}^{\mathrm{end}}$ to be $\mathbb{1}_{\{a^{\mathbf{end}}\}}(\cdot)$
8  $\mathrm{stats} = \{N : N, T : +\infty, v : \frac{\sum_{s \in \mathcal{S}^{\mathrm{init}}}(\mathrm{solution}.V)(s)}{|\mathcal{S}^{\mathrm{init}}|}\}$
9  **if** $N \leq \mathrm{thresh}.N_{\mathrm{max}}$ **and** $\|(\mathrm{solution}.V - \mathrm{solution}_{\mathrm{prev}}.V)\mid_{\mathcal{S}^{\mathrm{init}}}\|_\infty \leq \mathrm{solver\_end}.\epsilon$ **and** $\mathrm{stats}.V \geq \mathrm{thresh}.v_{\mathrm{min}}$ **and** $t_{\mathrm{bounds}}.\mathrm{max} > t_{\mathrm{bounds}}.\mathrm{min}$ **then**
10     $((\mathrm{solution}.\pi).T, \mathrm{err}) = \mathtt{compute\_timescale}(\mathrm{MDP}, \mathrm{solution}.\pi, t_{\mathrm{bounds}}, \mathrm{thresh}.T_{\mathrm{max}})$
11     $\mathrm{stats}.T = (\mathrm{solution}.\pi).T$
12  **end if**
13  **if** err.exist $= 1$ **then**
14     $\mathrm{solution} = \mathrm{NA}$
15  **end if**
**Output:** solution: the solution of MDP, possibly containing a learned policy and a value function associated to a learned policy; stats: summary of statistics when solving MDP; err: error information.

Algorithm 4: Solve an MDP $(\mathrm{solution}, \mathrm{stats}, \mathrm{err}) =$
$\mathtt{Solve\_MDP}(\mathrm{MDP}, t_{\mathrm{bounds}}, \mathrm{solver\_init}, \mathrm{solver\_end}, \mathrm{thresh})$

**Input:** $\{\text{MDP}_{L,n}\}_{L=1,n=1}^{L_{\max},n_L}$, $\{\text{hint}_{L,n}\}_{L=1,n=1}^{L_{\max},n_L}$: a series of MDPs ordered by difficulty as well as correspondingly a series of hints provided by the teacher with detailed descriptions as given in Alg. 6; $t_{\text{bounds}}$: bounds for the timescale of any policy and skill; $\{\textbf{solver\_end}_{L,n}\}_{L=1,n=1}^{L_{\max},n_L}$: a series of stopping criteria for solvers of MDPs in the MMDP constructed from $\text{MDP}_{L,n}$ for $L \in [L_{\max}], n \in [n_L]$; $\{\textbf{thresh}_{L,n}\}_{L=1,n=1}^{L_{\max},n_L}$: a series of thresholds for error detection on MDPs in the MMDP constructed from $\text{MDP}_{L,n}$ for $L \in [L_{\max}], n \in [n_L]$.

1    **Initialize:** $Skills = \{\text{id}\}$
2    **for** $L = 1, 2, \cdots, L_{\max}$ **do**
3      **for** $n = 1, 2, \cdots, n_L$ **do**
4        $(\textbf{solution}_{L,n}, \text{err}_{L,n})$
         $= \texttt{learn\_MDP}(\text{MDP}_{L,n}, \text{hint}_{L,n}, t_{\text{bounds}}, \textbf{solver\_end}_{L,n}, \textbf{thresh}_{L,n})$
5      **end for**
6    **end for**

**Output:** $\{\textbf{solution}_{L,n}\}_{L=1,n=1}^{L_{\max},n_L}$, $\{\text{err}_{L,n}\}_{L=1,n=1}^{L_{\max},n_L}$: the solutions of the MDPs in the series of MMDPs constructed from $\{\text{MDP}_{L,n}\}_{L=1,n=1}^{L_{\max},n_L}$ and the series of error information.

Algorithm 5: Learn a curriculum $(\{\textbf{solution}_{L,n}\}_{L=1,n=1}^{L_{\max},n_L}, \{\text{err}_{L,n}\}_{L=1,n=1}^{L_{\max},n_L}) = \texttt{learn\_curriculum}$
$(\{\text{MDP}_{L,n}\}_{L=1,n=1}^{L_{\max},n_L}, \{\text{hint}_{L,n}\}_{L=1,n=1}^{L_{\max},n_L}, t_{\text{bounds}}, \{\textbf{solver\_end}_{L,n}\}_{L=1,n=1}^{L_{\max},n_L}, \{\textbf{thresh}_{L,n}\}_{L=1,n=1}^{L_{\max},n_L})$

---

**Input:** MDP: the original MDP with difficulty $L$ and number $n$; hint: hints provided by the teacher with five fields: the first field $\mathcal{G}$ of hint helps the student derive action sets in compressed MDPs, which contains a sequence with length $L'-1$ of skill-embedding generator pair sets; a second field $r$ contains a sequence with length $L'$ of negative rewards for choosing actions in $(\mathcal{A}^l)^{\textbf{end}}$ for $l \in [L']$; a third field skill $\overline{\pi}$ and a fourth field embedding $e_{\text{comp}}$ help the student compose the initial policy for the most compressed MDP $\text{MDP}^{L'}$; the last field $e_{\text{decomp}}$ helps the assistant extract out new skills from the optimal policies for the MDPs, which contains a sequence with length $L'$ of embeddings; $t_{\text{bounds}}$: bounds for the timescale of any skill; $\{\text{solver\_end}^l\}_{l=1}^{L'}$: stopping criteria for solver of $\text{MDP}^l$ for $l \in [L']$; $\{\text{thresh}^l\}_{l=1}^{L'}$: thresholds for error detection on $\text{MDP}^l$ for $l \in [L']$.

1    **Initialize:** $\text{MDP}^1 = \text{MDP}$
2    **for** $l = 1, 2, \cdots, L'-1$ **do**
3      $\mathcal{G}^l = \varnothing$
4      **for** each $(\overline{g}, E) \in \text{hint}.\mathcal{G}(l)$ **do**
5        **if** $\overline{g} \neq \text{NA}$ **then**
6          $\mathcal{G}^l = \mathcal{G}^l \cup \texttt{compose}((\overline{g}, E), \text{MDP}^l.\mathcal{SA})$
7        **end if**
8      **end for**
9      $\mathcal{A}^{l+1} = \texttt{generate}(\text{MDP}^l, \mathcal{G}^l)$
10      $\text{MDP}^{l+1}.\mathcal{SA} = \text{MDP}.\mathcal{S} \times \mathcal{A}^{l+1}$, $\text{MDP}^{l+1}.\text{difficulty} = L - l$
11    **end for**
12    **if** $\text{hint}.\overline{\pi} \neq \text{NA}$ **and** $\text{hint}.e_{\text{comp}} \neq \text{NA}$ **then**
13      $\text{solver\_init}.\pi = \texttt{compose}(\text{hint}.\overline{\pi}, \text{hint}.e_{\text{comp}}, \text{MDP}^{L'}.\mathcal{SA})$
14    **else**
15      **Set** $\text{solver\_init}.\pi$ to be the diffusive policy with uniform distribution at each state
16    **end if**
17    $(\{\text{solution}\}_{l=1}^{L'}, \sim, \sim, \text{err})$
     $= \texttt{solve\_MMDP}(\text{MDP}, \{\mathcal{G}^l\}_{l=1}^{L'-1}, \text{hint}.r, \text{solver\_init}, \{\text{solver\_end}^l\}_{l=1}^{L'}, \{\text{thresh}^l\}_{l=1}^{L'})$
18    **for** $l = 1, 2, \cdots, L'$ **do**
19      **if** $(\text{hint}.e_{\text{decomp}})(l) \neq \text{NA}$ **and** $\text{solution}^l \neq \text{NA}$ **then**
20        $Skills = Skills \cup \{\texttt{decompose}(\text{solution}^l.\pi, (\text{hint}.e_{\text{decomp}})(l), t_{\text{bounds}})\}$
21      **end if**
22    **end for**

**Output:** the solutions $\{\text{solution}\}_{l=1}^{L'}$ of the MDPs in MMDP constructed from original MDP and the error information err.

Algorithm 6: Learn an MDP $(\{\text{solution}\}_{l=1}^{L'}, \text{err}) = \texttt{learn\_MDP}(\text{MDP}, \text{hint}, t_{\text{bounds}}, \{\text{solver\_end}^l\}_{l=1}^{L'}, \{\text{thresh}^l\}_{l=1}^{L'})$

# E  ANALYTICAL REALIZATION OF ALGORITHMS APPLIED TO OUR EXAMPLES

We conclude the appendix with the analytical realization of Algs.5–6 detailed in App. D.3 applied to our examples. We organize this section in the following way: for each example, we first provide inputs (problem settings and MDP definitions etc.) and how different objects (policies, skills, embeddings, etc.) should be constructed when using our algorithms detailed in App. D.3, then in the last section of each example, we describe how the algorithms construct and discover these objects as the algorithms progress, demonstrating their correctness.

## E.1  MAZEBASE+

### E.1.1  GEOMETRIC CONFIGURATION AND OBJECT STATES

We start with the geometric configuration of this example. With notations as in the previous example of navigation and transportation with traffic jams, we have a two-dimensional grid world $\Omega = [x_1, x_2] \times [y_1, y_2] \subseteq \mathbb{N} \times \mathbb{N}$ and a set of actions in different directions, $\mathcal{A}_{\mathtt{dir}} = \{(1,0), (0,1), (-1,0), (0,-1)\}$. This grid world contains some objects, including blocks, three doors, three keys, and a goal. To show the effects of transfer learning in Box 3, Fig. 2, we assume the initial locations of all the objects (including the blocks, keys, doors, and the goal) are fixed to certain locations. All the state spaces of the MDPs in Box 3 and Box 13 also satisfy the constraints given here unless specified:

- $s_{\mathtt{door}_1}, s_{\mathtt{door}_2}, s_{\mathtt{door}_3} \in \Omega$, the locations of $\mathtt{door}_1$, $\mathtt{door}_2$, and $\mathtt{door}_3$ respectively, satisfy the following conditions: $s_{\mathtt{door}_2}(1) < s_{\mathtt{door}_1}(1) = s_{\mathtt{door}_3}(1), s_{\mathtt{door}_1}(2) < s_{\mathtt{door}_2}(2) < s_{\mathtt{door}_3}(2)$, and $s_{\mathtt{door}_1}(1) \in \{x_1, x_1 + 3, \cdots\}, s_{\mathtt{door}_2}(2) \in \{y_1, y_1 + 3, \cdots\}$ (e.g., we may take $s_{\mathtt{door}_1}(1) = 10, s_{\mathtt{door}_2}(2) = 4$).
- the set of locations of the blocks $\Omega_{\mathtt{blocks}}$ is given by: $\Omega_{\mathtt{blocks}} = [x_1, x_2] \times \{s_{\mathtt{door}_2}(2)\} \cup \{s_{\mathtt{door}_1}(1)\} \times [y_1, y_2] - \{s_{\mathtt{door}_1}, s_{\mathtt{door}_2}, s_{\mathtt{door}_3}\}$.
- the blocks together with the doors separate the remaining grid points $\Omega - \Omega_{\mathtt{blocks}} - \cup_{i=1}^3 s_{\mathtt{door}_i}$ into four rooms: $\mathtt{room}_1$, $\mathtt{room}_2$, $\cdots$, $\mathtt{room}_4$, with corresponding regions $\Omega_{\mathtt{room}_1} := \{\omega \in \Omega : \omega(1) < s_{\mathtt{door}_1}(1), \omega(2) < s_{\mathtt{door}_2}(2)\}, \Omega_{\mathtt{room}_2} := \{\omega \in \Omega : \omega(1) > s_{\mathtt{door}_1}(1), \omega(2) < s_{\mathtt{door}_2}(2)\}, \Omega_{\mathtt{room}_3} := \{\omega \in \Omega : \omega(1) < s_{\mathtt{door}_1}(1), \omega(2) > s_{\mathtt{door}_2}(2)\}, \Omega_{\mathtt{room}_4} := \{\omega \in \Omega : \omega(1) > s_{\mathtt{door}_1}(1), \omega(2) > s_{\mathtt{door}_2}(2)\}$.
- $s_{\mathtt{cur}} \in \Omega_{\mathtt{blocks}}^c := \Omega - \Omega_{\mathtt{blocks}}$ is the current location of the agent, and the agent could not be at the location of a door unless the door is open.
- $s_{\mathtt{key}_1}, s_{\mathtt{key}_2}, s_{\mathtt{key}_3} \in \Omega_{\mathtt{blocks}}^c$ are the locations of $\mathtt{key}_1$, $\mathtt{key}_2$, and $\mathtt{key}_3$ respectively, and $\mathtt{key}_i$ could only open $\mathtt{door}_i$ for $i = 1, 2, 3$. $s_{\mathtt{goal}} \in \Omega_{\mathtt{blocks}}^c - \cup_{i=1}^3 s_{\mathtt{door}_i}$ is the location of the goal object.

We would like to comment here that our method actually does allow the locations of the keys and goals to change within the above constraints. To be more precise, with another new embedding to extract a more general skill following our framework, with the skill further abstracting out the logic of navigation within a single room while avoiding obstacles along the way (without assuming much on the geometry of the grid world/blocks), even the geometry of the grid world or the locations of blocks could be changed completely, the only necessary and sufficient conditions is that all the rooms are "geodesically complete" in the sense that each contains at least one shortest path between any two points, so that the agent could at least find one shortest path in the usual sense. We consider here a simpler setting which suffices to convey our message.

For $i \in [3]$, the state variable $s_{\mathtt{open}_i} \in \{0, 1\}$ indicates whether $\mathtt{door}_i$ is open or not respectively (1=open); the state variable $s_{\mathtt{pick}_i}$ indicates whether the agent has $\mathtt{key}_i$ (1=the agent has $\mathtt{key}_i$); the state variable $s_{\mathtt{done}}$ indicates whether the agent has in hand the goal object (1=the agent has $\mathtt{goal}$). In particular, if $s_{\mathtt{pick}_i} = 1$, then $s_{\mathtt{key}_i} = s_{\mathtt{cur}}$ and if $s_{\mathtt{pick}_i} = 1$, then $s_{\mathtt{goal}} = s_{\mathtt{cur}}$. Without losing generality, for objects in $\{\mathtt{key}_1, \mathtt{key}_2, \mathtt{key}_3, \mathtt{goal}\}$ not initially picked up by the agent, we assume initially $s_{\mathtt{key}_1}, s_{\mathtt{key}_2} \in \Omega_{\mathtt{room}_1}, s_{\mathtt{key}_3} \in \Omega_{\mathtt{room}_3}$, and $s_{\mathtt{goal}} \in \Omega_{\mathtt{room}_4}$.

For $i \in [3]$, the agent can open $\mathtt{door}_i$ by selecting the action $\mathtt{open}$ if the agent has in hand $\mathtt{key}_i$ and the agent is next to $\mathtt{door}_i$, meaning that $s_{\mathtt{cur}} \in \mathcal{N}(s_{\mathtt{door}_i}) := \{s : ||s - s_{\mathtt{door}_i}||_1 = 1\}$; the agent can pick up $\mathtt{key}_i$ by selecting the action $\mathtt{pick}$ if the agent is at the location of $\mathtt{key}_i$. The

agent can pick up the goal by selecting the action `pick` if the agent is at the location of the goal; the agent is allowed to pick up multiple objects in $\{\texttt{key}_1, \texttt{key}_2, \texttt{key}_3, \texttt{goal}\}$ in a single time step.

For ease of notation, we let $s_{\textbf{key}} := (s_{\text{key}_1}, s_{\text{key}_2}, s_{\text{key}_3})$, $s_{\textbf{door}} := (s_{\text{door}_1}, s_{\text{door}_2}, s_{\text{door}_3})$, $s_{\textbf{pick}} := (s_{\text{pick}_1}, s_{\text{pick}_2}, s_{\text{pick}_3})$, and $s_{\textbf{open}} := (s_{\text{open}_1}, s_{\text{open}_2}, s_{\text{open}_3})$. We also denote $\Omega_{\text{room}} := \Omega_{\text{room}_1}, s_{\text{key}} := s_{\text{key}_1}, s_{\text{door}} := s_{\text{door}_1}, s_{\text{pick}} := s_{\text{pick}_1}, s_{\text{open}} := s_{\text{open}_1}$. Besides, in the definition of $\text{MDP}_{3,1}$, we let $\Omega(s)$ denote the set of the agent's all the possible locations given the objects' locations and states determined by the current state $s$, i.e., for $s = (s_{\text{cur}}, s_{\textbf{pick}}, s_{\textbf{open}}, s_{\text{done}}) \in \mathcal{S}_{3,1}$, $\Omega(s) := \Omega_{\text{blocks}}^c \setminus \{s_{\text{door}_i} : s_{\text{open}_i} = 0, 1 \leq i \leq 3\}$. Consequently, we add the restriction that $s_{\text{cur}} \in \Omega(s)$, meaning that if a door is closed, then the agent could not be at the location of that door (nor of a block, of course). As a summary, Table 1 lists all the parameters values in the highest-difficulty MDPs throughout this example. For the rest MDPs in this section, $\text{MDP}_{1,1}$ is exactly the same as $\text{MDP}_{\text{dense}}^{\text{nav}}$ (including applicable parameter values) in the example of navigation and transportation with traffic jams; all the other MDPs use the same parameter values as in the corresponding highest-difficulty MDP: for instance, $\text{MDP}_{2,1}$ and $\text{MDP}_{2,2}$ use the same parameter values as in $\text{MDP}_{3,1}$. Fig. 1 plots the geometric configurations of the grid world and the objects in it for these MDPs described in Box 1 and detailed in App. E.1.2.

Finally, recall that in the definition of an MDP, the set of initial states of the agent, $\mathcal{S}^{\text{init}}$, is set to be the set of states starting from which the agent can reach the goal; in this example starting the agent in $\mathcal{S}^{\text{init}}$ means that the locations of the agent and keys, and the status of the doors are such that the puzzle is indeed solvable.

Table 1: Parameters in the MazeBase+ example

| | $x_1$ | $x_2$ | $x_3$ | $y_1$ | $y_2$ | $y_3$ | $s_{\text{key}_1}$ | $s_{\text{key}_2}$ | $s_{\text{key}_3}$ | $s_{\text{door}_1}$ | $s_{\text{door}_2}$ | $s_{\text{door}_3}$ | $s_{\text{goal}}$ | $p_s$ | $R_0$ | $r_0$ |
|---|---|---|---|---|---|---|---|---|---|---|---|---|---|---|---|---|
| $\text{MDP}_{3,1}$ | 1 | 15 | 10 | 1 | 8 | 4 | $(1,3)$ | $(1,1)$ | $(1,8)$ | $(10,2)$ | $(9,4)$ | $(10,5)$ | $(15,8)$ | 0.9 | $10^4$ | $-10$ |
| $\text{MDP}_{3,1}'$ | 1 | 15 | 10 | 1 | 8 | 4 | $(1,1)$ | $(15,1)$ | $(15,8)$ | $(10,3)$ | $(11,4)$ | $(10,5)$ | $(1,8)$ | 0.9 | $10^4$ | $-10$ |
| $\text{MDP}_{3,1}''$ | 1 | 15 | 10 | 1 | 8 | 4 | $(1,3)$ | $(1,1)$ | $(2,1)$ | $(10,2)$ | $(9,4)$ | $(10,5)$ | $(15,8)$ | 0.9 | $10^4$ | $-10$ |

### E.1.2 FORMAL DEFINITION OF MDPS

The formal definitions of these MDPs are as follows, with the definition of $\text{MDP}_{1,1}$, which is exactly $\text{MDP}_{\text{dense}}^{\text{nav}}$ in the example of navigation and transportation with traffic jams, postponed to the forthcoming App. E.2.1 when we introduction the example of navigation and transportation with traffic jams: $\text{MDP}_{2,1} := (\mathcal{S}_{2,1}, \mathcal{S}_{2,1}^{\text{init}}, \mathcal{S}_{2,1}^{\text{end}}, \mathcal{A}_{2,1}, P_{2,1}, R_{2,1}, \Gamma_{2,1})$ models navigation through $\Omega$ while avoiding blocks assuming all the doors are open:

$$\mathcal{S}_{2,1} := \{(s_{\text{cur}}, s_{\text{dest}}) : s_{\text{cur}}, s_{\text{dest}} \in \Omega_{\text{blocks}}^c\} = \Omega_{\text{blocks}}^c \times \Omega_{\text{blocks}}^c,$$

$$\mathcal{S}_{2,1}^{\text{init}} := \{(s_{\text{cur}}, s_{\text{dest}}) \in \mathcal{S}_{1,1} : s_{\text{cur}} \neq s_{\text{dest}}\},$$

$$\mathcal{S}_{2,1}^{\text{end}} := \{(s_{\text{cur}}, s_{\text{dest}}) \in \mathcal{S}_{1,1} : s_{\text{cur}} = s_{\text{dest}}\},$$

$$\mathcal{A}_{2,1} := \mathcal{A}_{\text{dir}} \cup \{a^{\text{end}}\},$$

$$P_{2,1}((s_{\text{cur}}, s_{\text{dest}}), a, (s_{\text{cur}}', s_{\text{dest}}')) \tag{E.1}$$

$$:= \mathbb{1}_{\{s_{\text{dest}}\}}(s_{\text{dest}}') \times \big[[1 - \mathbb{1}_{\Omega_{\text{blocks}}^c}(s_{\text{cur}} + a)] \times \mathbb{1}_{\{s_{\text{cur}}\}}(s_{\text{cur}}')$$

$$+ \mathbb{1}_{\Omega_{\text{blocks}}^c}(s_{\text{cur}} + a) \times [p_s \times \mathbb{1}_{\{s_{\text{cur}}+a\}}(s_{\text{cur}}') + (1 - p_s) \times \mathbb{1}_{\{s_{\text{cur}}\}}(s_{\text{cur}}')]\big],$$

$$R_{2,1}((s_{\text{cur}}, s_{\text{dest}}), a, (s_{\text{cur}}', s_{\text{dest}}')) := R_0 \times \mathbb{1}_{\{s_{\text{dest}}'\}}(s_{\text{cur}}') + r_0 \times [1 - \mathbb{1}_{\{s_{\text{dest}}'\}}(s_{\text{cur}}')],$$

$$\Gamma_{2,1}((s_{\text{cur}}, s_{\text{dest}}), a, (s_{\text{cur}}', s_{\text{dest}}')) := 1.$$

In the above, the teacher sets the following quantities: $0 < p_s \leq 1$ (set here to 0.9), the probability for an action to succeed in the intended movement; $R_0 > 0$ (set here to 10000), a large positive reward; $r_0 < 0$ (set here to $-10$), a small negative reward. For $\text{MDP}_{2,2} :=$

$(\mathcal{S}_{2,2}, \mathcal{S}_{2,2}^{\text{init}}, \mathcal{S}_{2,2}^{\text{end}}, \mathcal{A}_{2,2}, P_{2,2}, R_{2,2}, \Gamma_{2,2})$ related to picking up a key and opening the door, we have

$$\mathcal{S}_{2,2} := \{(s_{\text{cur}}, s_{\text{pick}}, s_{\text{open}}) : s_{\text{cur}} \in \Omega_{\text{room}}, s_{\text{pick}}, s_{\text{open}} \in \{0,1\}\} = \Omega_{\text{room}} \times \{0,1\}^2 \quad,$$

$$\mathcal{S}_{2,2}^{\text{init}} := \{(s_{\text{cur}}, s_{\text{pick}}, s_{\text{open}}) \in \mathcal{S}_{2,2} : s_{\text{open}} = 0\} \quad, \quad \mathcal{S}_{2,2}^{\text{end}} := \{(s_{\text{cur}}, s_{\text{pick}}, s_{\text{open}}) \in \mathcal{S}_{2,2} : s_{\text{open}} = 1\} \quad,$$

$$\mathcal{A}_{2,2} := \mathcal{A}_{\text{dir}} \cup \{\text{pick}, \text{open}, a^{\text{end}}\} \quad,$$

$$P_{2,2}((s_{\text{cur}}, s_{\text{pick}}, s_{\text{open}}), a, (s'_{\text{cur}}, s'_{\text{pick}}, s'_{\text{open}}))$$

$$:= \begin{cases} \mathbb{1}_{\{s_{\text{pick}}\}}(s'_{\text{pick}}) \times \mathbb{1}_{\{s_{\text{open}}\}}(s'_{\text{open}}) \times \{p_s \times \mathbb{1}_{\Omega_{\text{room}}}(s_{\text{cur}} + a) \times \mathbb{1}_{\{s_{\text{cur}}+a\}}(s'_{\text{cur}}) \\ + [1 - p_s \times \mathbb{1}_{\Omega_{\text{room}}}(s_{\text{cur}} + a)] \times \mathbb{1}_{\{s_{\text{cur}}\}}(s'_{\text{cur}})\} & , \text{if } a \in \mathcal{A}_{\text{dir}} \\ \\ \mathbb{1}_{\{s_{\text{cur}}\}}(s'_{\text{cur}}) \times \{\mathbb{1}_{\{\text{open}\}}(a) \times \mathbb{1}_{\{s_{\text{pick}}\}}(s'_{\text{pick}}) \\ \times [p_s \times \mathbb{1}_{\mathcal{N}(s_{\text{door}})}(s_{\text{cur}}) \times \mathbb{1}_{\{1\}}(s_{\text{pick}}) \times \mathbb{1}_{\{1\}}(s'_{\text{open}}) \\ + [1 - p_s \times \mathbb{1}_{\mathcal{N}(s_{\text{door}})}(s_{\text{cur}}) \times \mathbb{1}_{\{1\}}(s_{\text{pick}})] \times \mathbb{1}_{\{s_{\text{open}}\}}(s'_{\text{open}})] \\ + \mathbb{1}_{\{\text{pick}\}}(a) \times \mathbb{1}_{\{s_{\text{open}}\}}(s'_{\text{open}}) \times [p_s \times \mathbb{1}_{\{s_{\text{key}}\}}(s_{\text{cur}}) \times \mathbb{1}_{\{1\}}(s'_{\text{pick}}) \\ + [1 - p_s \times \mathbb{1}_{\{s_{\text{key}}\}}(s_{\text{cur}})] \times \mathbb{1}_{\{s_{\text{pick}}\}}(s'_{\text{pick}})]\} & , \text{otherwise} \end{cases}$$

$$R_{2,2}((s_{\text{cur}}, s_{\text{pick}}, s_{\text{open}}), a, (s'_{\text{cur}}, s'_{\text{pick}}, s'_{\text{open}})) := R_0 \times \mathbb{1}_{\{1\}}(s'_{\text{open}}) + r_0 \times [1 - \mathbb{1}_{\{1\}}(s'_{\text{open}})] \quad,$$

$$\Gamma_{2,2}((s_{\text{cur}}, s_{\text{pick}}, s_{\text{open}}), a, (s'_{\text{cur}}, s'_{\text{pick}}, s'_{\text{open}})) := 1 \quad.$$

Now we move to the target MDP of difficulty 3. For $\text{MDP}_{3,1} :=$ $(\mathcal{S}_{3,1}, \mathcal{S}_{3,1}^{\text{init}}, \mathcal{S}_{3,1}^{\text{end}}, \mathcal{A}_{3,1}, P_{3,1}, R_{3,1}, \Gamma_{3,1})$ related to picking up the goal, we have, for $s = (s_{\text{cur}}, s_{\textbf{pick}}, s_{\textbf{open}}, s_{\text{done}})$ (and similarly for $s'$)

$$\mathcal{S}_{3,1} := \{s\},$$

$$\mathcal{S}_{3,1}^{\text{init}} := \{s \in \mathcal{S}_{3,1} : s_{\text{cur}} \notin \Omega_{\text{room}_2}, s_{\text{done}} = 0\} \cup \{s \in \mathcal{S}_{3,1} : s_{\text{open}_1} = 1, s_{\text{done}} = 0\},$$

$$\mathcal{S}_{3,1}^{\text{end}} := \{s \in \mathcal{S}_{3,1} : s_{\text{done}} = 1\},$$

$$\mathcal{A}_{3,1} := \mathcal{A}_{\text{dir}} \cup \{\text{pick}, \text{open}, a^{\text{end}}\},$$

$$P_{3,1}(s, a, s')$$

$$:= \begin{cases} \mathbb{1}_{\{s_{\textbf{pick}}\}}(s'_{\textbf{pick}}) \times \mathbb{1}_{\{s_{\textbf{open}}\}}(s'_{\textbf{open}}) \times \mathbb{1}_{\{s_{\text{done}}\}}(s'_{\text{done}}) \\ \times [p_s \times \mathbb{1}_{\Omega(s)}(s_{\text{cur}} + a) \times \mathbb{1}_{\{s_{\text{cur}}+a\}}(s'_{\text{cur}}) \\ + (1 - p_s \times \mathbb{1}_{\Omega(s)}(s_{\text{cur}} + a)) \times \mathbb{1}_{\{s_{\text{cur}}\}}(s'_{\text{cur}})] & , \text{if } a \in \mathcal{A}_{\text{dir}} \\ \\ \mathbb{1}_{\{s_{\text{cur}}\}}(s'_{\text{cur}}) \times \mathbb{1}_{\{s_{\textbf{pick}}\}}(s'_{\textbf{pick}}) \times \mathbb{1}_{\{s_{\text{done}}\}}(s'_{\text{done}}) \\ \times \{(1 - p_s) \times \mathbb{1}_{\{s_{\textbf{open}}\}}(s'_{\textbf{open}}) \\ + p_s \times \prod_{i=1}^{3} [\mathbb{1}_{\mathcal{N}(s_{\text{door}_i})}(s_{\text{cur}}) \times \mathbb{1}_{\{1\}}(s_{\text{pick}_i}) \times \mathbb{1}_{\{1\}}(s'_{\text{open}_i}) \\ + [1 - \mathbb{1}_{\mathcal{N}(s_{\text{door}_i})}(s_{\text{cur}}) \times \mathbb{1}_{\{1\}}(s_{\text{pick}_i})] \times \mathbb{1}_{\{s_{\text{open}_i}\}}(s'_{\text{open}_i})]\} & , \text{if } a = \text{open} \\ \\ \mathbb{1}_{\{s_{\text{cur}}\}}(s'_{\text{cur}}) \times \mathbb{1}_{\{s_{\textbf{open}}\}}(s'_{\textbf{open}}) \\ \times \{(1 - p_s) \times \mathbb{1}_{\{s_{\textbf{pick}}\}}(s'_{\textbf{pick}}) \times \mathbb{1}_{\{s_{\text{done}}\}}(s'_{\text{done}}) \\ + p_s \times \prod_{i=1}^{3} [\mathbb{1}_{\{s_{\text{key}_i}\}}(s_{\text{cur}}) \times \mathbb{1}_{\{1\}}(s'_{\text{pick}_i}) \\ + \mathbb{1}_{\{s_{\text{key}_i}\}^c}(s_{\text{cur}}) \times \mathbb{1}_{\{s_{\text{pick}_i}\}}(s'_{\text{pick}_i})] \\ \times [\mathbb{1}_{\{s_{\text{goal}}\}}(s_{\text{cur}}) \times \mathbb{1}_{\{1\}}(s'_{\text{done}}) + \mathbb{1}_{\{s_{\text{goal}}\}^c}(s_{\text{cur}}) \times \mathbb{1}_{\{s_{\text{done}}\}}(s'_{\text{done}})]\} & , \text{if } a = \text{pick} \end{cases}$$

$$R_{3,1}(s, a, s') := R_0 \times \mathbb{1}_{\{1\}}(s'_{\text{done}}) + r_0 \times \mathbb{1}_{\{1\}^c}(s'_{\text{done}}),$$

$$\Gamma_{3,1}(s, a, s') := 1.$$

(E.2)

We use $\text{cur}, \text{dest}\cdot$ in Fig. 1 instead of $s_{\text{cur}}, s_{\text{dest}}\cdot$ for simplicity, and similarly for other objects, here and elsewhere. Here the teacher has the role to set all the parameters of the problems, including the reward for retrieving $\text{goal}$ or the negative reward for taking one extra step without making progress.

### E.1.3 Compressed MDPs

For $\text{MDP}_{2,1}$, the student constructs the second level MDP $\text{MDP}^2_{2,1} = (\mathcal{S}_{2,1}, \mathcal{S}^{\text{init}}_{2,1}, \mathcal{S}^{\text{end}}_{2,1}, \overline{\Pi^1_{2,1}}, P^2_{2,1}, R^2_{2,1}, \Gamma^2_{2,1})$, with $P^2_{2,1}, R^2_{2,1}, \Gamma^2_{2,1}$ as follows:

$$P^2_{2,1}(s, (\pi^1_{2,1})_\theta, s') = \mathbb{1}_{\{(d^1_{2,1})_\theta(s,0)\}}((d^1_{2,1})_\theta(s, s')),$$

$$R^2_{2,1}(s, (\pi^1_{2,1})_\theta, s') = r^1_{2,1} + r_0 \times \left[ \mathbb{E}[X \mid X \sim \text{NB}((d^1_{2,1})_\theta(s, s'), p_s)] + (d^1_{2,1})_\theta(s, s') \right], \quad \text{(E.3)}$$

$$\Gamma^2_{2,1}(s, (\pi^1_{2,1})_\theta, s') = 1,$$

where $r^1_{2,1} < 0$ is a negative reward set by the teacher (here equal to $-10$); $(d^1_{2,1})_\theta : \{(s, s') : s, s' \in \mathcal{S}_{2,1}\} \to \mathbb{N} \cup \{+\infty\}$, defined similarly to $(d^1_{2,2})_\theta$, is a parametric family of distance functions parametrized by $\theta \in \Theta^1_{2,1}$. Given the parameter $\theta$, the agent's starting state $s$ and current state $s'$, the value of $(d^1_{2,1})_\theta(s, s')$ incorporates the information of $(\pi^1_{2,1})_\theta$ by defining the infinite sequence $\{s_i\}_{i=0}^\infty$ inductively starting from $s_0 = s$: for $i \in \mathbb{N}$, there exists a unique action $a \in \mathcal{A}^1_{2,1}$ such that $(\pi^1_{2,1})_\theta(s_i, a) = 1$ because $\overline{\pi}^{\text{nav}}_{\text{dense}}$ is assumed to be deterministic, and we let

$$s_{i+1} := \begin{cases} \sup_{s \in \mathcal{S}_{2,1} - \{s_i\}} P_{2,1}(s_i, a, s) & , P_{2,1}(s_i, a, s_i) < 1 \\ s_i & , \text{otherwise} \end{cases}.$$

Then, we set $(d^1_{2,1})_\theta(s, s')$ to be the smallest $i \in \mathbb{N}$ such that $s_i = s'$ if there exists such an $i$, and set it to be $+\infty$ otherwise. Consequently, we denote $(d^1_{2,1})_\theta(s, 0) := \sup\{(d^1_{2,1})_\theta(s, s') : s' \in \mathcal{S}_{2,1}, (d^1_{2,1})_\theta(s, s') < +\infty\}$.

For $\text{MDP}_{2,2}$, the student constructs the second level MDP $\text{MDP}^2_{2,2} = (\mathcal{S}_{2,2}, \mathcal{S}^{\text{init}}_{2,2}, \mathcal{S}^{\text{end}}_{2,2}, \overline{\Pi^1_{2,2}}, P^2_{2,2}, R^2_{2,2}, \Gamma^2_{2,2})$, with $P^2_{2,2}, R^2_{2,2}, \Gamma^2_{2,2}$ as follows:

$$P^2_{2,2}(s, (\pi^1_{2,2})^\lambda_\theta, s')$$
$$= \begin{cases} \begin{aligned} &\mathbb{1}_{\{s_{\text{cur}}\}}(s'_{\text{cur}}) \times \{\mathbb{1}_{\{\text{open}\}}(\theta) \times \mathbb{1}_{\{s_{\text{pick}}\}}(s'_{\text{pick}}) \\ &\times \big[ p_s \times \mathbb{1}_{\mathcal{N}(s_{\text{door}})}(s_{\text{cur}}) \times \mathbb{1}_{\{1\}}(s_{\text{pick}}) \times \mathbb{1}_{\{1\}}(s'_{\text{open}}) \\ &+ [1 - p_s \times \mathbb{1}_{\mathcal{N}(s_{\text{door}})}(s_{\text{cur}}) \times \mathbb{1}_{\{1\}}(s_{\text{pick}})] \times \mathbb{1}_{\{s_{\text{open}}\}}(s'_{\text{open}}) \big] \\ &+ \mathbb{1}_{\{\text{pick}\}}(\theta) \times \mathbb{1}_{\{s_{\text{open}}\}}(s'_{\text{open}}) \times \big[ p_s \times \mathbb{1}_{\{s_{\text{key}}\}}(s_{\text{cur}}) \times \mathbb{1}_{\{1\}}(s'_{\text{pick}}) \\ &+ [1 - p_s \times \mathbb{1}_{\{s_{\text{key}}\}}(s_{\text{cur}})] \times \mathbb{1}_{\{s_{\text{pick}}\}}(s'_{\text{pick}}) \big]\} \end{aligned} & , \text{if } \lambda = \alpha \\[2pt] \mathbb{1}_{\{(d^1_{2,2})_\theta(s,0)\}}((d^1_{2,2})_\theta(s, s')) & , \text{if } \lambda = \beta \end{cases}$$

$$R^2_{2,2}(s, (\pi^1_{2,2})^\lambda_\theta, s')$$
$$= \begin{cases} R_0 \times \mathbb{1}_{\{1\}}(s'_{\text{open}}) + r_0 \times [1 - \mathbb{1}_{\{1\}}(s'_{\text{open}})] & , \text{if } \lambda = \alpha \\ r^1_{2,2} + r_0 \times \left[ \mathbb{E}[X \mid X \sim \text{NB}((d^1_{2,2})_\theta(s, s'), p_s)] + (d^1_{2,2})_\theta(s, s') \right] & , \text{if } \lambda = \beta \end{cases}$$

$$\Gamma^2_{2,2}(s, (\pi^1_{2,2})^\lambda_\theta, s') = 1,$$

where $s = (s_{\text{cur}}, s_{\text{pick}}, s_{\text{open}}), s' = (s'_{\text{cur}}, s'_{\text{pick}}, s'_{\text{open}}); r^1_{2,2} < 0$ is a negative reward set by the teacher, and here it is set to $-10$; $X \sim \text{NB}(r, p)(r \in \mathbb{Z}^+, 0 < p \leq 1)$ means $X$ follows negative binomial (or Pascal) distribution with $r$ successes and success probability $p$, with the probability mass function $f_{nb}(k; r, p) := \binom{k + r - 1}{k}(1 - p)^k p^r (k \geq 0)$, so $\mathbb{E}[X \mid X \sim \text{NB}(r, p)] = \frac{(1-p) \times r}{p}$; $(d^1_{2,2})_\theta : \{(s, s') : s, s' \in \mathcal{S}_{2,2}\} \to \mathbb{N} \cup \{+\infty\}$, is a parametric family of distance functions parametrized by $\theta \in (\Theta^1_{2,2})_\beta$. Given the parameter $\theta$, the agent's starting state $s$ and current state $s'$, the value of $(d^1_{2,2})_\theta(s, s')$ incorporates the information of $(\pi^1_{2,2})^\beta_\theta$ by defining the infinite sequence $\{s_i\}_{i=0}^\infty$ inductively starting from $s_0 = s$: for $i \in \mathbb{N}$, there exists a unique action $a \in \mathcal{A}^1_{2,2}$ such that $(\pi^1_{2,2})_\theta(s_i, a) = 1$ because $\overline{\pi}^{\text{nav}}$ is assumed to be deterministic, and we let

$$s_{i+1} := \begin{cases} \sup_{s \in \mathcal{S}_{2,2} - \{s_i\}} P_{2,2}(s_i, a, s) & , P_{2,2}(s_i, a, s_i) < 1 \\ s_i & , \text{otherwise} \end{cases}.$$

Then, we set $(d_{2,2}^1)_\theta(s, s')$ to be the smallest $i \in \mathbb{N}$ such that $s_i = s'$ if there exists such an $i$, and set it to be $+\infty$ otherwise. Consequently, we denote $(d_{2,2}^1)_\theta(s, 0) := \sup\{(d_{2,2}^1)_\theta(s, s') : s' \in \mathcal{S}_{2,2}, (d_{2,2}^1)_\theta(s, s') < +\infty\}$. Notice that if $\overline{\pi}^{\text{nav}}$ is not deterministic, then the sequence defined here is stochastic, making the calculations slightly more complicated.

Now we move to the target MDP of difficulty 3. For $\text{MDP}_{3,1}$, the student constructs the second level MDP $\text{MDP}_{3,1}^2 = (\mathcal{S}_{3,1}, \mathcal{S}_{3,1}^{\text{init}}, \mathcal{S}_{3,1}^{\text{end}}, \overline{\Pi_{3,1}^1}, P_{3,1}^2, R_{3,1}^2, \Gamma_{3,1}^2)$, where $P_{3,1}^2$, $R_{3,1}^2$, and $\Gamma_{3,1}^2$ are as follows:

$$P_{3,1}^2(s, (\pi_{3,1}^1)_\theta^\lambda, s') \tag{E.5}$$

$$= \begin{cases} \begin{aligned} & \mathbb{1}_{\{s_{\text{cur}}\}}(s'_{\text{cur}}) \times \mathbb{1}_{\{s_{\mathbf{pick}}\}}(s'_{\mathbf{pick}}) \times \mathbb{1}_{\{s_{\text{done}}\}}(s'_{\text{done}}) \\ & \times \{(1 - p_s) \times \mathbb{1}_{\{s_{\mathbf{open}}\}}(s'_{\mathbf{open}}) \\ & + p_s \times \prod_{i=1}^3 \left[ \mathbb{1}_{\mathcal{N}(s_{\text{door}_i})}(s_{\text{cur}}) \times \mathbb{1}_{\{1\}}(s_{\text{pick}_i}) \times \mathbb{1}_{\{1\}}(s'_{\text{open}_i}) \right. \\ & \left. + [1 - \mathbb{1}_{\mathcal{N}(s_{\text{door}_i})}(s_{\text{cur}}) \times \mathbb{1}_{\{1\}}(s_{\text{pick}_i})] \times \mathbb{1}_{\{s_{\text{open}_i}\}}(s'_{\text{open}_i})] \} \end{aligned} & , \text{if } \theta = \text{open} \\ \begin{aligned} & \mathbb{1}_{\{s_{\text{cur}}\}}(s'_{\text{cur}}) \times \mathbb{1}_{\{s_{\mathbf{open}}\}}(s'_{\mathbf{open}}) \times \{(1 - p_s) \times \mathbb{1}_{\{s_{\mathbf{pick}}\}}(s'_{\mathbf{pick}}) \times \mathbb{1}_{\{s_{\text{done}}\}}(s'_{\text{done}}) \\ & + p_s \times \prod_{i=1}^3 \left[ \mathbb{1}_{\{s_{\text{key}_i}\}}(s_{\text{cur}}) \times \mathbb{1}_{\{1\}}(s'_{\text{pick}_i}) + \mathbb{1}_{\{s_{\text{key}_i}\}^c}(s_{\text{cur}}) \times \mathbb{1}_{\{s_{\text{pick}_i}\}}(s'_{\text{pick}_i}) \right] \\ & \times \left[ \mathbb{1}_{\{s_{\text{goal}}\}}(s_{\text{cur}}) \times \mathbb{1}_{\{1\}}(s'_{\text{done}}) + \mathbb{1}_{\{s_{\text{goal}}\}^c}(s_{\text{cur}}) \times \mathbb{1}_{\{s_{\text{done}}\}}(s'_{\text{done}})] \} \end{aligned} & , \text{if } \theta = \text{pick} \\ \mathbb{1}_{\{(d_{3,1}^1)_\theta(s,0)\}}((d_{3,1}^1)_\theta(s, s')) & , \text{if } \lambda = \beta \end{cases}$$

$$\tag{E.6}$$

$$R_{3,1}^2(s, (\pi_{3,1}^1)_\theta^\lambda, s')$$

$$= \begin{cases} R_0 \times \mathbb{1}_{\{1\}}(s'_{\text{done}}) + r_0 \times \mathbb{1}_{\{0\}}(s'_{\text{done}}) & , \text{if } \lambda = \alpha \\ r_{3,1}^1 + r_0 \times \left[ \mathbb{E}[X \mid X \sim \text{NB}((d_{3,1}^1)_\theta(s, s'), p_s)] + (d_{3,1}^1)_\theta(s, s') \right] & , \text{if } \lambda = \beta \end{cases},$$

$$\Gamma_{3,1}^2(s, (\pi_{3,1}^1)_\theta^\lambda, s') = 1,$$

where $(d_{3,1}^1)_\theta$ is defined similarly to $(d_{2,2}^1)$; $r_{3,1}^1 < 0$ is some negative reward set by the teacher and here it is set to $-10$.

Then, the student constructs the third-level MDP $\text{MDP}_{3,1}^3 = (\mathcal{S}_{3,1}, \mathcal{S}_{3,1}^{\text{init}}, \mathcal{S}_{3,1}^{\text{end}}, \overline{\Pi_{3,1}^2}, P_{3,1}^3, R_{3,1}^3, \Gamma_{3,1}^3)$, where $P_{3,1}^3, R_{3,1}^3, \Gamma_{3,1}^3$ are as follows:

$$P_{3,1}^3(s, (\pi_{3,1}^2)_\theta \mid_{\{\alpha, \beta\}}, s') = \mathbb{1}_{\{(d_{3,1}^2)_\theta(s,0)\}}((d_{3,1}^2)_\theta(s, s')),$$

$$R_{3,1}^3(s, (\pi_{3,1}^2)_\theta \mid_{\{\alpha, \beta\}}, s') = R_0 \times \mathbb{1}_{\{1\}}(s'_{\text{done}}) + r_{3,1}^2$$
$$+ r_{3,1}^1 \times [\mathbb{1}_{(0,+\infty)}(((d_{3,1}^2)_\theta(s, s'))_1) + \mathbb{1}_{(0,+\infty)}(((d_{3,1}^2)_\theta(s, s'))_3)]$$
$$+ r_0 \times \left[ \mathbb{E}[X \mid X \sim \text{NB}(\sum_{i=1}^4 ((d_{3,1}^2)_\theta(s, s'))_i, p_s)] + \sum_{i=1}^4 ((d_{3,1}^2)_\theta(s, s'))_i \right],$$

$$\Gamma_{3,1}^3(s, (\pi_{3,1}^2)_\theta \mid_{\{\alpha, \beta\}}, s') = 1,$$

$$\tag{E.7}$$

where $r_{3,1}^2 < 0$ is some negative reward set by the teacher, and here it is set to $-10$; $(d_{3,1}^2)_\theta : \{(s, s') : s, s' \in \mathcal{S}_{3,1}\} \to \mathbb{N}^4 \cup \{+\infty\}$ generalizing $(d_{3,1}^1)_\theta$ is a parametric family of vector distance functions parametrized by $\theta \in \Theta_{3,1}^2$. Given the parameter $\theta$, the agent's starting state $s$ and current state $s'$, the value of $(d_{3,1}^2)_\theta(s, s')$ incorporates the information of $(\pi_{3,1}^2)_\theta \mid_{\{\alpha, \beta\}}$ $(\theta \in \Theta_{3,1}^2)$ by defining the infinite sequence $\{s_i\}_{i=0}^\infty$ inductively starting from $s_0 := s$: for $i \in \mathbb{N}$, there exists a unique action $\pi_i \in \mathcal{A}_{3,1}^2$ such that $(\pi_{3,1}^2)_\theta \mid_{\{\alpha, \beta\}} (s_i, \pi_i) = 1$, and then there exists a unique action $a_i \in \mathcal{A}_{3,1}^1$ such that $\pi_i(s_i, a_i) = 1$ because $\overline{\pi}^{\text{nav}}$ is assumed to be deterministic, ($a_i = a^{\text{end}}$ if $\pi_i = a^{\text{end}}$), and we let

$$s_{i+1} := \begin{cases} \sup_{s \in \mathcal{S}_{3,1} - \{s_i\}} P_{3,1}(s_i, a_i, s) & , P_{3,1}(s_i, a_i, s) < 1 \\ s_i & , \text{otherwise} \end{cases}.$$

Then, we find the smallest $i_0 \in \mathbb{N}$ such that $s_{i_0} = s'$ (we set $(d_{3,1}^2)_\theta(s, s')$ to be $+\infty$ if there does not exist such an $i_0$), and we set $(d_{3,1}^2)_\theta(s, s')$ to be $(\sum_{i=0}^{i_0-1} \mathbb{1}_{\{\pi_{\mathrm{key}_{i'}}\}}(\pi_i), \sum_{i=0}^{i_0-1} \mathbb{1}_{\{\pi_{\mathrm{pick}}\}}(\pi_i), \sum_{i=0}^{i_0-1} \mathbb{1}_{\{\pi_{\mathrm{door}_{i'}}\}}(\pi_i),$ $\sum_{i=0}^{i_0-1} \mathbb{1}_{\{\pi_{\mathrm{open}}\}}(\pi_i))$ for $\theta = (\mathrm{key}_{i'}, \mathrm{door}_{i'})(i' = 1, 2, 3)$ and we set $(d_{3,1}^2)_\theta(s, s')$ to be $(\sum_{i=0}^{i_0-1} \mathbb{1}_{\{\pi_{\mathrm{goal}}\}}(\pi_i),$ $\sum_{i=0}^{i_0-1} \mathbb{1}_{\{\pi_{\mathrm{pick}}\}}(\pi_i), 0, \sum_{i=0}^{i_0-1} \mathbb{1}_{\{\pi_{\mathrm{open}}\}}(\pi_i))$ for $\theta = (\mathrm{goal}, \mathrm{goal})$. Then, since for any fixed $s$, there is a natural ordering between any two elements in $\{(d_{3,1}^2)_\theta(s, s') : s' \in \mathcal{S}_{3,1}, (d_{3,1}^2)_\theta(s, s') < +\infty\}$ induced by ordering on $\mathbb{N}$, we still denote $(d_{3,1}^2)_\theta(s, 0) := \sup\{(d_{3,1}^2)_\theta(s, s') : s' \in \mathcal{S}_{3,1}, (d_{3,1}^2)_\theta(s, s') \neq +\infty\}$.

### E.1.4 (PARTIAL) POLICIES AND (PARTIAL) POLICY GENERATORS

For $\mathrm{MDP}_{2,1}$:

$g_{2,1}^1 : \Theta_{2,1}^1 \to \{(\pi_{2,1}^1)_\theta : \theta \in \Theta_{2,1}^1\}$ is as follows:

$$g_{2,1}^1(\theta) = (\pi_{2,1}^1)_\theta, \tag{E.8}$$

and here $(\pi_{2,1}^1)_\theta : \mathcal{SA}_{2,1}^1 \to [0, 1]$ $(\theta \in \Theta_{2,1}^1)$ are as follows:

$$(\pi_{2,1}^1)_\theta((s_{\mathrm{cur}}, s_{\mathrm{dest}}), a) = (\overline{\pi}_{\mathrm{dense}}^{\mathrm{nav}} \circ (e_{2,1}^1)_\theta)((s_{\mathrm{cur}}, s_{\mathrm{dest}}), a)) = \overline{\pi}_{\mathrm{dense}}^{\mathrm{nav}}(s_{\mathrm{cur}}, s_\theta, a).$$

For $\mathrm{MDP}_{2,2}$:

$$(\pi_{2,2}^1)_\theta^\beta((s_{\mathrm{cur}}, s_{\mathrm{pick}}, s_{\mathrm{open}}), a)$$
$$= \begin{cases} \overline{\pi}^{\mathrm{nav}}(s_{\mathrm{cur}}, s_\theta, a) & \text{, if } \overline{\pi}^{\mathrm{nav}}(s_{\mathrm{cur}}, s_\theta, s_{\mathrm{door}} - s_{\mathrm{cur}}) = 0 \text{ and } a \in \mathcal{A}_{\mathrm{dir}} \cup \{a^{\mathrm{end}}\} \\ \mathbb{1}_{\{a^{\mathrm{end}}\}}(a) & \text{, otherwise} \end{cases}. \tag{E.9}$$

Here, the navigation skill $\overline{\pi}^{\mathrm{nav}}$ is extracted from $\mathrm{MDP}_{2,1}$. One comment here is that $(\pi_{2,2}^1)_{\mathrm{key}}^\beta$ takes the action $a^{\mathrm{end}}$ with probability one at states satisfying $s_{\mathrm{cur}} = s_{\mathrm{key}}$, telling the agent it could stop the current policy because it has reached the current subgoal and should move on to the next one. This shows why forcing a policy $\pi$ to satisfy $\pi(s, a) = \mathbb{1}_{\{a^{\mathrm{end}}\}}(a)$ for any $s \in \mathcal{S}^{\mathrm{end}}$ is important in the construction of MMDPs and policy transfer.

Another more subtle comment here is that there is a little inconsistency between $(\pi_{2,2}^1)_{\mathrm{door}}^\beta$ and its semantic representation "go from $\mathrm{cur}$ to $\mathrm{door}$" in Fig. 1, since $(\pi_{2,2}^1)_{\mathrm{door}}^\beta$ actually stops at $\mathcal{N}(s_{\mathrm{door}})$ instead of stopping at $s_{\mathrm{door}}$, because the door may be closed initially. The way this was realized is that we ruled out the final step of reaching $s_{\mathrm{door}}$ from $(\pi_{2,2}^1)_{\mathrm{door}}^\beta$ generated from the policy generator $(g_{2,2}^1)_\beta$, and replaced it by taking the action $a^{\mathrm{end}}$ with probability one at states satisfying $s_{\mathrm{cur}} \in \mathcal{N}(s_{\mathrm{door}})$. This is another important instance showcasing the necessity of incorporating the special "$a^{\mathrm{end}}$". This applies to both $(\pi_{2,2}^1)_{\mathrm{door}}^\beta$ here and the following ingredients related to "going to a door" throughout all the examples.

$$\pi_{2,2,*}^2(s, (\pi_{2,2}^1)_\theta^\lambda) = \mathbb{1}_{\{(\theta_{0,*}(s), \lambda_{0,*}(s))\}}((\theta, \lambda)), \tag{E.10}$$

with

$$(\theta_{0,*}(s), \lambda_{0,*}(s)) \in \tag{E.11}$$
$$\underset{(\theta_0, \lambda_0)}{\arg\max} \underset{(\theta_1, \lambda_1), \cdots, (\theta_{\tau-1}, \lambda_{\tau-1})}{\max} \mathbb{E}_{\tau, S_{1:\tau}}[(R_{2,2}^2)_{0,\tau} | S_0 = s, A_t = (\pi_{2,2}^1)_{\theta_t}^{\lambda_t} \text{ for any } 0 \leq t \leq \tau - 1].$$

$$\pi_{2,2,*}^2((s_{\mathrm{cur}}, s_{\mathrm{pick}}, s_{\mathrm{open}}), a)$$
$$= \mathbb{1}_{\{1\}}(s_{\mathrm{open}}) \times \mathbb{1}_{\{a^{\mathrm{end}}\}}(a) + \mathbb{1}_{\{0\}}(s_{\mathrm{open}}) \times \mathbb{1}_{\{0\}}(s_{\mathrm{pick}}) \times \left[\mathbb{1}_{\{s_{\mathrm{key}}\}}(s_{\mathrm{cur}}) \times \mathbb{1}_{\{(\pi_{2,2}^1)_{\mathrm{pick}}^\alpha\}}(a)\right]$$

$$+ \mathbb{1}_{\{s_{\mathrm{key}}\}^c}(s_{\mathrm{cur}}) \times \mathbb{1}_{\{(\pi^1_{2,2})^\beta_{\mathrm{key}}\}}(a)] + \mathbb{1}_{\{0\}}(s_{\mathrm{open}}) \times \mathbb{1}_{\{1\}}(s_{\mathrm{pick}}) \tag{E.12}$$

$$\times \left[ \mathbb{1}_{\mathcal{N}(s_{\mathrm{door}})}(s_{\mathrm{cur}}) \times \mathbb{1}_{\{(\pi^1_{2,2})^\alpha_{\mathrm{open}}\}}(a) + \mathbb{1}_{\mathcal{N}(s_{\mathrm{door}})^c}(s_{\mathrm{cur}}) \times \mathbb{1}_{\{(\pi^1_{2,2})^\beta_{\mathrm{door}}\}}(a) \right].$$

$$\pi_{2,2}((s_{\mathrm{cur}}, s_{\mathrm{pick}}, s_{\mathrm{open}}), a) \tag{E.13}$$

$$= \begin{cases} \begin{aligned} &\mathbb{1}_{\{0\}}(s_{\mathrm{pick}}) \times \mathbb{1}_{\{s_{\mathrm{key}}\}^c}(s_{\mathrm{cur}}) \times \overline{\pi}^{\mathrm{nav}}(s_{\mathrm{cur}}, s_{\mathrm{key}}, a) \\ &+ \mathbb{1}_{\{1\}}(s_{\mathrm{pick}}) \times \mathbb{1}_{\mathcal{N}(s_{\mathrm{door}})^c}(s_{\mathrm{cur}}) \times \overline{\pi}^{\mathrm{nav}}(s_{\mathrm{cur}}, s_{\mathrm{door}}, a) \quad , \text{if } a \in \mathcal{A}_{\mathrm{dir}} \\ \\ &\mathbb{1}_{\{1\}}(s_{\mathrm{open}}) \times \mathbb{1}_{\{a^{\mathrm{end}}\}}(a) + \mathbb{1}_{\{0\}}(s_{\mathrm{open}}) \\ &\times \left[ \mathbb{1}_{\{0\}}(s_{\mathrm{pick}}) \times \mathbb{1}_{\{s_{\mathrm{key}}\}}(s_{\mathrm{cur}}) \times \mathbb{1}_{\{\mathrm{pick}\}}(a) \right. \\ &\left. + \mathbb{1}_{\{1\}}(s_{\mathrm{pick}}) \times \mathbb{1}_{\mathcal{N}(s_{\mathrm{door}})}(s_{\mathrm{cur}}) \times \mathbb{1}_{\{\mathrm{open}\}}(a) \right] \quad\quad , \text{otherwise} \end{aligned} \end{cases} \tag{E.14}$$

For $\mathrm{MDP}_{3,1}$:

$(g^1_{3,1})_\alpha : (\Theta^1_{3,1})_\alpha \to \{(\pi^1_{3,1})^\alpha_\theta : \theta \in (\Theta^1_{3,1})_\alpha\}$ is defined as

$$(g^1_{3,1})_\alpha(\theta) = (\pi^1_{3,1})^\alpha_\theta, \tag{E.15}$$

and here $(\pi^1_{3,1})^\alpha_\theta = \mathrm{id} \circ (e^1_{3,1})^\alpha_\theta = (e^1_{3,1})^\alpha_\theta$ $(\theta = \mathtt{pick}, \mathtt{open})$.

$(g^1_{3,1})_\beta : (\Theta^1_{3,1})_\beta \to \{(\pi^1_{3,1})^\beta_\theta : \theta \in (\Theta^1_{3,1})_\beta\}$ is defined as

$$(g^1_{3,1})_\beta(\theta) := (\pi^1_{3,1})^\beta_\theta, \tag{E.16}$$

and here $(\pi^1_{3,1})^\beta_\theta : \mathcal{SA}^1_{3,1} \to [0,1]$ for $\theta \in \{\mathtt{key}_1, \mathtt{key}_2, \mathtt{key}_3, \mathtt{goal}\}$ are as follows:

$$(\pi^1_{3,1})^\beta_\theta((s_{\mathrm{cur}}, s_{\mathbf{pick}}, s_{\mathbf{open}}, s_{\mathrm{done}}), a) = (\overline{\pi}^{\mathrm{nav}} \circ (e^1_{3,1})^\beta_\theta)((s_{\mathrm{cur}}, s_{\mathrm{pick}}, s_{\mathrm{open}}), a)$$

$$= \begin{cases} \overline{\pi}^{\mathrm{nav}}(s_{\mathrm{cur}}, s_\theta, a) & , \text{if } a \in \mathcal{A}_{\mathrm{dir}} \cup \{a^{\mathrm{end}}\} \\ 0 & , \text{otherwise} \end{cases},$$

$(\pi^1_{3,1})^\beta_\theta : \mathcal{SA}^1_{3,1} \to [0,1]$ for $\theta \in \{\mathtt{door}_1, \mathtt{door}_2, \mathtt{door}_3\}$ are as follows:

$$(\pi^1_{3,1})^\beta_\theta((s_{\mathrm{cur}}, s_{\mathbf{pick}}, s_{\mathbf{open}}, s_{\mathrm{done}}), a)$$

$$= (\overline{\pi}^{\mathrm{nav}} \circ (e^1_{3,1})^\beta_\theta)((s_{\mathrm{cur}}, s_{\mathrm{pick}}, s_{\mathrm{open}}), a)$$

$$= \begin{cases} \overline{\pi}^{\mathrm{nav}}(s_{\mathrm{cur}}, s_\theta, a) & , \text{if } \overline{\pi}^{\mathrm{nav}}(s_{\mathrm{cur}}, s_\theta, s_\theta - s_{\mathrm{cur}}) = 0 \text{ and } a \in \mathcal{A}_{\mathrm{dir}} \cup \{a^{\mathrm{end}}\} \\ \mathbb{1}_{\{a^{\mathrm{end}}\}}(a) & , \text{otherwise} \end{cases}.$$

$g^2_{3,1}|_{\{\alpha,\beta\}} : \Theta^2_{3,1} \to \{(\pi^2_{3,1})_\theta |_{\{\alpha,\beta\}} : \theta \in \Theta^2_{3,1}\}$ is as follows:

$$g^2_{3,1}|_{\{\alpha,\beta\}}(\theta) = (\pi^2_{3,1})_\theta |_{\{\alpha,\beta\}}, \tag{E.17}$$

and here for $\theta = (\mathtt{key}_i, \mathtt{door}_i)(i = 1, 2, 3)$, $(\pi^2_{3,1})_\theta |_{\{\alpha,\beta\}} : \mathcal{SA}^2_{3,1} \to [0,1]$ are as follows:

$$(\pi^2_{3,1})_\theta |_{\{\alpha,\beta\}}((s_{\mathrm{cur}}, s_{\mathbf{pick}}, s_{\mathbf{open}}, s_{\mathrm{done}}), a)$$

$$= (\overline{\pi}^{\mathrm{concat}} \circ (e^2_{3,1})_\theta)((s_{\mathrm{cur}}, s_{\mathbf{pick}}, s_{\mathbf{open}}, s_{\mathrm{done}}), a)$$

$$= \begin{cases} \overline{\pi}^{\mathrm{concat}}(\mathbb{1}_{\{s_{\mathrm{key}_i}\}}(s_{\mathrm{cur}}), \mathbb{1}_{\{1\}}(s_{\mathrm{pick}_i}) \times \mathbb{1}_{\mathcal{N}(s_{\mathrm{door}_i})}(s_{\mathrm{cur}}), s_{\mathrm{pick}_i}, s_{\mathrm{open}_i}, \pi_\theta) & , \text{if } a = (\pi^1_{3,1})^\alpha_\theta \\ \overline{\pi}^{\mathrm{concat}}(\mathbb{1}_{\{s_{\mathrm{key}_i}\}}(s_{\mathrm{cur}}), \mathbb{1}_{\{1\}}(s_{\mathrm{pick}_i}) \times \mathbb{1}_{\mathcal{N}(s_{\mathrm{door}_i})}(s_{\mathrm{cur}}), s_{\mathrm{pick}_i}, s_{\mathrm{open}_i}, \pi_{\mathrm{key}}) & , \text{if } a = (\pi^1_{3,1})^\beta_{\mathrm{key}_i} \\ \overline{\pi}^{\mathrm{concat}}(\mathbb{1}_{\{s_{\mathrm{key}_i}\}}(s_{\mathrm{cur}}), \mathbb{1}_{\{1\}}(s_{\mathrm{pick}_i}) \times \mathbb{1}_{\mathcal{N}(s_{\mathrm{door}_i})}(s_{\mathrm{cur}}), s_{\mathrm{pick}_i}, s_{\mathrm{open}_i}, \pi_{\mathrm{door}}) & , \text{if } a = (\pi^1_{3,1})^\beta_{\mathrm{door}_i} \\ \overline{\pi}^{\mathrm{concat}}(\mathbb{1}_{\{s_{\mathrm{key}_i}\}}(s_{\mathrm{cur}}), \mathbb{1}_{\{1\}}(s_{\mathrm{pick}_i}) \times \mathbb{1}_{\mathcal{N}(s_{\mathrm{door}_i})}(s_{\mathrm{cur}}), s_{\mathrm{pick}_i}, s_{\mathrm{open}_i}, a) & , \text{if } a = a^{\mathrm{end}} \\ 0 & , \text{otherwise} \end{cases},$$

for $\theta = (\mathtt{goal}, \mathtt{goal})$, $(\pi^2_{3,1})_\theta |_{\{\alpha,\beta\}} : \mathcal{SA}^2_{3,1} \to [0,1]$ is as follows:

$$(\pi^2_{3,1})_\theta |_{\{\alpha,\beta\}}((s_{\mathrm{cur}}, s_{\mathbf{pick}}, s_{\mathbf{open}}, s_{\mathrm{done}}), a)$$

$$= (\overline{\pi}^{\mathrm{concat}} \circ (e_{3,1}^2)_\theta)((s_{\mathrm{cur}}, s_{\mathbf{pick}}, s_{\mathbf{open}}, s_{\mathrm{done}}), a)$$

$$= \begin{cases} \overline{\pi}^{\mathrm{concat}}(\mathbb{1}_{\{s_{\mathrm{goal}}\}}(s_{\mathrm{cur}}), \mathbb{1}_{\{1\}}(s_{\mathrm{done}}) \times \mathbb{1}_{\{s_{\mathrm{goal}}\}}(s_{\mathrm{cur}}), s_{\mathrm{done}}, s_{\mathrm{done}}, \pi_\theta) & , \text{if } a = (\pi_{3,1}^1)_\theta^\alpha \\ \overline{\pi}^{\mathrm{concat}}(\mathbb{1}_{\{s_{\mathrm{goal}}\}}(s_{\mathrm{cur}}), \mathbb{1}_{\{1\}}(s_{\mathrm{done}}) \times \mathbb{1}_{\{s_{\mathrm{goal}}\}}(s_{\mathrm{cur}}), s_{\mathrm{done}}, s_{\mathrm{done}}, \pi_{\mathrm{key}}) & , \text{if } a = (\pi_{3,1}^1)_{\mathrm{goal}}^\beta \\ \overline{\pi}^{\mathrm{concat}}(\mathbb{1}_{\{s_{\mathrm{goal}}\}}(s_{\mathrm{cur}}), \mathbb{1}_{\{1\}}(s_{\mathrm{done}}) \times \mathbb{1}_{\{s_{\mathrm{goal}}\}}(s_{\mathrm{cur}}), s_{\mathrm{done}}, s_{\mathrm{done}}, a) & , \text{if } a = a^{\mathrm{end}} \\ 0 & , \text{otherwise} \end{cases}.$$

$$\pi_{3,1,*}^3((s_{\mathrm{cur}}, s_{\mathbf{pick}}, s_{\mathbf{open}}, s_{\mathrm{done}}), a)$$
$$= \mathbb{1}_{\{1\}}(s_{\mathrm{done}}) \times \mathbb{1}_{\{a^{\mathrm{end}}\}}(a)$$
$$+ [\mathbb{1}_{\Omega_{\mathrm{room}_4} \cup \{\mathrm{door}_3\}}(s_{\mathrm{cur}}) + \mathbb{1}_{\Omega_{\mathrm{room}_3} \cup \{\mathrm{door}_2\}}(s_{\mathrm{cur}}) \times \mathbb{1}_{\{1\}}(s_{\mathrm{open}_3})$$
$$+ \mathbb{1}_{\Omega_{\mathrm{room}_1} \cup \Omega_{\mathrm{room}_2} \cup \{\mathrm{door}_1\}}(s_{\mathrm{cur}}) \times \mathbb{1}_{\{1\}}(s_{\mathrm{open}_2}) \times \mathbb{1}_{\{1\}}(s_{\mathrm{open}_3})] \times \mathbb{1}_{\{(\pi_{3,1}^2)_{\mathrm{goal}}\}}(a) \qquad (E.18)$$
$$+ [\mathbb{1}_{\Omega_{\mathrm{room}_3} \cup \{\mathrm{door}_2\}}(s_{\mathrm{cur}}) + \mathbb{1}_{\Omega_{\mathrm{room}_1} \cup \Omega_{\mathrm{room}_2} \cup \{\mathrm{door}_1\}}(s_{\mathrm{cur}}) \times \mathbb{1}_{\{1\}}(s_{\mathrm{open}_2})] \times \mathbb{1}_{\{0\}}(s_{\mathrm{open}_3}) \times \mathbb{1}_{\{(\pi_{3,1}^2)_{\mathrm{door}_3}\}}(a)$$
$$+ \mathbb{1}_{\Omega_{\mathrm{room}_1} \cup \Omega_{\mathrm{room}_2} \cup \{\mathrm{door}_1\}}(s_{\mathrm{cur}}) \times \mathbb{1}_{\{0\}}(s_{\mathrm{open}_2}) \times \mathbb{1}_{\{(\pi_{3,1}^2)_{\mathrm{door}_2}\}}(a).$$

$$\pi_{3,1,*}^2(s, a)$$
$$= \mathbb{1}_{\{1\}}(s_{\mathrm{done}}) \times \mathbb{1}_{\{a^{\mathrm{end}}\}}(a)$$
$$+ [\mathbb{1}_{\Omega_{\mathrm{room}_4} \cup \{\mathrm{door}_3\}}(s_{\mathrm{cur}}) + \mathbb{1}_{\Omega_{\mathrm{room}_3} \cup \{\mathrm{door}_2\}}(s_{\mathrm{cur}}) \times \mathbb{1}_{\{1\}}(s_{\mathrm{open}_3}) \qquad (E.19)$$
$$+ \mathbb{1}_{\Omega_{\mathrm{room}_1} \cup \Omega_{\mathrm{room}_2} \cup \{\mathrm{door}_1\}}(s_{\mathrm{cur}}) \times \mathbb{1}_{\{1\}}(s_{\mathrm{open}_2}) \times \mathbb{1}_{\{1\}}(s_{\mathrm{open}_3})] \times (\pi_{3,1}^2)_{\mathrm{goal}}(s, a)$$
$$+ [\mathbb{1}_{\Omega_{\mathrm{room}_3} \cup \{\mathrm{door}_2\}}(s_{\mathrm{cur}}) + \mathbb{1}_{\Omega_{\mathrm{room}_1} \cup \Omega_{\mathrm{room}_2} \cup \{\mathrm{door}_1\}}(s_{\mathrm{cur}}) \times \mathbb{1}_{\{1\}}(s_{\mathrm{open}_2})] \times \mathbb{1}_{\{0\}}(s_{\mathrm{open}_3}) \times (\pi_{3,1}^2)_{\mathrm{door}_3}(s, a)$$
$$+ \mathbb{1}_{\Omega_{\mathrm{room}_1} \cup \Omega_{\mathrm{room}_2} \cup \{\mathrm{door}_1\}}(s_{\mathrm{cur}}) \times \mathbb{1}_{\{0\}}(s_{\mathrm{open}_2}) \times (\pi_{3,1}^2)_{\mathrm{door}_2}(s, a).$$

$$\pi_{3,1,*}(s, a)$$
$$= \mathbb{1}_{\{1\}}(s_{\mathrm{done}}) \times \mathbb{1}_{\{a^{\mathrm{end}}\}}(a)$$
$$+ [\mathbb{1}_{\Omega_{\mathrm{room}_4} \cup \{\mathrm{door}_3\}}(s_{\mathrm{cur}}) + \mathbb{1}_{\Omega_{\mathrm{room}_3} \cup \{\mathrm{door}_2\}}(s_{\mathrm{cur}}) \times \mathbb{1}_{\{1\}}(s_{\mathrm{open}_3})$$
$$+ \mathbb{1}_{\Omega_{\mathrm{room}_1} \cup \Omega_{\mathrm{room}_2} \cup \{\mathrm{door}_1\}}(s_{\mathrm{cur}}) \times \mathbb{1}_{\{1\}}(s_{\mathrm{open}_2}) \times \mathbb{1}_{\{1\}}(s_{\mathrm{open}_3})] \times \sum_{\pi \in \Pi_{3,1}^1} (\pi_{3,1}^2)_{\mathrm{goal}}(s, \pi) \times \pi(s, a)$$

$$(E.20)$$

$$+ [\mathbb{1}_{\Omega_{\mathrm{room}_3} \cup \{\mathrm{door}_2\}}(s_{\mathrm{cur}}) + \mathbb{1}_{\Omega_{\mathrm{room}_1} \cup \Omega_{\mathrm{room}_2} \cup \{\mathrm{door}_1\}}(s_{\mathrm{cur}}) \times \mathbb{1}_{\{1\}}(s_{\mathrm{open}_2})]$$
$$\times \mathbb{1}_{\{0\}}(s_{\mathrm{open}_3}) \times \sum_{\pi \in \Pi_{3,1}^1} (\pi_{3,1}^2)_{\mathrm{door}_3}(s, \pi) \times \pi(s, a)$$
$$+ \mathbb{1}_{\Omega_{\mathrm{room}_1} \cup \Omega_{\mathrm{room}_2} \cup \{\mathrm{door}_1\}}(s_{\mathrm{cur}}) \times \mathbb{1}_{\{0\}}(s_{\mathrm{open}_2}) \times \sum_{\pi \in \Pi_{3,1}^1} (\pi_{3,1}^2)_{\mathrm{door}_2}(s, \pi) \times \pi(s, a).$$

For $\mathrm{MDP}_{3,1}'$:

$$(\pi_{3,1,*}^3)'((s_{\mathrm{cur}}, s_{\mathbf{pick}}, s_{\mathbf{open}}, s_{\mathrm{done}}), a)$$
$$= \mathbb{1}_{\{1\}}(s_{\mathrm{done}}) \times \mathbb{1}_{\{a^{\mathrm{end}}\}}(a)$$
$$+ [\mathbb{1}_{\Omega_{\mathrm{room}_3} \cup \{\mathrm{door}_3\}}(s_{\mathrm{cur}}) + \mathbb{1}_{\Omega_{\mathrm{room}_4} \cup \{\mathrm{door}_2\}}(s_{\mathrm{cur}}) \times \mathbb{1}_{\{1\}}(s_{\mathrm{open}_3})$$
$$+ \mathbb{1}_{\Omega_{\mathrm{room}_2} \cup \{\mathrm{door}_1\}}(s_{\mathrm{cur}}) \times \mathbb{1}_{\{1\}}(s_{\mathrm{open}_2}) \times \mathbb{1}_{\{1\}}(s_{\mathrm{open}_3})$$
$$+ \mathbb{1}_{\Omega_{\mathrm{room}_1}}(s_{\mathrm{cur}}) \times \mathbb{1}_{\{1\}}(s_{\mathrm{open}_1}) \times \mathbb{1}_{\{1\}}(s_{\mathrm{open}_2}) \times \mathbb{1}_{\{1\}}(s_{\mathrm{open}_3})] \times \mathbb{1}_{\{(\pi_{3,1}^2)_{\mathrm{goal}}'\}}(a) \qquad (E.21)$$
$$+ [\mathbb{1}_{\Omega_{\mathrm{room}_4} \cup \{\mathrm{door}_2\}}(s_{\mathrm{cur}}) + \mathbb{1}_{\Omega_{\mathrm{room}_2} \cup \{\mathrm{door}_1\}}(s_{\mathrm{cur}}) \times \mathbb{1}_{\{1\}}(s_{\mathrm{open}_2})$$
$$+ \mathbb{1}_{\Omega_{\mathrm{room}_1}}(s_{\mathrm{cur}}) \times \mathbb{1}_{\{1\}}(s_{\mathrm{open}_1}) \times \mathbb{1}_{\{1\}}(s_{\mathrm{open}_2})]$$
$$\times \mathbb{1}_{\{0\}}(s_{\mathrm{open}_3}) \times \mathbb{1}_{\{(\pi_{3,1}^2)_{\mathrm{door}_3}'\}}(a)$$
$$+ [\mathbb{1}_{\Omega_{\mathrm{room}_2} \cup \{\mathrm{door}_1\}}(s_{\mathrm{cur}}) + \mathbb{1}_{\Omega_{\mathrm{room}_1}}(s_{\mathrm{cur}}) \times \mathbb{1}_{\{1\}}(s_{\mathrm{open}_1})] \times \mathbb{1}_{\{0\}}(s_{\mathrm{open}_2}) \times \mathbb{1}_{\{(\pi_{3,1}^2)_{\mathrm{door}_2}'\}}(a)$$
$$+ \mathbb{1}_{\Omega_{\mathrm{room}_1}}(s_{\mathrm{cur}}) \times \mathbb{1}_{\{0\}}(s_{\mathrm{open}_1}) \times \mathbb{1}_{\{(\pi_{3,1}^2)_{\mathrm{door}_1}'\}}(a).$$

### E.1.5 Embeddings, embedding generators and skills

For $\mathrm{MDP}_{2,1}$:

$E^1_{2,1} : \Theta^1_{2,1} \to \{(e^1_{2,1})_\theta : \theta \in \Theta^1_{2,1}\}$ with $\Theta^1_{2,1} := \{\mathtt{door}_1, \mathtt{door}_2, \mathtt{door}_3, \mathtt{dest}\}$ is defined as

$$E^1_{2,1}(\theta) := (e^1_{2,1})_\theta \,, \tag{E.22}$$

with $(e^1_{2,1})_\theta : \mathcal{SA}^1_{2,1} \to \{0, 1\}$ ($\theta = \mathtt{door}_1, \mathtt{door}_2, \mathtt{door}_3, \mathtt{dest}$) being defined as

$$(e^1_{2,1})_\theta((s_{\mathrm{cur}}, s_{\mathrm{dest}}), a) := (s_{\mathrm{cur}}, s_\theta, a) \,.$$

$(e_{\mathrm{decomp}})^1_{2,1} : \mathcal{SA}^1_{2,1} \to \{(s_{\mathrm{cur}}, s_{\mathrm{dest}}, a) : s_{\mathrm{cur}}, s_{\mathrm{dest}} \in \Omega^c_{\mathtt{blocks}}, a \in \mathcal{A}_{\mathrm{dir}} \cup \{a^{\mathrm{end}}\}\}$ is defined as:

$$(e_{\mathrm{decomp}})^1_{2,1}((s_{\mathrm{cur}}, s_{\mathrm{dest}}), a) := (s_{\mathrm{cur}}, s_{\mathrm{dest}}, a) \,. \tag{E.23}$$

$\overline{\pi}^{\mathrm{nav}} : (e_{\mathrm{decomp}})^1_{2,1}(\mathcal{SA}^1_{2,1}) \to [0, 1]$ is as follows:

$$\overline{\pi}^{\mathrm{nav}}(s_{\mathrm{cur}}, s_{\mathrm{dest}}, a) = \pi^1_{2,1,*}((s_{\mathrm{cur}}, s_{\mathrm{dest}}), a) \,. \tag{E.24}$$

For $\mathrm{MDP}_{2,2}$:

$(E^1_{2,2})_\beta : (\Theta^1_{2,2})_\beta \to \{(e^1_{2,2})^\beta_\theta : \theta \in (\Theta^1_{2,2})_\beta\}$ with $(\Theta^1_{2,2})_\beta := \{\mathtt{key}, \mathtt{door}\}$ is defined as

$$(E^1_{2,2})_\beta(\theta) := (e^1_{2,2})^\beta_\theta \,, \tag{E.25}$$

with $(e^1_{2,2})^\beta_\theta : \{((s_{\mathrm{cur}}, s_{\mathrm{pick}}, s_{\mathrm{open}}), a) \in \mathcal{S}_{2,2} \times (\mathcal{A}_{\mathrm{dir}} \cup \{a^{\mathrm{end}}\}) : s_{\mathrm{cur}} + a \neq s_{\mathrm{door}}\} \subseteq \mathcal{SA}^1_{2,2} \to$ $\{(s_{\mathrm{cur}}, s_{\mathrm{goal}}, a) : s_{\mathrm{cur}}, s_{\mathrm{goal}} \in \Omega^c_{\mathtt{blocks}}, a \in \mathcal{A}_{\mathrm{dir}} \cup \{a^{\mathrm{end}}\}\}$ ($\theta \in (\Theta^1_{2,2})_\beta$) being defined as

$$(e^1_{2,2})^\beta_\theta((s_{\mathrm{cur}}, s_{\mathrm{pick}}, s_{\mathrm{open}}), a) := (s_{\mathrm{cur}}, s_\theta, a) \,.$$

$(e_{\mathrm{decomp}})^2_{2,2}((s_{\mathrm{cur}}, s_{\mathrm{key}}, s_{\mathrm{door}}, s_{\mathrm{pick}}, s_{\mathrm{open}}), a)$

$$:= \begin{cases} (\mathbb{1}_{\{s_{\mathrm{key}}\}}(s_{\mathrm{cur}}), \mathbb{1}_{\{1\}}(s_{\mathrm{pick}}) \times \mathbb{1}_{\mathcal{N}(s_{\mathrm{door}})}(s_{\mathrm{cur}}), s_{\mathrm{pick}}, s_{\mathrm{open}}, \pi_\theta) & , \text{if } a = (\pi^1_{2,2})^\lambda_\theta \\ (\mathbb{1}_{\{s_{\mathrm{key}}\}}(s_{\mathrm{cur}}), \mathbb{1}_{\{1\}}(s_{\mathrm{pick}}) \times \mathbb{1}_{\mathcal{N}(s_{\mathrm{door}})}(s_{\mathrm{cur}}), s_{\mathrm{pick}}, s_{\mathrm{open}}, a) & , \text{otherwise} \end{cases} \,. \tag{E.26}$$

$\overline{\pi}^{\mathrm{concat}}(e_{s_{\mathrm{cur}}=s_{\mathrm{key}}}, e_{\|s_{\mathrm{cur}}-s_{\mathrm{door}}\|_1=1}, s_{\mathrm{pick}}, s_{\mathrm{open}}, a)$

$= \mathbb{1}_{\{1\}}(s_{\mathrm{open}}) \times \mathbb{1}_{\{a^{\mathrm{end}}\}}(a) + \mathbb{1}_{\{0\}}(s_{\mathrm{open}}) \times \mathbb{1}_{\{0\}}(s_{\mathrm{pick}})$ $\tag{E.27}$

$\times \left[ e_{s_{\mathrm{cur}}=s_{\mathrm{key}}} \times \mathbb{1}_{\{\pi_{\mathrm{pick}}\}}(a) + (1 - e_{s_{\mathrm{cur}}=s_{\mathrm{key}}}) \times \mathbb{1}_{\{\pi_{\mathrm{key}}\}}(a) \right]$

$+ \mathbb{1}_{\{0\}}(s_{\mathrm{open}}) \times \left[ e_{\|s_{\mathrm{cur}}-s_{\mathrm{door}}\|_1=1} \times \mathbb{1}_{\{\pi_{\mathrm{open}}\}}(a) + (\mathbb{1}_{\{1\}}(s_{\mathrm{pick}}) - e_{\|s_{\mathrm{cur}}-s_{\mathrm{door}}\|_1=1}) \times \mathbb{1}_{\{\pi_{\mathrm{door}}\}}(a) \right] \,.$

For $\mathrm{MDP}_{3,1}$:

$(E^1_{3,1})_\alpha : (\Theta^1_{3,1})_\alpha \to \{(e^1_{3,1})^\alpha_\theta : \theta \in (\Theta^1_{3,1})_\alpha\}$ with $(\Theta^1_{3,1})_\alpha := \{\mathtt{pick}, \mathtt{open}\}$ is defined as

$$(E^1_{3,1})_\alpha(\theta) := (e^1_{3,1})^\alpha_\theta \,, \tag{E.28}$$

with $(e^1_{3,1})^\alpha_\theta : \mathcal{SA}^1_{3,1} \to \{0, 1\}$ ($\theta = \mathtt{pick}, \mathtt{open}$) being defined as

$$(e^1_{3,1})^\alpha_\theta((s_{\mathrm{cur}}, s_{\mathbf{pick}}, s_{\mathbf{open}}, s_{\mathrm{done}}), a) := \mathbb{1}_{\{\theta\}}(a) \,.$$

$(E^1_{3,1})_\beta : (\Theta^1_{3,1})_\beta \to \{(e^1_{3,1})^\beta_\theta : \theta \in (\Theta^1_{3,1})_\beta\}$ with $(\Theta^1_{3,1})_\beta := \{\mathtt{key}_1, \mathtt{key}_2, \mathtt{key}_3, \mathtt{door}_1, \mathtt{door}_2, \mathtt{door}_3, \mathtt{goal}\}$ is defined as

$$(E^1_{3,1})_\beta(\theta) := (e^1_{3,1})^\beta_\theta \,, \tag{E.29}$$

with $(e_{3,1}^1)_\theta^\beta : \mathcal{S}_{3,1} \times (\mathcal{A}_{\text{dir}} \cup \{a^{\text{end}}\}) \subseteq \mathcal{SA}_{3,1}^1 \to \{(s_{\text{cur}}, s_{\text{dest}}, a) : s_{\text{cur}}, s_{\text{dest}} \in \Omega_{\text{blocks}}^c, a \in \mathcal{A}_{\text{dir}} \cup \{a^{\text{end}}\}\}$ $(\theta = \text{key}_1, \text{key}_2, \text{key}_3, \text{goal})$ being defined as

$$(e_{3,1}^1)_\theta^\beta((s_{\text{cur}}, s_{\textbf{pick}}, s_{\textbf{open}}, s_{\text{done}}), a) := (s_{\text{cur}}, s_\theta, a),$$

and $(e_{3,1}^1)_\theta^\beta : \{((s_{\text{cur}}, s_{\textbf{pick}}, s_{\textbf{open}}, s_{\text{done}}), a) \in \mathcal{S}_{3,1} \times (\mathcal{A}_{\text{dir}} \cup \{a^{\text{end}}\}) : s_{\text{cur}} + a \neq s_\theta\} \subseteq \mathcal{SA}_{3,1}^1 \to \{(s_{\text{cur}}, s_{\text{dest}}, a) : s_{\text{cur}}, s_{\text{dest}} \in \Omega_{\text{blocks}}^c, a \in \mathcal{A}_{\text{dir}} \cup \{a^{\text{end}}\}\}$ $(\theta = \text{door}_1, \text{door}_2, \text{door}_3)$ being defined as

$$(e_{3,1}^1)_\theta^\beta((s_{\text{cur}}, s_{\textbf{pick}}, s_{\textbf{open}}, s_{\text{done}}), a) := (s_{\text{cur}}, s_\theta, a).$$

$E_{3,1}^2 : \Theta_{3,1}^2 \to \{(e_{3,1}^2)_\theta : \theta = (\theta_{\text{key}}, \theta_{\text{door}}) \in \Theta_{3,1}^2\}$ with $\Theta_{3,1}^2 := \{(\text{key}_1, \text{door}_1), (\text{key}_2, \text{door}_2), (\text{key}_3, \text{door}_3), (\text{goal}, \text{goal})\}$ is defined as

$$E_{3,1}^2(\theta) := (e_{3,1}^2)_\theta, \tag{E.30}$$

and for $\theta = (\text{key}_i, \text{door}_i)$ $(i = 1, 2, 3)$, $(e_{3,1}^2)_\theta : \mathcal{S}_{3,1} \times (\{(\pi_{3,1}^1)_{\text{pick}}^\alpha, (\pi_{3,1}^1)_{\text{open}}^\alpha, (\pi_{3,1}^1)_{\text{key}_i}^\beta, (\pi_{3,1}^1)_{\text{door}_i}^\beta, a^{\text{end}}\}) \subseteq \mathcal{SA}_{3,1}^2 \to \{0,1\}^4 \times \{\pi_{\text{pick}}, \pi_{\text{open}}, \pi_{\text{key}}, \pi_{\text{door}}, a^{\text{end}}\}\}$ are defined as

$(e_{3,1}^2)_\theta((s_{\text{cur}}, s_{\textbf{pick}}, s_{\textbf{open}}, s_{\text{done}}), a)$

$$:= \begin{cases} (\mathbb{1}_{\{s_{\text{key}_i}\}}(s_{\text{cur}}), \mathbb{1}_{\{1\}}(s_{\text{pick}_i}) \times \mathbb{1}_{\mathcal{N}(s_{\text{door}_i})}(s_{\text{cur}}), s_{\text{pick}_i}, s_{\text{open}_i}, \pi_\theta) & , \text{if } a = (\pi_{3,1}^1)_\theta^\alpha \\ (\mathbb{1}_{\{s_{\text{key}_i}\}}(s_{\text{cur}}), \mathbb{1}_{\{1\}}(s_{\text{pick}_i}) \times \mathbb{1}_{\mathcal{N}(s_{\text{door}_i})}(s_{\text{cur}}), s_{\text{pick}_i}, s_{\text{open}_i}, \pi_{\text{key}}) & , \text{if } a = (\pi_{3,1}^1)_{\text{key}_i}^\beta \\ (\mathbb{1}_{\{s_{\text{key}_i}\}}(s_{\text{cur}}), \mathbb{1}_{\{1\}}(s_{\text{pick}_i}) \times \mathbb{1}_{\mathcal{N}(s_{\text{door}_i})}(s_{\text{cur}}), s_{\text{pick}_i}, s_{\text{open}_i}, \pi_{\text{door}}) & , \text{if } a = (\pi_{3,1}^1)_{\text{door}_i}^\beta \\ (\mathbb{1}_{\{s_{\text{key}_i}\}}(s_{\text{cur}}), \mathbb{1}_{\{1\}}(s_{\text{pick}_i}) \times \mathbb{1}_{\mathcal{N}(s_{\text{door}_i})}(s_{\text{cur}}), s_{\text{pick}_i}, s_{\text{open}_i}, a) & , \text{otherwise} \end{cases},$$

for $\theta = (\text{goal}, \text{goal})$, $(e_{3,1}^2)_\theta : \mathcal{S}_{3,1} \times (\{(\pi_{3,1}^1)_{\text{pick}}^\alpha, (\pi_{3,1}^1)_{\text{open}}^\alpha, (\pi_{3,1}^1)_{\text{goal}}^\beta, a^{\text{end}}\}) \subseteq \mathcal{SA}_{3,1}^2 \to \{0,1\}^4 \times \{\pi_{\text{pick}}, \pi_{\text{open}}, \pi_{\text{key}}, a^{\text{end}}\}\}$ is defined as

$(e_{3,1}^2)_\theta((s_{\text{cur}}, s_{\textbf{pick}}, s_{\textbf{open}}, s_{\text{done}}), a)$

$$:= \begin{cases} (\mathbb{1}_{\{s_{\text{goal}}\}}(s_{\text{cur}}), \mathbb{1}_{\{1\}}(s_{\text{done}}) \times \mathbb{1}_{\{s_{\text{goal}}\}}(s_{\text{cur}}), s_{\text{done}}, s_{\text{done}}, \pi_\theta) & , \text{if } a = (\pi_{3,1}^1)_\theta^\alpha \\ (\mathbb{1}_{\{s_{\text{goal}}\}}(s_{\text{cur}}), \mathbb{1}_{\{1\}}(s_{\text{done}}) \times \mathbb{1}_{\{s_{\text{goal}}\}}(s_{\text{cur}}), s_{\text{done}}, s_{\text{done}}, \pi_{\text{key}}) & , \text{if } a = (\pi_{3,1}^1)_{\text{goal}}^\beta \\ (\mathbb{1}_{\{s_{\text{goal}}\}}(s_{\text{cur}}), \mathbb{1}_{\{1\}}(s_{\text{done}}) \times \mathbb{1}_{\{s_{\text{goal}}\}}(s_{\text{cur}}), s_{\text{done}}, s_{\text{done}}, a) & , \text{otherwise} \end{cases}.$$

### E.1.6 ANALYTICAL RUNNING OF ALGORITHMS

The following goes through how Algs. 5–6 solve these MDPs by constructing MMDPs. All timescales here are $t_{\min} = t_{\max} = +\infty$. See Apps. E.1.2–E.1.5 for the equations needed here as well as the mathematical notations mentioned here.

$\text{MDP}_{1,1}$ is exactly $\text{MDP}_{\text{dense}}^{\text{nav}}$ in the example of navigation and transportation with traffic jams. In particular, we use the navigation skill $\overline{\pi}_{\text{dense}}^{\text{nav}}$ extracted there, which we assume to be deterministic to simplify the calculations of compressed MDPs and better illustrate how the MMDP is constructed. This skill is then added to the set of public skills $Skills$.

$\text{MDP}_{2,1} := (\mathcal{S}_{2,1}, \mathcal{S}_{2,1}^{\text{init}}, \mathcal{S}_{2,1}^{\text{end}}, \mathcal{A}_{2,1}, P_{2,1}, R_{2,1}, \Gamma_{2,1})$, defined in equation E.1, models navigation through $\Omega$ while avoiding blocks assuming all the doors are open.

Because $\text{MDP}_{2,1}$ is of difficulty 2, the student constructs a two-level MMDP to solve it, and for the first level, $\text{MDP}_{2,1}^1 := (\mathcal{S}_{2,1}, (\mathcal{S}_{2,1}^{\text{init}})^1, (\mathcal{S}_{2,1}^{\text{end}})^1, \mathcal{A}_{2,1}^1, P_{2,1}^1, R_{2,1}^1, \Gamma_{2,1}^1) = (\mathcal{S}_{2,1}, \mathcal{S}_{2,1}^{\text{init}}, \mathcal{S}_{2,1}^{\text{end}}, \mathcal{A}_{2,1}, P_{2,1}, R_{2,1}, \Gamma_{2,1})$. To construct the second-level MDP, the teacher provides the following information: $\mathcal{G}_{2,1}^1 := \{(\overline{\pi}_{\text{dense}}^{\text{nav}}, E_{2,1}^1)\}$, where $E_{2,1}^1$ is defined in equation E.22. For clarification, the teacher does not know the skill $\overline{\pi}_{\text{dense}}^{\text{nav}}$, but sees the set $Skills$ and can refer to its elements, for example here to provide the students with hints about using $\overline{\pi}_{\text{dense}}^{\text{nav}}$ for constructing $\Pi_{2,1}^1$. Similar considerations will apply to all examples in this paper.

Then, the student constructs the policy generators $g_{2,1}^1$ from $(\overline{\pi}_{\text{dense}}^{\text{nav}}, E_{2,1}^1)$, where $g_{2,1}^1$ is as defined in equation E.8, with timescales $t_{(\pi_{2,1}^1)_\theta} = t_{g_{2,1}^1} = +\infty$. In particular, notice here $(\pi_{2,1}^1)_\theta$ takes the

action $a^{\text{end}}$ with probability one at states satisfying $s_{\text{cur}} = s_\theta$, telling the agent it could stop the current policy because it has reached the current subgoal and should move on to the next one. This shows why forcing a policy $\pi$ to satisfy $\pi(s,a) = \mathbb{1}_{\{a^{\text{end}}\}}(a)$ for any $s \in \mathcal{S}^{\text{end}}$ is important in the construction of MMDPs and policy transfer.

To summarize, with the help of the teacher, the student concludes that

$$\Pi^1_{2,1} = \{(\pi^1_{2,1})_\theta : \theta \in \Theta^1_{2,1}\},$$

where $(\pi^1_{2,1})_\theta$ $(\theta \in \Theta^1_{2,1})$ all have timescales $+\infty$.

Then, the student constructs the second level MDP $\text{MDP}^2_{2,1} = (\mathcal{S}_{2,1}, \mathcal{S}^{\text{init}}_{2,1}, \mathcal{S}^{\text{end}}_{2,1}, \overline{\Pi^1_{2,1}}, P^2_{2,1}, R^2_{2,1}, \Gamma^2_{2,1})$, with $P^2_{2,1}$, $R^2_{2,1}$, $\Gamma^2_{2,1}$ as in equation E.3. Notice that if $\overline{\pi}^{\text{nav}}_{\text{dense}}$ is not deterministic, then the sequence defined here is stochastic, making the calculations below more complicated.

Then, the student solves $\text{MDP}^2_{2,1}$ and finds its optimal policy $\pi^2_{2,1,*}$ with timescale $t_{\pi^2_{2,1,*}} = +\infty$. The essential meaning of $\pi^2_{2,1,*}$ is that since there are multiple rooms separated by blocks and doors in $\Omega$, the student needs to walk across several rooms when going from his initial location to the target location by surpassing the blocks around the route, but the navigation skill $\overline{\pi}^{\text{nav}}_{\text{dense}}$ only teaches the student to navigate within a single room while avoiding blocks and doors, so the student may need to stitch $\overline{\pi}^{\text{nav}}_{\text{dense}}$ for multiple times with the middle locations being the locations of the doors. For instance, when going from some location in $\text{room}_2$ to some location in $\text{room}_4$, the student needs to first go to $\text{door}_1$, then $\text{door}_2$, then $\text{door}_3$ in the middle, because there are no doors connecting between $\text{room}_2$ and $\text{room}_4$. The student could focus on learning this higher-order function at level 2 because the details of going from A to B within a single room while avoiding blocks and doors has been encapsulated by the navigation skill $\overline{\pi}^{\text{nav}}_{\text{dense}}$. Consequently, the student derives the optimal policy $\pi_{2,1,*} = \pi^2_{2,1,*}$ for $\text{MDP}_{2,1}$ without any more iterations. The reason is that $\Omega_{\text{blocks}} \cup \{\text{door}_i : 1 \le i \le 3\}$ here is contained in $\Omega'_{\text{jams}}$ in $\text{MDP}^{\text{nav}}_{\text{dense}}$ per our previous assumption, and the light traffic will repel the student to not touch $\Omega_{\text{blocks}} \cup \{\text{door}_i : 1 \le i \le 3\}$ whenever the total distance the student needs to travel does not increase.

Next, if the teacher provides the embedding $(e_{\text{decomp}})^1_{2,1}$ as defined in equation E.23, then the assistant extracts $\overline{\pi}^{\text{nav}}$, a skill of navigation through $\Omega$ while avoiding blocks assuming all the doors are open, as in equation E.24, with timescale $t_{\overline{\pi}^{\text{nav}}} = t_{\pi^1_{2,1,*}} = +\infty$. Similar to $\overline{\pi}^{\text{nav}}_{\text{dense}}$, we assume hereafter $\overline{\pi}^{\text{nav}}$ is deterministic to simplify the calculations of compressed MDPs and better illustrate how the MMDP is constructed without losing too much generality.

Now we move to $\text{MDP}_{2,2}$, which we have discussed thoroughly in the previous boxes. It is related to picking up a key and opening the door, and see equation E.4 for its definition. From this MDP, the assistant extracts a concatenation skill $\overline{\pi}^{\text{concat}}$.

Now we move to the target MDP of difficulty 3. For $\text{MDP}_{3,1} := (\mathcal{S}_{3,1}, \mathcal{S}^{\text{init}}_{3,1}, \mathcal{S}^{\text{end}}_{3,1}, \mathcal{A}_{3,1}, P_{3,1}, R_{3,1}, \Gamma_{3,1})$ related to picking up the goal, we have, for $s = (s_{\text{cur}}, s_{\text{pick}}, s_{\text{open}}, s_{\text{done}})$ (and similarly for $s'$), it is defined as in equation E.2.

Because $\text{MDP}_{3,1}$ is of difficulty 3, the student constructs a three-level MMDP to solve it, and for the first level, $\text{MDP}^1_{3,1} := (\mathcal{S}^1_{3,1}, (\mathcal{S}^{\text{init}}_{3,1})^1, (\mathcal{S}^{\text{end}}_{3,1})^1, \mathcal{A}^1_{3,1}, P^1_{3,1}, R^1_{3,1}, \Gamma^1_{3,1}) = (\mathcal{S}_{3,1}, \mathcal{S}^{\text{init}}_{3,1}, \mathcal{S}^{\text{end}}_{3,1}, \mathcal{A}_{3,1}, P_{3,1}, R_{3,1}, \Gamma_{3,1})$. To construct the second-level MDP, the teacher provides the following information:

$$\mathcal{G}^1_{3,1} := \{(\text{id}, (E^1_{3,1})_\alpha), (\overline{\pi}^{\text{nav}}, (E^1_{3,1})_\beta)\},$$

where $(E^1_{3,1})_\alpha$ as defined in equation E.28, and $(E^1_{3,1})_\beta$ as defined in equation E.29.

The student then composes the policy generators $(g^1_{3,1})_\alpha$, $(g^1_{3,1})_\beta$ coming from $(\text{id}, (E^1_{3,1})_\alpha)$, $(\overline{\pi}^{\text{nav}}, (E^1_{3,1})_\beta)$ respectively, where $(g^1_{3,1})_\alpha$ is defined in equation E.15, with timescale $t_{(g^1_{3,1})_\alpha} = 1$, and here $(\pi^1_{3,1})^\alpha_\theta (\theta = \text{pick}, \text{open})$, have timescales $t_{(\pi^1_{3,1})^\alpha_\theta} = t_{(g^1_{3,1})_\alpha} = 1$.

$(g^1_{3,1})_\beta$ is as defined in equation E.16, with timescale $t_{(g^1_{3,1})_\beta} = t_{\overline{\pi}^{\text{nav}}} = +\infty$, and here for all $\theta \in (\Theta^1_{3,1})_\beta$, the timescales $t_{(\pi^1_{3,1})^\beta_\theta} = t_{\overline{\pi}^{\text{nav}}} = +\infty$. As a reminder, here we use $\overline{\pi}^{\text{nav}} \circ (e^1_{3,1})^\beta_\theta$ to

represent the composite partial policy coming from the skill $\overline{\pi}^{\mathrm{nav}}$ and the embedding $(e_{3,1}^1)_\theta^\beta$, which is a slight abuse of notations, because according to Def. B.1, it is not exactly the output of function composition: there is a second extra step of normalization. Similar notations apply hereafter.

To summarize, with the help of the teacher, the student concludes that

$$\Pi_{3,1}^1 = \{(\pi_{3,1}^1)_\theta^\alpha : \theta \in (\Theta_{3,1}^1)_\alpha\} \cup \{(\pi_{3,1}^1)_\theta^\beta : \theta \in (\Theta_{3,1}^1)_\beta\},$$

where $(\pi_{3,1}^1)_\theta^\alpha$ $(\theta \in (\Theta_{3,1}^1)_\alpha)$ are two degenerate extended policies with timescales one, and $(\pi_{3,1}^1)_\theta^\beta$ $(\theta \in (\Theta_{3,1}^1)_\beta)$ are seven extended policies with timescales $+\infty$. In addition, all the policies in $\Pi_{3,1}^1$ could be represented by an element in the product set $((\Theta_{3,1}^1)_\alpha \cup \{a^{\mathrm{end}}, \texttt{null}\}) \times ((\Theta_{3,1}^1)_\beta \cup \{a^{\mathrm{end}}, \texttt{null}\})$: for instance, $(\pi_{3,1}^1)_{\texttt{pick}}^\alpha$ is represented by $(\texttt{pick}, \texttt{null})$. So, $\Pi_{3,1}^1$ has two action factors, and for simplicity, we index them as $\alpha, \beta$ respectively.

Then, the student constructs the second level MDP $\mathrm{MDP}_{3,1}^2 = (\mathcal{S}_{3,1}, \mathcal{S}_{3,1}^{\mathrm{init}}, \mathcal{S}_{3,1}^{\mathrm{end}}, \overline{\Pi_{3,1}^1}, P_{3,1}^2, R_{3,1}^2, \Gamma_{3,1}^2)$, where $P_{3,1}^2, R_{3,1}^2$, and $\Gamma_{3,1}^2$ are as in equation E.5.

To construct the third-level MDP, the teacher provides the following information:

$$\mathcal{G}_{3,1}^2 := \{(\overline{\pi}^{\mathrm{concat}}, E_{3,1}^2)\},$$

where $E_{3,1}^2$ is as defined in equation E.30.

The student then derives that the policy generator $g_{3,1}^2 |_{\{\alpha,\beta\}}$ coming from $(\overline{\pi}^{\mathrm{concat}}, E_{3,1}^2)$, where $g_{3,1}^2 |_{\{\alpha,\beta\}}$ is as defined in equation E.17, with timescale $t_{g_{3,1}^2|_{\{\alpha,\beta\}}} = t_{\overline{\pi}^{\mathrm{concat}}} = +\infty$, and here for all $\theta \in \Theta_{3,1}^2$, the timescales $t_{(\pi_{3,1}^2)_\theta|_{\{\alpha,\beta\}}} = t_{\overline{\pi}^{\mathrm{concat}}} = +\infty$.

To summarize, with the help of the teacher, the student concludes that

$$\Pi_{3,1}^2 = \{(\pi_{3,1}^2)_\theta |_{\{\alpha,\beta\}} : \theta \in \Theta_{3,1}^2\},$$

wherein all the four policies have timescales $+\infty$. Here, we use the two skills, the navigation skill $\overline{\pi}^{\mathrm{nav}}$ and the concatenation logic used behind opening a door $\overline{\pi}^{\mathrm{concat}}$, to generate by their combinatorial combinations many new policies we do not see directly previously. This is the reason why we could not combine the two skills at a single level, and have to go to the third level.

Then, the student constructs the third-level MDP $\mathrm{MDP}_{3,1}^3 = (\mathcal{S}_{3,1}, \mathcal{S}_{3,1}^{\mathrm{init}}, \mathcal{S}_{3,1}^{\mathrm{end}}, \overline{\Pi_{3,1}^2}, P_{3,1}^3, R_{3,1}^3, \Gamma_{3,1}^3)$, where $P_{3,1}^3, R_{3,1}^3, \Gamma_{3,1}^3$ are as in equation E.7.

Then, the student solves $\mathrm{MDP}_{3,1}^3$ and finds its optimal policy $\pi_{3,1,*}^3$ as in equation E.18, with timescale $+\infty$. For simplicity, we do not guarantee the functions (such as policies, and more generally, partial policies) written down are correct at states never explored unless specified, and in particular the normalization property may not be satisfied at those states. This applies to both $\pi_{3,1,*}^3$ here and such functions elsewhere. Consequently, the student derives $\pi_{3,1,*}^2$ as in equation E.19. and thus, the student concludes $\pi_{3,1,*}$ as in equation E.20.

Regarding the skill we could learn from $\pi_{3,1,*}^3$, same as $\overline{\pi}^{\mathrm{nav}}$, essentially it is about finding the shortest path, where this time each room is a vertex in the graph, and there are edges between two rooms if there is at least one openable door connecting them. To avoid adding the complexity of algorithms in learning the shortest path, which is not the main point of this example, we do not learn this skill.

## E.2 NAVIGATION AND TRANSPORTATION WITH TRAFFIC JAMS

### E.2.1 FORMAL DEFINITION OF MDPS

The formal definition of the family of target MDPs is as follows: $\mathrm{MDP}_\kappa :=$ $(\mathcal{S}, \mathcal{S}^{\mathrm{init}}, \mathcal{S}^{\mathrm{end}}, \mathcal{A}, P, R_\kappa, \Gamma)$, for $\kappa$ in some finite set $\mathcal{K}$ ($|\mathcal{K}| = 6$), with

$$
\begin{aligned}
&\mathcal{S} := \{(s_{\mathrm{cur}}, s_{\mathrm{dest}}) : s_{\mathrm{cur}}, s_{\mathrm{dest}} \in \Omega\} = \Omega \times \Omega, \\
&\mathcal{S}^{\mathrm{init}} := \{(s_{\mathrm{cur}}, s_{\mathrm{dest}}) \in \mathcal{S} : s_{\mathrm{cur}} \neq s_{\mathrm{dest}}\}, \quad \mathcal{S}^{\mathrm{end}} := \{(s_{\mathrm{cur}}, s_{\mathrm{dest}}) \in \mathcal{S} : s_{\mathrm{cur}} = s_{\mathrm{dest}}\}, \\
&\mathcal{A} := (\mathcal{A}_{\mathrm{dir}} \cup \{a^{\mathrm{end}}\}) \times (\mathcal{A}_{\mathrm{means}} \cup \{a^{\mathrm{end}}\}) \\
&P((s_{\mathrm{cur}}, s_{\mathrm{dest}}), (a_{\mathrm{dir}}, a_{\mathrm{means}}), (s'_{\mathrm{cur}}, s'_{\mathrm{dest}})) \\
&:= \mathbb{1}_{\{s_{\mathrm{dest}}\}}(s'_{\mathrm{dest}}) \times \left[[1 - \mathbb{1}_\Omega(s_{\mathrm{cur}} + a_{\mathrm{dir}})] \times \mathbb{1}_{\{s_{\mathrm{cur}}\}}(s'_{\mathrm{cur}}) \right. \\
&\left. \quad + \mathbb{1}_\Omega(s_{\mathrm{cur}} + a_{\mathrm{dir}}) \times [p_s \times \mathbb{1}_{\{s_{\mathrm{cur}}+a_{\mathrm{dir}}\}}(s'_{\mathrm{cur}}) + (1 - p_s) \times \mathbb{1}_{\{s_{\mathrm{cur}}\}}(s'_{\mathrm{cur}})]\right], \\
&R_\kappa((s_{\mathrm{cur}}, s_{\mathrm{dest}}), (a_{\mathrm{dir}}, a_{\mathrm{means}}), (s'_{\mathrm{cur}}, s'_{\mathrm{dest}})) \\
&:= R_{\mathrm{dest}} \times \mathbb{1}_{\{s'_{\mathrm{dest}}\}}(s'_{\mathrm{cur}}) \\
&\quad + \left[(R_{\mathrm{jams}} + \frac{r_0}{v_{\mathrm{mc}}}) \times \mathbb{1}_{\{\mathrm{mc}\}}(a_{\mathrm{means}}) + \frac{r_0}{\kappa} \times \mathbb{1}_{\{\mathrm{car}\}}(a_{\mathrm{means}})\right] \\
&\quad \times \left[1 - \mathbb{1}_{\Omega^c_{\mathrm{jams}}}(s_{\mathrm{cur}}) \times \mathbb{1}_{\Omega^c_{\mathrm{jams}}}(s'_{\mathrm{cur}})\right] \\
&\quad + \left[\frac{r_0}{v_{\mathrm{mc}}} \times \mathbb{1}_{\{\mathrm{mc}\}}(a_{\mathrm{means}}) + \frac{r_0}{v_{\mathrm{car}}} \times \mathbb{1}_{\{\mathrm{car}\}}(a_{\mathrm{means}})\right] \times \mathbb{1}_{\Omega^c_{\mathrm{jams}}}(s_{\mathrm{cur}}) \times \mathbb{1}_{\Omega^c_{\mathrm{jams}}}(s'_{\mathrm{cur}}), \\
&\Gamma((s_{\mathrm{cur}}, s_{\mathrm{dest}}), (a_{\mathrm{dir}}, a_{\mathrm{means}}), (s'_{\mathrm{cur}}, s'_{\mathrm{dest}})) := \gamma_0 .
\end{aligned}
$$

(E.31)

For clarification, the "+" signs here in expressions such as $s_{\mathrm{cur}} + a_{\mathrm{dir}}$ are just vector sums in the grid world. Also, for the purpose of simplicity only, we set $x + a^{\mathrm{end}} = a^{\mathrm{end}}$ for any numerical value $x$ throughout, such as in the transition probabilities here. Also for brevity, we do not guarantee the functions (such as transition probabilities, rewards, discount factors, policies, and their variants) written down are defined and correct at $s \in \mathcal{S}^{\mathrm{end}}$ or $s \notin \mathcal{S}^{\mathrm{end}}$, $a \in \mathcal{A}^{\mathbf{end}}$ unless specified. These apply to all the examples. The teacher has the role to set all the parameters of the problems: $0 < p_s \leq 1$, is the probability for any action to succeed, and here we set $p_s = 0.9$; $R_{\mathrm{dest}} > 0$ is some large positive reward (here $R_{\mathrm{dest}} = 10^4$) set for reaching $s_{\mathrm{dest}}$; $R_{\mathrm{jams}} < \frac{r_0}{\kappa} - \frac{r_0}{v_{\mathrm{mc}}}$ is some large negative reward (here $R_{\mathrm{jams}} = -10^3$) if the agent drives along, enters, or leaves $\Omega_{\mathrm{jams}}$ using mc; $0 < \gamma_0 < 1$ is some large discount factor (here $\gamma_0 = 0.999$). $1/\kappa = 2.4, 2.8, 3.2, 3.6, 4, 4.4$ respectively for $\{\mathrm{MDP}_\kappa\}_{\kappa \in \mathcal{K}}$, modeling the dependence of the speed of the car in traffic regions as a function of the heaviness of traffic. As a summary, Table 2 lists all the values of the parameters of the MDPs in this example.

Table 2: Parameters in the example of navigation and transportation with traffic jams

| | $\Omega_{\mathrm{jams}}$ | $p_s$ | $v_{\mathrm{mc}}$ | $v_{\mathrm{car}}$ | $\kappa$ | $R_{\mathrm{dest}}$ | $R_{\mathrm{jams}}$ | $r_0$ | $\gamma_0$ |
|---|---|---|---|---|---|---|---|---|---|
| $\mathrm{MDP}_{2,1}$ | $[1,15] \times \{6\} \cup \{5\} \times [1,8]$ | 0.9 | 1 | 0.6 | $1/2.4$ | $10^4$ | $-10^3$ | $-10$ | 0.999 |
| $\mathrm{MDP}_{2,2}$ | $[1,15] \times \{6\} \cup \{5\} \times [1,8]$ | 0.9 | 1 | 0.6 | $1/2.8$ | $10^4$ | $-10^3$ | $-10$ | 0.999 |
| $\mathrm{MDP}_{2,3}$ | $[1,15] \times \{6\} \cup \{5\} \times [1,8]$ | 0.9 | 1 | 0.6 | $1/3.2$ | $10^4$ | $-10^3$ | $-10$ | 0.999 |
| $\mathrm{MDP}_{2,4}$ | $[1,15] \times \{6\} \cup \{5\} \times [1,8]$ | 0.9 | 1 | 0.6 | $1/3.6$ | $10^4$ | $-10^3$ | $-10$ | 0.999 |
| $\mathrm{MDP}_{2,5}$ | $[1,15] \times \{6\} \cup \{5\} \times [1,8]$ | 0.9 | 1 | 0.6 | $1/4$ | $10^4$ | $-10^3$ | $-10$ | 0.999 |
| $\mathrm{MDP}_{2,6}$ | $[1,15] \times \{6\} \cup \{5\} \times [1,8]$ | 0.9 | 1 | 0.6 | $1/4.4$ | $10^4$ | $-10^3$ | $-10$ | 0.999 |
| $\mathrm{MDP}_{2,7}$ | $[1,15] \times \{1,4,7\} \cup \{1,4,7,10,13\} \times [1,8]$ | 0.9 | 1 | $1/1.05$ | $1/1.1$ | $10^4$ | $-10^3$ | $-10$ | 0.999 |
| $\mathrm{MDP}_{1,1}$ | $[1,15] \times \{6\} \cup \{5\} \times [1,8]$ | 0.9 | 1 | 0.6 | $1/2.5$ | $10^4$ | $-10^3$ | $-10$ | 0.999 |
| $\mathrm{MDP}_{1,2}$ | $[1,15] \times \{6\} \cup \{5\} \times [1,8]$ | 0.9 | 1 | 0.6 | $1/4$ | $10^4$ | $-10^3$ | $-10$ | 0.999 |
| $\mathrm{MDP}_{1,3}$ | $[1,15] \times \{1,4,7\} \cup \{1,4,7,10,13\} \times [1,8]$ | 0.9 | 1 | $1/1.05$ | $1/1.1$ | $10^4$ | $-10^3$ | $-10$ | 0.999 |

Here is the detailed definition of $\texttt{MDP}_{1,n} := (\mathcal{S}, \mathcal{S}^{\text{init}}, \mathcal{S}^{\text{end}}, \mathcal{A}_1, P_1, R_{1,n}, \Gamma_1)(1 \le n \le n_1 = 2)$:

$$\mathcal{A}_1 := \mathcal{A}_{\text{dir}} \cup \{a^{\text{end}}\},$$

$$P_1((s_{\text{cur}}, s_{\text{dest}}), a, (s'_{\text{cur}}, s'_{\text{dest}})) := P((s_{\text{cur}}, s_{\text{dest}}), (a, \texttt{mc}), (s'_{\text{cur}}, s'_{\text{dest}})),$$

$$R_{1,n}((s_{\text{cur}}, s_{\text{dest}}), a, (s'_{\text{cur}}, s'_{\text{dest}})) := R_{\text{dest}} \times \mathbb{1}_{\{s'_{\text{dest}}\}}(s'_{\text{cur}})$$
$$+ \frac{r_0}{\kappa} \times [1 - \mathbb{1}_{\Omega^c_{\text{jams}}}(s_{\text{cur}}) \times \mathbb{1}_{\Omega^c_{\text{jams}}}(s'_{\text{cur}})] \qquad \text{(E.32)}$$
$$+ \frac{r_0}{v_{\text{mc}}} \times \mathbb{1}_{\Omega^c_{\text{jams}}}(s_{\text{cur}}) \times \mathbb{1}_{\Omega^c_{\text{jams}}}(s'_{\text{cur}}),$$

$$\Gamma_1((s_{\text{cur}}, s_{\text{dest}}), a, (s'_{\text{cur}}, s'_{\text{dest}})) := \gamma_0,$$

where $1/\kappa = 2.5, 4$ in $\texttt{MDP}_{1,1}, \texttt{MDP}_{1,2}$, respectively.

### E.2.2 NOTATIONS INTRODUCED

For ease of notation, we first define in this example the transition region $\mathcal{D}_{\text{pause}} := \{((s_{\text{cur}}, s_{\text{dest}}), (a_{\text{dir}}, 0)) \in \mathcal{S}\mathcal{A}^1_{\text{dir}} : s_{\text{cur}} \in \Omega_{\text{jams}} \text{ and } s_{\text{cur}} + a_{\text{dir}} \in \Omega^c_{\text{jams}}, \text{ or } s_{\text{cur}} \in \Omega^c_{\text{jams}} \text{ and } s_{\text{cur}} + a_{\text{dir}} \in \Omega_{\text{jams}}\}$, which contains all the state-action pairs that either enter or leave the region $\Omega_{\text{jams}}$ with traffic jams, $\mathcal{D}^c_{\text{pause}} := \mathcal{S}\mathcal{A}^1_{\text{dir}} - \mathcal{D}_{\text{pause}}$. See Fig. 7 for an illustration of these important subsets of $\mathcal{S}\mathcal{A}^1_{\text{dir}} = \{(s_{\text{cur}}, s_{\text{dest}}), (a_{\text{dir}}, 0)\}$, with the directions of edges omitted because the existence and colors of the edges are always the same when reversing the directions of the edges.

Here $\mathcal{D}_{\text{jams}} := \{(s, (\texttt{dir}, 0)) \in \mathcal{S}\mathcal{A}^1_{\text{dir}} : s_{\text{cur}} \in \Omega_{\text{jams}} \text{ or } s_{\text{cur}} + a_{\text{dir}} \in \Omega_{\text{jams}}\}$ is the set of state-action pairs such that either the agent's current location is in $\Omega_{\text{jams}}$, or the agent intends to move to $\Omega_{\text{jams}}$; $\mathcal{D}^c_{\text{jams}} := \mathcal{S}\mathcal{A}^1_{\text{dir}} - \mathcal{D}_{\text{jams}}$. See Fig. 7 for an illustration of these important subsets of $\mathcal{S}\mathcal{A}_{\text{dir}} = \{(s_{\text{cur}}, s_{\text{dest}}), (a_{\text{dir}}, 0)\}$ appearing in this higher-order policy $\pi^2_{\kappa,*}$, with the directions of edges omitted because the existence and colors of the edges are always the same when reversing the directions of the edges.

### E.2.3 (PARTIAL) POLICIES FOR TARGET MDPs

$$(\pi_{\theta'})_{\text{means}}((s_{\text{cur}}, s_{\text{dest}}), (0, a_{\text{means}})) := \mathbb{1}_{\{\theta'\}}(a_{\text{means}}). \qquad \text{(E.33)}$$

$$((\pi_\theta^{\text{nav}_\kappa})_{\text{dir}} \otimes (\pi_{\theta'})_{\text{means}})((s_{\text{cur}}, s_{\text{dest}}), (a_{\text{dir}}, a_{\text{means}}))$$
$$= (\pi_\theta^{\text{nav}_\kappa})_{\text{dir}}((s_{\text{cur}}, s_{\text{dest}}), (a_{\text{dir}}, 0)) \times \mathbb{1}_{\{\theta'\}}(a_{\text{means}}). \qquad \text{(E.34)}$$

$$\pi^2_{\kappa,*}(s, (\pi_\theta^{\text{nav}_{\kappa'}})_{\text{dir}} \otimes (\pi_{\theta'})_{\text{means}})$$
$$= \mathbb{1}_{\{\theta_{0,*}(s)\}}(\theta) \times \mathbb{1}_{\{\kappa'\}}(\kappa) \times \sum_{a_{\text{dir}} \in \mathcal{A}_{\text{dir}}} (\pi_{\theta_{0,*}(s)}^{\text{nav}_\kappa})_{\text{dir}}(s, (a_{\text{dir}}, 0)) \qquad \text{(E.35)}$$
$$\times [\mathbb{1}_{\{\texttt{car}\}}(\theta') \times \mathbb{1}_{\mathcal{D}_{\text{jams}}}(s, (a_{\text{dir}}, 0)) + \mathbb{1}_{\{\texttt{mc}\}}(\theta') \times \mathbb{1}_{\mathcal{D}^c_{\text{jams}}}(s, (a_{\text{dir}}, 0))],$$

with

$$\theta_{0,*}(s) \in \text{argmin}_{\theta_0} \min_{\theta_1, \cdots, \theta_{\tau-1}, \theta'_0, \theta'_1, \cdots, \theta'_{\tau-1}} \mathbb{E}_{\tau, S_{1:\tau}}[(R^2_\kappa)_{0,\tau}|S_0 = s, A_t = (\pi_{\theta_t}^{\text{nav}_\kappa})_{\text{dir}} \otimes (\pi_{\theta'_t})_{\text{means}}$$
$$\text{(E.36)}$$
$$\text{for any } 0 \le t \le \tau - 1]. $$

$$\pi_\kappa(s, (a_{\text{dir}}, a_{\text{means}})) = (\pi_{\theta_{0,*}(s)}^{\text{nav}_\kappa})_{\text{dir}}(s, (a_{\text{dir}}, 0))$$
$$\times [\mathbb{1}_{\{\texttt{car}\}}(a_{\text{means}}) \times \mathbb{1}_{\mathcal{D}_{\text{jams}}}((s, (a_{\text{dir}}, 0))) \qquad \text{(E.37)}$$
$$+ \mathbb{1}_{\{\texttt{mc}\}}(a_{\text{means}}) \times \mathbb{1}_{\mathcal{D}^c_{\text{jams}}}((s, (a_{\text{dir}}, 0))).$$

### E.2.4 EMBEDDINGS, EMBEDDING GENERATORS AND SKILLS

For $\mathrm{MDP}_{1,n}$ $(1 \leq n \leq n_1 = 2)$ of difficulty 1:

$(e_{\mathrm{decomp}})^1_1 : \mathcal{SA}^1_1 \to \{(s_{\mathrm{cur}}, s_{\mathrm{dest}}, a_{\mathrm{dir}}) : s_{\mathrm{cur}}, s_{\mathrm{dest}} \in \Omega, a_{\mathrm{dir}} \in \mathcal{A}_{\mathrm{dir}} \cup \{a^{\mathbf{end}}\}\}$ is defined as

$$(e_{\mathrm{decomp}})^1_1((s_{\mathrm{cur}}, s_{\mathrm{dest}}), a_{\mathrm{dir}}) := (s_{\mathrm{cur}}, s_{\mathrm{dest}}, a_{\mathrm{dir}}). \tag{E.38}$$

$\overline{\pi}^{\mathrm{nav}_n}_{\mathrm{obstacles}} : (e_{\mathrm{decomp}})^1_1(\mathcal{SA}^1_1) \to [0, 1]$ are as follows:

$$\overline{\pi}^{\mathrm{nav}_n}_{\mathrm{obstacles}}(s_{\mathrm{cur}}, s_{\mathrm{dest}}, a_{\mathrm{dir}}) = \pi^1_{1,n,*}((s_{\mathrm{cur}}, s_{\mathrm{dest}}), a_{\mathrm{dir}}). \tag{E.39}$$

For target $\mathrm{MDP}_\kappa$ $(\kappa \in \mathcal{K})$ of difficulty 2:

$E^1_\alpha : \Theta_{\mathrm{dir}} \to \{(e^1)^{\mathrm{dir}}_\theta : \theta \in \Theta_{\mathrm{dir}}\}$, with $\Theta_{\mathrm{dir}} := \{\mathcal{D}_{\mathrm{pause}}, \mathcal{D}^c_{\mathrm{pause}}\}$, is defined as

$$E^1_\alpha(\theta) := (e^1)^{\mathrm{dir}}_\theta, \tag{E.40}$$

with $(e^1)^{\mathrm{dir}}_\theta : \theta \to \{(s_{\mathrm{cur}}, s_{\mathrm{dest}}, a_{\mathrm{dir}}) : s_{\mathrm{cur}}, s_{\mathrm{dest}} \in \Omega, a_{\mathrm{dir}} \in \mathcal{A}_{\mathrm{dir}} \cup \{a^{\mathbf{end}}\}\}$ given by $(e^1)^{\mathrm{dir}}_\theta((s_{\mathrm{cur}}, s_{\mathrm{dest}}), (a_{\mathrm{dir}}, 0)) := (s_{\mathrm{cur}}, s_{\mathrm{dest}}, a_{\mathrm{dir}})$.

$E^1_\beta : \mathcal{A}_{\mathrm{means}} \to \{(e^1)^{\mathrm{means}}_\theta : \theta \in \mathcal{A}_{\mathrm{means}}\}$ is defined as

$$E^1_\beta(\theta) := (e^1)^{\mathrm{means}}_\theta, \tag{E.41}$$

with $(e^1)^{\mathrm{means}}_\theta : \mathcal{SA}^1_{\mathrm{means}} \to \{0, 1\}$ being defined as $(e^1)^{\mathrm{means}}_\theta((s_{\mathrm{cur}}, s_{\mathrm{dest}}), (0, a_{\mathrm{means}})) := \mathbb{1}_{\{\theta\}}(a_{\mathrm{means}})$, which is the $(\pi_\theta)_{\mathrm{means}}$ we introduced in equation E.33.

$$(e_{\mathrm{decomp}})^2_{2,1}((s_{\mathrm{cur}}, s_{\mathrm{dest}}), ((\pi^{\mathrm{nav}_{\kappa_1}}_\theta)_{\mathrm{dir}} \otimes (\pi_{\theta'})_{\mathrm{means}}))$$
$$:= (\{\mathbb{1}_\theta(((s_{\mathrm{cur}}, s_{\mathrm{dest}}), (a_{\mathrm{dir}}, 0)))\}_{a_{\mathrm{dir}} \in \mathcal{A}_{\mathrm{dir}}}, \{\mathbb{1}_{\mathcal{D}_{\mathrm{jams}}}(((s_{\mathrm{cur}}, s_{\mathrm{dest}}), (a_{\mathrm{dir}}, 0)))\}_{a_{\mathrm{dir}} \in \mathcal{A}_{\mathrm{dir}}},$$
$$\{(\pi^{\mathrm{nav}_{\kappa_1}})_{\mathrm{dir}}(((s_{\mathrm{cur}}, s_{\mathrm{dest}}), a_{\mathrm{dir}}))\}_{a_{\mathrm{dir}} \in \mathcal{A}_{\mathrm{dir}}}, \quad \theta'). \tag{E.42}$$

The higher-order function for selecting the index of the navigation policy and the means of transportation $\overline{\pi}^{\mathrm{transport}} : \{(e_{\mathrm{pause}}, e_{\mathrm{jams}}, e^{\mathrm{nav}}, a_{\mathrm{means}})\} \to [0, 1]$ is as follows:

$$\overline{\pi}^{\mathrm{transport}}(e_{\mathrm{pause}}, e_{\mathrm{jams}}, e^{\mathrm{nav}}, a_{\mathrm{means}})$$
$$= \sum_{a_{\mathrm{dir}} \in \mathcal{A}_{\mathrm{dir}}} e^{\mathrm{nav}}(a_{\mathrm{dir}}) \times e_{\mathrm{pause}}(a_{\mathrm{dir}}) \times [\mathbb{1}_{\{\mathrm{car}\}}(a_{\mathrm{means}}) \times e_{\mathrm{jams}}(a_{\mathrm{dir}}) \tag{E.43}$$
$$+ \mathbb{1}_{\{\mathrm{mc}\}}(a_{\mathrm{means}}) \times (1 - e_{\mathrm{jams}}(a_{\mathrm{dir}}))].$$

### E.2.5 ANALYTICAL RUNNING OF ALGORITHMS

The following goes through how Algs. 5–6 solve these MDPs by constructing MMDPs. All timescales here are $t_{\mathrm{min}} = t_{\mathrm{max}} = +\infty$. See Apps. E.2.1–E.2.4 for the equations needed here as well as the mathematical notations mentioned here.

The student finds an optimal policy $\pi^1_{1,n,*}$ for $\mathrm{MDP}^1_{1,n} := \mathrm{MDP}_{1,n}(1 \leq n \leq n_1)$, and when the teacher provides the trivial embedding $(e_{\mathrm{decomp}})^1_1$ as in equation E.38, the assistant extracts two navigation skills $\overline{\pi}^{\mathrm{nav}_n}_{\mathrm{obstacles}}$ as in equation E.39, with timescales $+\infty$. Both navigation skills are about finding the shortest path between $s_{\mathrm{cur}}$ and $s_{\mathrm{dest}}$, where each grid point is a vertex in the graph, and there are edges between two grid points if the agent can use one step in $\mathcal{A}_{\mathrm{dir}}$ to move between them, with edge weights reflecting the time the agent needs to take for that single step as shown in the reward functions.

Let $\mathrm{MDP}_{2,n} := \mathrm{MDP}_{\kappa_n}, 1 \leq n \leq n_2, \kappa_n \in \mathcal{K}$, be an MDP of difficulty 2, which we have discussed thoroughly in the previous boxes. The student constructs, for each $\mathrm{MDP}_{2,n}$ a two-level MMDP, as in Box 18, to solve it; for $\theta \in \Theta_{\mathrm{dir}}, \theta' \in \mathcal{A}_{\mathrm{means}}$, the timescales for $(\pi^{\mathrm{nav}_\kappa}_\theta)_{\mathrm{dir}} \otimes (\pi_{\theta'})_{\mathrm{means}}$ in $\Pi^1$ are $t_{(\pi^{\mathrm{nav}_\kappa}_\theta)_{\mathrm{dir}} \otimes (\pi_{\theta'})_{\mathrm{means}}} = \min\{t_{(\pi^{\mathrm{nav}_\kappa}_\theta)_{\mathrm{dir}}}, t_{(\pi_{\theta'})_{\mathrm{means}}}\} = +\infty$. Then, the student solves level 2 $\mathrm{MDP}_{2,n}$ and finally solves level 1 of $\mathrm{MDP}_{2,n}$ (See Box 18 for details).

