# OpenReview forum: "Multi-level meta-reinforcement learning with skill-based curriculum"
_ICLR.cc/2026/Conference — Submitted to ICLR 2026_

### Official Review · Reviewer_1rHg · 2025-10-31

**Soundness:** 3
**Presentation:** 2
**Contribution:** 3
**Rating:** 6
**Confidence:** 3

**Summary:**

This paper presents an novel approach to hierarchical RL that tackle complex decision making problem through skill transfer and task decomposition. The proposed framework that compresses MDPs, where policies at lower levels becomes abstract actions at higher levels, is novel and efficient for handling complex tasks. In addition, factors policies into embedding and skills enable effective transfer learning. This paper also incorporates systematic curriculum design to gradually increase task difficulty for efficient skill transfer.

**Strengths:**

The proposed framework provides an efficient solution for planning complex tasks through significant reduction in state/action spaces. Detailed implementation information is provided with practical explanation.

**Weaknesses:**

1. In the main paper, the authors always refer to the appendix for giving more detailed information and examples, which makes the paper hard to read.
2. Heavy teacher dependency, which requires expert knowledge for hints, embeddings, and curriculum structure.
3. Limited empirical comparison with other hierarchical RL methods

**Questions:**

The authors mentioned the proposed framework's effciency in complex grid-world environments.
 However, the experiment are limited to relatively simple grid-world environements. Do you have any other results?

---

### Official Review · Reviewer_q7ZV · 2025-10-31

**Soundness:** 2
**Presentation:** 2
**Contribution:** 2
**Rating:** 2
**Confidence:** 3

**Summary:**

This paper presents a curriculum-based RL framework that recursively compresses MDPs into reusable abstractions (such as options), facilitating hierarchical transfer to more complex tasks. The authors demonstrate the approach in grid-world environments and provide theoretical analysis to support the proposed framework. In my assessment, I used LLMs to polish the text.

**Strengths:**

Strengths:

**Originality:**
The originality is moderate, as most of the underlying ideas, such as hierarchical abstractions, curriculum learning, and skill reuse, have been explored in prior literature. Nevertheless, the authors demonstrate good awareness of related work, providing a comprehensive contextualization, which is a positive aspect.

**Quality:**
Overall, the paper is well written and conceptually sound. However, it is somewhat difficult to follow due to the interleaving of methodological exposition and experimental discussion, which occasionally blurs the focus of the main contribution.

**Clarity:**
From my perspective, this is the main limitation of the current version. The paper’s objectives are clear, but the presentation is overly dense and touches on many interrelated topics that are usually treated separately. At this stage, the work does not yet achieve a coherent flow between overlapping ideas, and additional effort is needed to improve structure and readability.

**Significance:**
Reinforcement learning remains known for its sample inefficiency. Decomposing tasks into simpler subtasks and leveraging knowledge progressively through curriculum learning is a critical and still-open direction for improving efficiency. In this sense, the paper addresses an important and relevant problem.

**Weaknesses:**

Weaknesses:

**Limited empirical validation and lack of baselines**:
The experimental evaluation is restricted to simple, discrete grid-world environments. While these setups illustrate the concepts clearly, they do not demonstrate scalability or generalization to more realistic domains. The paper does not provide baseline comparisons against established hierarchical or curriculum-based RL methods.

**Ambiguous algorithmic implementation**:
 The paper lacks sufficient detail about how it can be implemented or learned in practice. Critical components such as the discovery of
$\mathcal{S}^{init}$, $\mathcal{S}^{end}$, $a^{end}$ are not grounded in concrete learning procedures.

**Dense and unfocused exposition**:
The paper covers a broad range of interconnected ideas (hierarchical RL, curriculum learning, and skill abstraction) in a single narrative. This makes the main technical contribution difficult to isolate and the methodological flow challenging to follow.


Minor suggestions
- The authors do not provide an explanation of $\Omega$ in Box 1; it is only defined later in the Appendix without being referenced in the main text. A brief clarification or forward reference would improve readability.

- The paper seems to have several potential limitations, particularly regarding the design of the curriculum and the application of the method to more complex tasks. However, these issues are not explicitly discussed in a dedicated section, which would have strengthened the overall analysis.

- In Box 1, the text mentions prematurely that MazeBase+ is more challenging than the original version, even though this has not yet been discussed. Readers unfamiliar with the environment may find it difficult to understand why it is considered more challenging. Providing a short explanation or example at that point would improve clarity.

**Questions:**

**Q1.** Grid-worlds are used to illustrate the proposed method, as they conveniently allow a didactic hierarchical decomposition. However, I am curious about the method’s scalability to higher-dimensional or continuous problems. Do the authors foresee potential limitations in such scenarios? In particular, would it be feasible to define or identify sets such as $\mathcal{S}^{init}$ and $\mathcal{S}^{end}$
 in more complex state spaces?

**Q2.** It is not entirely clear to me why the inclusion of the special termination action $a^{end}$ is necessary. Could the authors clarify its role and why explicit termination cannot be handled implicitly?

**Q3.** How was the curriculum used in the experiments designed? Was it manually specified by the authors, or derived through any form of automated process?

---

### Official Review · Reviewer_3u7v · 2025-11-01

**Soundness:** 1
**Presentation:** 1
**Contribution:** 1
**Rating:** 0
**Confidence:** 3

**Summary:**

As far as I can tell, this work aims to identify sequences of actions which themselves perform clear subtasks in the environment. The exact relation to options is not clear to me (and the link to options is only really made in the appendix), however it would seem that the key contributions are that: 1) the combined sequences of actions are parametric which aids their flexibility, 2) the combination of actions is recursive which forms a hierarchy of skills. Transfer learning is highlighted as a particularly important use case for the proposed approach.

**Strengths:**

The paper aims to address an important problem - improving the robust discovery of high-level skills in an environment. I agree that there is room for improvement on this line of work and if the claims of the paper are taken literally then the work does stand to be impactful and will lead to subsequent work.

Given my concerns on clarity I am not able to fairly assess the originality or quality of the work and will aim to work with the authors during the discussion period to flesh out this portion of the review if need be.

**Weaknesses:**

## Clarity
The notation of this work is unclear and inconsistent. It is not clear if this is trying to convey subtleties in the formalism or just presenting things poorly. The bottom paragraph of page 2 serves as one example of this, where the sentence running from line 99 to 101 ("Given an active ... See App. B.1 for detailed definitions") being particularly unhelpful and confusing. This undermines the entire work unfortunately.

The structure of the paper is also really unhelpful. The use of the boxes to squeeze in information just means they are impenetrable and unclear. The boxes themselves are even just positioned out of order, it is unclear when to read them as they appear in between sentences across multiple pages, and are referenced out of order in the main text. It is also unclear why key definitions and discussion around prior work would be relegated to the appendix to such a detriment on clarity.

Lastly, the sentence "ChatGPT and Grok were used to compress several paragraphs to satisfy space constraints" -- I appreciate the fact that this was clearly stated but it probably shouldn't be the last line of the conclusion or just placed arbitrarily at the bottom of the paper.

**Questions:**

I would really appreciate the authors comments on my summary and whether it is even on the right track. I am happy to update my review if very mitigating details arise. However, I am fairly confident that this paper is not ready for publication at ICLR in its current form.

---

### Official Review · Reviewer_n7Tw · 2025-11-09

**Soundness:** 1
**Presentation:** 1
**Contribution:** 1
**Rating:** 0
**Confidence:** 4

**Summary:**

The paper proposes a framework that combines hierarchical reinforcement learning (HRL) and curriculum learning to improve sample efficiency and transfer learning across related tasks. The authors introduce a “multi-level MDP compression” mechanism that recursively abstracts lower-level MDPs into higher-level deterministic ones by treating parametric policy families as single composite actions. This structure, paired with policy factorization into *embeddings* and *skills*, is argued to promote modularity and transfer across levels and tasks. A teacher-student meta-learning setup is then described for organizing curricula based on increasing task difficulty. The framework is evaluated on toy gridworld examples, claiming abstraction and transfer benefits.

**Strengths:**

The work identifies a real challenge which is of interest to the community (hierarchical compositionality and efficient reuse of subskills). Also their integration of multi-level compression and skill-based curricula could, if formalised, provide an elegant lens on abstraction in RL.

**Weaknesses:**

* **Clarity and Writing Quality:** The paper is extremely poorly written and formatted, with unclear notation and undefined terms. E.g.,
  - Use of $\mathcal{A}\mathcal{S}$ instead of $\mathcal{A} \times \mathcal{S}$ throughout.
  - Undefined $S_{1:\tau}$, $A_{0:\tau-1}$, and $R_{0,\tau}$ in the value function definition. Also the value function is only defined for initial states (line 106).
  - The precise definition of difficulty levels is not given, but it is used in statements like "MDP of difficulty 3" on line 113.
  - The notation $(g\_1)\_{I\_1}$ is undefined. Only $g_I$ was somewhat defined, so maybe the authors meant $(g\_{I\_1})\_1$ instead. On page 5 the notation ${\bar\Pi_{G^1}}$ (with a long top bar) is also undefined.
  - Subscripts and superscripts are both used for indexing throughout, which makes it confusing if/when integer superscripts corresponds to their usual "power" meaning.
  - Heavy use of “Boxes” and appendix references makes the main text incoherent and unreadable.
  - In general the Boxes are hard to understand. E.g. Box 1/Figure 1, which is unreferenced/uncontextualized on page 3 in the problem statement, then only referenced on page 7.
* **Questionable Use of LLM Editing:** The authors explicitly state "ChatGPT and Grok were used to compress several paragraphs to satisfy space constraints." at the end of page 9, which likely contributed to the disjoint and unnatural phrasing throughout.
* **Problem Definition:** The problem formulation is ambiguous. It looks similar to a factored MDP (which has a precise definition in the literature), but the authors do not use nor cite it and instead seem to define their own variation of MDPs.
* **Methodology and Theory:** No clear algorithm, pseudocode, or theoretical guarantees are presented. Claims such as “semantic preservation” and “variance reduction” are made without formal justification or proofs.
* **Experimental Validation:** Extremely weak. Only a single gridworld example using value iteration is shown, with no baselines, ablations, or variance reporting. Results do not substantiate the paper’s ambitious claims.
* **Formatting and Accessibility:** The PDF itself is poorly formatted, with large graphical boxes slowing rendering (possibly used to bypass page limits).

Despite addressing an interesting and relevant topic, the paper’s execution is severely deficient. Unclear definitions, sloppy mathematics, incoherent exposition, and unsubstantiated experimental claims make it unsuitable for publication in its current form.

**Questions:**

Please see the weaknesses above. For example:

1. Can the authors provide a clear and formal definition of the proposed “multi-level compression” procedure, including assumptions and proof of semantic preservation? Can the authors provide a pseudocode/algorithm block for it?
2. What exactly is meant by “tensor product structure” in the context of MDPs?
3. How does the “teacher-student-assistant” framework differ algorithmically from prior curriculum learning or meta-RL setups? How does the proposed framework relate to SMDPs in the context of options?
4. Why were no baselines (e.g., value iteration, HIRO, HAC, Option-Critic) included for comparison? Why are there no ablations?
5. What metrics were used to measure sample efficiency or transferability as claimed in the contributions (page 1), and how are they quantified? What specific graphs/results in the paper support those claims (especially since the paper includes no baseline to compare to)?

---

### Meta-Review · Area_Chair_BUkS · 2026-01-10

**Summary:**

This paper aims to study hierarchical abstraction and transfer via a curriculum-based RL framework. This manuscript suffers from severely unclear notation, inconsistent definitions, and an incoherent presentation structure. The technical contributions are not specified with a clear algorithm/pseudocode or formal guarantees, and several key claims (e.g., semantic preservation, variance reduction, sample efficiency, transfer) are unsupported by the provided theory and limited experiments. A substantially revised manuscript with rigorous formalization and a thorough experimental study would be required before this work can be further evaluated.

**Reviewer Concerns:**

Since the authors did not provide a rebuttal, the reviewers’ concerns regarding unclear notation, inconsistent definitions, an incoherent presentation, and limited empirical validation are not addressed.

**Reviewer Scores:**

No reviewer would have changed their scores as no rebuttal is provided.

---

### Decision · Program_Chairs · 2026-01-26

Reject